# Local Function Complexity for Active Learning via Mixture of Gaussian Processes

**Danny Panknin**                                             *danny.panknin@tu-berlin.de*
*Uncertainty, Inverse Modeling and Machine Learning Group, Berlin Institute of Technology, 10587 Berlin, Germany*
*Physikalisch-Technische Bundesanstalt, 10587 Berlin, Germany*

**Stefan Chmiela**                                                  *stefan@chmiela.com*
*Machine Learning Department, Berlin Institute of Technology, 10587 Berlin, Germany*
*BIFOLD-Berlin Institute for the Foundations of Learning and Data, Germany*

**Klaus-Robert Müller**                              *klaus-robert.mueller@tu-berlin.de*
*Machine Learning Department, Berlin Institute of Technology, 10587 Berlin, Germany*
*BIFOLD-Berlin Institute for the Foundations of Learning and Data, Germany*
*Department of Artificial Intelligence, Korea University, Seoul 136-713, South Korea*
*Max Planck Institute for Informatics, 66123 Saarbrücken, Germany*

**Shinichi Nakajima**                                          *nakajima@tu-berlin.de*
*Machine Learning Department, Berlin Institute of Technology, 10587 Berlin, Germany*
*BIFOLD-Berlin Institute for the Foundations of Learning and Data, Germany*
*RIKEN AIP, 1-4-1 Nihonbashi, Chuo-ku, Tokyo, Japan*

**Reviewed on OpenReview:** *https://openreview.net/forum?id=w4MoQ39zmc*

## Abstract

Inhomogeneities in real-world data, e.g., due to changes in the observation noise level or variations in the structural complexity of the source function, pose a unique set of challenges for statistical inference. Accounting for them can greatly improve predictive power when physical resources or computation time is limited. In this paper, we draw on recent theoretical results on the estimation of *local function complexity* (LFC), derived from the domain of *local polynomial smoothing* (LPS), to establish a notion of local structural complexity, which is used to develop a model-agnostic *active learning* (AL) framework. Due to its reliance on pointwise estimates, the LPS model class is not robust and scalable concerning large input space dimensions that typically come along with real-world problems. Here, we derive and estimate the *Gaussian process regression* (GPR)-based analog of the LPS-based LFC and use it as a substitute in the above framework to make it robust and scalable. We assess the effectiveness of our LFC estimate in an AL application on a prototypical low-dimensional synthetic dataset, before taking on the challenging real-world task of reconstructing a quantum chemical force field for a small organic molecule and demonstrating state-of-the-art performance with a significantly reduced training demand.

## 1 Introduction

Inference problems from real-world data often exhibit inhomogeneities, e.g., the noise level, the density of the data distribution, or the complexity of the target function may change over the input space. There exist different approaches from various domains that treat specific kinds of inhomogeneities. For example, Kersting et al. (2007); Cawley et al. (2006) deal with heteroscedasticity by reconstructing a *local noise variance* function that is used to adapt the regularization of the model locally. Some approaches adjust bandwidths locally with respect to the input density (Wang & Wang, 2007; Mackenzie & Tieu, 2004; Moody & Darken, 1989; Benoudjit et al., 2002). Inhomogeneous complexity can also be captured using a combination of several

kernel-linear models with different bandwidths, either learned jointly (Zheng et al., 2006; Guigue et al., 2005) or hierarchically (Ferrari et al., 2010; Bellocchio et al., 2012). The most widely applicable models treat all types of aforementioned inhomogeneities in a unified way (Tresp, 2001; Panknin et al., 2021). Namely, they locally adapt bandwidths or regularization according to the inhomogeneities in noise, complexity, and data density. While this is the path we will pursue, the focus in this work will be on inhomogeneous complexity under the assumption of homoscedastic noise. In addition, we will investigate our proposed estimates in a heteroscedastic setting to demonstrate negligible practical limitations.

Exposing inhomogeneities sheds light on the informativeness of certain locations of the input space, which subsequently can be used to guide the sampling process during training—also known as *active learning* (AL). AL (Kiefer, 1959; MacKay, 1992; Seung et al., 1992; Seo et al., 2000) is a powerful tool to enhance the training process of a model when the acquisition of labeled training data is expensive. It has been successfully implemented in various regression applications like reinforcement learning (Teytaud et al., 2007), wind speed forecasting (Douak et al., 2013), and optimal control (Wu et al., 2020).

Nowadays, *machine learning* (ML) methods are increasingly deployed in physical modeling applications across various disciplines. In that setting, the labels that are necessary for model training are typically expensive as they stem, e.g., from computationally expensive first-principles calculations (Chmiela et al., 2017) or even laboratory experiments. Due to the need for effective training datasets, AL has become an integral part of ever-growing importance in real-world applications, e.g., in the domains of pharmaceutics (Warmuth et al., 2003) and quantum chemistry (Gubaev et al., 2018; Tang & de Jong, 2019; Huang & von Lilienfeld, 2020)—which raises the demand for AL solutions and the importance of AL research in general.

Through the advance of ML in scientific fields that hold the potential for significant impact, new regression problems emerge, for which there is initially only scarce domain knowledge while they simultaneously require thousands to tens of thousands of training samples for ML models to operate at an acceptable performance level. Regarding AL, these two characteristics of regression problems are hard to reconcile:

Due to insufficient domain knowledge on the one hand, a suitable AL approach shall be robust, since unjustified assumptions may result in a training performance that is even worse than *random test sampling*. By *random test sampling*, we refer to the naive training data construction that draws samples i.i.d. according to the test distribution. Additionally, the AL approach shall be model-agnostic since the state-of-the-art is ever-evolving for this particular kind of regression problem. For these reasons, practitioners prefer model-free AL approaches for regression (Wu, 2019) over sophisticated, model-based AL approaches with strong assumptions as the former are inherently robust and model-agnostic by ignoring label information.

On the other hand, it is preferable that an AL approach outperforms *random test sampling* even at large training sizes. In the following, we will measure the AL performance by the *relative required sample size* $\varrho > 0$, which asymptotically equates the performance of $n \cdot \varrho$ active training samples to $n$ random test samples (see Definition 2). Accordingly, we call an AL approach asymptotically superior to *random test sampling*, if $\varrho < 1$. Unfortunately, the performance gain of model-free AL approaches that we observe at small training sizes over *random test sampling* eventually diminishes completely ($\varrho = 1$) with growing training size.

For the described learning task, we therefore require an AL approach that is model-based but comes with mild regularity assumptions at the same time to feature robustness and model-agnosticity to a certain extent.

Recently, Panknin et al. (2021) addressed the outlined AL scenario, where the fundamental idea is to analyze the distribution of the optimal training set of a model in the asymptotic limit of the sample size.

Assuming that this limiting distribution exists, they then propose to sample training data in a *top-down* manner from this very distribution, knowing that with growing sample size the training set will eventually become optimal. By a *top-down* AL approach, we mean an (infinite) training data refinement process $x'_1, x'_2, \ldots$ such that—when optimizing an AL criterion with respect to $\{x_1, \ldots, x_n\}$—$\{x'_1, \ldots, x'_n\}$ asymptotically becomes a respective optimizer as $n \to \infty$. They have shown for the *local polynomial smoothing* (LPS) model class (Cleveland & Devlin, 1988) that the asymptotically optimal distribution exists, whose density furthermore factorizes into contributions of the test density, heteroscedastic noise, and *local function complexity* (LFC)—a measure of the local structural complexity of the regression function. Intuitively, LFC scales with the local amount of variation of the regression function. It is essentially estimated as the reciprocal determinant of the

*locally optimal kernel bandwidth* (LOB) of the LPS model, calibrated for the local effects of the training input density and noise level. Given a small but sufficient training set, these factors can be estimated, allowing the construction of the optimal training density and subsequently enabling the refinement of the training data towards asymptotic optimality.

While the previous work by Panknin et al. (2021) provides a theoretically sound solution to our considered AL scenario, the required pointwise estimates that are inherent to the LPS model class prevent scalability with regard to the input space dimension $d$. The goal of our work is to extend the above approach in a scalable way. The key idea is to build the required estimate of LFC based on the LOB of the related *Gaussian process* (GP) model class that can naturally deal with high input space dimensions. Subsequently, we plug our scalable LFC estimate into the AL framework of the existing method, whose functioning is justified by the method's model-agnostic nature.

It is particularly the almost assumption-free nature of LPS that made the results of Panknin et al. (2021) model-agnostic. To that effect, we base our results on the nonparametric, adaptive bandwidth *Gaussian process regression* (GPR) model to preserve this property. While we lose the strict asymptotic sampling optimality this way, we expect it to be reasonably close due to the model-agnosticity nevertheless. We refer to the resulting training density as the *superior training density* for *locally adaptive models*, by which we refer to models that adapt to the considered inhomogeneities, namely heteroscedasticity and inhomogeneous complexity. Note that it is first and foremost superior for our adaptive bandwidth GPR model.

Fig. 1 summarizes all steps of our contribution and shows how they are interlinked and in which sections they will be discussed. Specifically, we contribute in two ways:

**Theoretical contribution**  Assuming homoscedastic data, we propose a GPR-based LFC estimate which is inspired by the design of the LPS-based LFC estimate. Here, we need to respect the scaling behavior of LOB of GPR that differs from the LPS case, where we use asymptotic results on the scaling of optimal bandwidths for GPR, as described in Sec. 3.3. Making use of the model-agnosticity of the optimal training density of LPS, we replace the herein contained LPS-based LFC estimate for our GPR-based LFC estimate to obtain a *superior training density* for *locally adaptive models*. From this point, we can implement the AL framework by Panknin et al. (2021), for which we propose a novel pool-based formulation. Both, our LFC and density estimate will inherit the scalability of the deployed GPR-based LOB estimate.

**Methodological contribution**  We propose a scalable LOB estimate for GPR as the weighted average of bandwidth candidates, where the weights are given by the gate function of a *sparse mixture of GPs* model—a special case of a *mixture of experts* (MoE) model (Jacobs et al., 1991; Jordan & Jacobs, 1994; Pawelzik et al., 1996). Here, each expert of the MoE is a GPR model that holds an individual, fixed bandwidth candidate. We construct the MoE in *PyTorch* (Paszke et al., 2019) out of well-established components from the related work and design a training objective that is regularized with respect to small bandwidth choices to obtain a robust and reasonable LOB estimate in the end. In addition, we provide an implementation[1] of both, the AL framework and our model. Finally, we propose a novel model-agnostic way of choosing *inducing points* (IPs) of sparse GPR models: Respecting LFC, we place additional basis functions of a kernel method in more complex regions while removing basis functions in simpler regions of the input space.

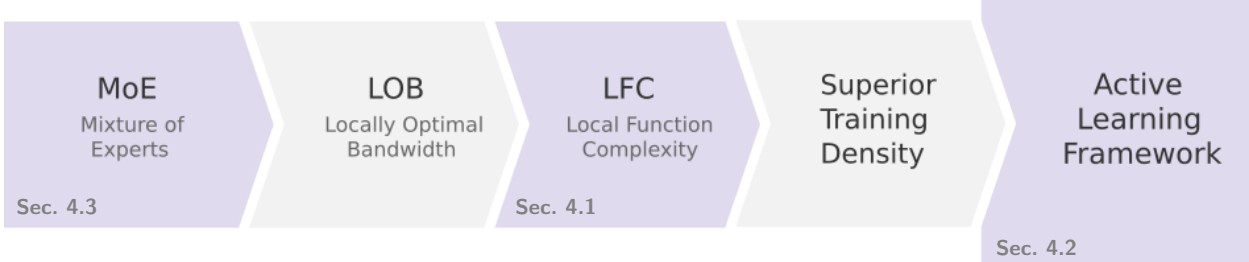

Figure 1: An overview of the steps of our contribution and how they are interlinked. We will elaborate on the main steps in the specified sections.

---

[1] https://github.com/DPanknin/modelagnostic_superior_training

To show the capabilities of our approach, we consider two inhomogeneously complex regression problems:

**The Doppler function** In a controlled setting of 1-dimensional synthetic data, we will first analyze our MoE model and the proposed estimates of LFC and the *superior training density*, where we will demonstrate the asymptotic superior performance of our *superior sampling scheme* and compare to related work. By the *superior sampling scheme*, we refer to i.i.d. sampling from the *superior training density*.

**Force field reconstruction** Quantum interactions exhibit multi-scale behavior due to the complex electronic interactions that give rise to any observable property of interest, like the total energy or atomic forces of a system (Bereau et al., 2018; Yao et al., 2018; Grisafi & Ceriotti, 2019; Ko et al., 2021; Unke et al., 2021a). To demonstrate the scalability of our approach, we consider a force field reconstruction problem of a molecule with 27 dimensions, where the application of the LPS-based AL framework by Panknin et al. (2021) is intractable. Besides the asymptotic superior AL performance, we gain insights into the local structural complexity of this high-dimensional molecular configuration space through visualizations of the scalar-valued LFC function.

We begin by discussing our work in the context of related work in Sec. 2. Next, we give a formal definition of the considered regression problem and the asymptotic AL task, and review asymptotic results for LPS and GPR in Sec. 3. In Sec. 4, we describe our MoE model and derive the GPR-based LFC and *superior training density* estimates. In Sec. 5, we then describe our experiments and results, which will be further discussed in Sec. 6. We finally conclude in Sec. 7.

## 2 Related work

**Choice of MoE, experts and the gate** The common assumption of MoE approaches is that the overall problem to infer is too complex for a single, comparably simple expert. This is the case, for example in regression of nonstationary or piecewise continuous data, and naturally in classification where each cluster shape may follow its own pattern. In such a scenario each expert of the MoE model can specialize in modeling an individual, (through the lens of a single expert) incompatible subset of the data, where the gate learns a soft assignment of data to the experts. Under these assumptions, the hyperparameters of each expert can be tuned individually on the respective assigned data subset. In the light of this paradigm, there exist several instances of mixture of GPs, for example, Tresp (2001); Meeds & Osindero (2006); Yuan & Neubauer (2009); Yang & Ma (2011); Chen et al. (2014).

In our work, we aim to infer a single regression problem, where there is no such segmentation as described above: Each individual (reasonably specified) expert of our mixture model is eventually capable of modeling the whole problem on its own. Yet, if the problem possesses an inhomogeneous structure, the prediction performance can be increased by allowing for a local individual bandwidth choice. Therefore, we deviate from the common MoE paradigm, sharing all those parameters across the experts that describe the regression function. This less common assumption was also made by Pawelzik et al. (1996), where—locally dependent—some experts are expected to perform superior compared to the others.

For the expert and gate components of our MoE model, we focus on the sparse, variational GPR model (see, e.g., Hensman et al. (2015)) trained by stochastic gradient descent. However, there exist other (sparse) GPR approaches that could be considered for the gate or the experts of our MoE model. Some are computationally appealing as they solve for the inducing value distribution analytically (Seeger et al., 2003; Snelson & Ghahramani, 2005; Titsias, 2009) or do not require inducing points in the first place in the case of a full GPR model (see, e.g. Williams & Rasmussen (1996)).

Particularly the expert models of our MoE model can be exchanged for arbitrary sparse and full formulations of GPR, as long as we can access the posterior predictive distribution. We give a short summary of these model alternatives in Appendix B.1 and B.2, which are also included in our provided implementation.

For the gate, however, analytic approaches come with complications as they require labels. Such labels do not exist for the gate, and so we would need to train the MoE in an *expectation maximization* loop, where the likelihood of an expert to have produced a training label functions as a pseudo-label to the gate.

**Alternative nonstationary GPs**  As opposed to a standard GP that features a stationary covariance structure, our MoE model with GP experts of individual bandwidths can be interpreted as a nonstationary GP. Apart from MoE model architectures, there exist other approaches to construct a nonstationary GP:

Closely related to the MoE model, (Rullière et al., 2018) proposed to aggregate individual expert model outputs through a nested GP. Note that this approach is not well-suited for our purpose as the aggregation weights lack the interpretation of a hidden classifier, leaving the subsequent LOB estimation (as in (24)) open. (Gramacy & Lee, 2008) deploy individual stationary GPs on local patches of the input space that are given by an input space partitioning of a tree.

(Gramacy & Apley, 2015) identify subsets of the training data that are necessary to resemble the GP covariance structure at each individual evaluation point. Careful bandwidth choice for each subset then yields a nonstationary GP as well as local bandwidths. Roininen et al. (2019) obtain local bandwidths by imposing a hyperprior on the bandwidth of a GP. Note that our work intends to elaborate the estimation of LFC and model-agnostic superior training, given any estimate of LOB of GPR. We deployed an MoE approach as a simple means to obtain these estimates. The MoE component in our LFC and superior training density estimates may be readily replaced by the approaches of Gramacy & Apley (2015) or Roininen et al. (2019).

In *deep Gaussian process* (DGP) regression (Damianou & Lawrence, 2013), the inputs are mapped through one or more hidden (stationary) GPs. This warping of the input space yields a nonstationary covariance structure of DGP. An approximate DGP model through random feature expansions was exercised by Roininen et al. (2019). Sauer et al. (2023b) discuss active learning for DGP regression. We will implement the DGP model as well as the AL scheme of Sauer et al. (2023b) and compare the AL performance of our superior AL scheme on this very model to demonstrate the model-agnosticity of our work in Sec. 5.1.

**IP selection**  In our work (see Sec. 4.5), we choose the IP locations of the gate and the experts of our MoE model in a diverse and representative way but also in alignment with the structural complexity of the target function, interpreting this choice as a nested AL problem. There are a variety of IP selection approaches in the literature. Zhang et al. (2008) interpreted the choice of IP locations from a geometric view that is similar to ours: They derived a bound on the reconstruction error of a full kernel matrix by a Nyström low-rank approximation in terms of the sum of distances of all training points to their nearest IP. This exposes a local minimum by letting the IP locations be the result of *k-Means clustering*. This choice of IP locations is representative and diverse, while it solely considers input space information. In this sense, our approach extends their work by additionally considering label information. This and our approach draw a fixed number of IPs at once. There are also a lot of Nyström method based IP selection approaches that select columns of the full kernel matrix according to a fixed distribution (Drineas et al., 2005) or one-by-one in a greedy, adaptive way (Smola & Schölkopf, 2000; Fine & Scheinberg, 2001; Seeger et al., 2003). An intensive overview of Nyström method based IP selection methods was given by Kumar et al. (2012), where they also analyzed ensembles of low-rank approximations. We compare our proposed IP choice (29) to the greedy fast forward IP selection approach by Seeger et al. (2003) in Sec. 5.1.

Moss et al. (2023) incorporate a *quality function* into a diverse IP selection process that can be specified flexibly. They consider Bayesian optimization rather than regression, they exercise a quality function proportional to the label. However, other measures of informativeness that are better suited for regression could be deployed. Note that LFC would be a possible candidate for this purpose.

**The AL scenario**  In this work, we consider model-agnostic AL with persistent performance at large (or even asymptotic) training size as opposed to the common AL paradigm that is concerned with small sample sizes. In this sense, we delimit ourselves from AL approaches that are tied to a model, e.g., when they are based on a parametric model (Kiefer, 1959; MacKay, 1992; He, 2010; Sugiyama & Nakajima, 2009; Gubaev et al., 2018), or which refine training data *bottom-up* in a greedy way to maximize its information content at small sample size, where the information is either based on the inputs only (Seo et al., 2000; Teytaud et al., 2007; Yu & Kim, 2010; Wu, 2019; Liu et al., 2021) or also incorporates the labels (Burbidge et al., 2007; Cai et al., 2013). By a *bottom-up* AL approach, we mean a training data refinement process that is constructed by choosing the $n^{\text{th}}$ input $x_n$ as the optimizer of an AL criterion with respect to $\{x_1, \ldots, x_n\}$, when keeping the previously drawn inputs $\{x_1, \ldots, x_{n-1}\}$ (with labels $\{y_1, \ldots, y_{n-1}\}$) fixed.

Our work is therefore complementary to the latter kind of approaches which can be better suited in another AL scenario. For example, if there is enough domain knowledge such that we can deduce a reasonable parametric model without the need for a model change in hindsight, an active sampling scheme based on this model will be best. Our category of interest is for the other case, when domain knowledge is scarce, where we have no idea about the regularity or structure of the problem to decide on a terminal model. Here, for small training sizes (and particularly from scratch), input space geometric arguments (Teytaud et al., 2007; Yu & Kim, 2010; Wu, 2019; Liu et al., 2021) are applied in practice. However, as already noted in the introduction, their benefit is limited to this small sample size regime, which we will demonstrate on our synthetic dataset. They serve reasonably for the initialization of supervised AL approaches, including ours, nevertheless.

Regarding our considered AL scenario, Panknin et al. (2021) have recently proposed an AL framework based on the LPS model class, where training samples are added so as to minimize the *mean integrated squared error* (MISE) in the asymptotic limit. This approach is therefore provably asymptotically superior to *random test sampling*. Additionally, it is robust since the LPS model is almost free of regularity assumptions. Finally, their LPS-based solution then showed to be model-agnostic: On the one hand, this is indicated theoretically by the fact that the LPS model has only indirect influence on the asymptotic form of LFC and the optimal training density since the predictor is asymptotically not involved (see, e.g., Eq. (8)); On the other hand, this is validated empirically by assessing the performance of their LPS-based training dataset construction under *reasonable* model change in hindsight. This model change is restricted to *locally adaptive models*. Here, Panknin et al. (2021) observed a consistent performance superior to *random test sampling* when training a random forest model and a *radial basis function* (RBF)-network (Moody & Darken, 1989), using their proposed training dataset.

**AL for classification**   Note that the outlined AL scenario can be solved more easily for classification:

Here, AL is intuitively about the identification and rendering of the decision boundaries, which is inherently a model-agnostic task. In addition, since the decision boundaries are a submanifold of the input space $\mathcal{X}$, a substantial part of $\mathcal{X}$ can be spared when selecting training samples. Therefore, AL for classification leverages the decay of the generalization error from a polynomial to an exponential law (Seung et al., 1992) over *random test sampling*. For the above reasons, AL for classification has been applied successfully in practice (Lewis & Gale, 1994; Roy & McCallum, 2001; Goudjil et al., 2018; Warmuth et al., 2003; Pasolli & Melgani, 2010; Saito et al., 2015; Bressan et al., 2019; Sener & Savarese, 2018; Beluch et al., 2018; Haut et al., 2018; Tong & Chang, 2001; He, 2010). In contrast, the performance gain of AL for regression is more limited in the sense that, under weak assumptions, we are tied to the decay law of the generalization error of *random test sampling* (Györfi et al., 2002; Willett et al., 2005).

**GP uncertainty sampling**   There exists a lot of research on AL for GPR (Seo et al., 2000; Pasolli & Melgani, 2011; Schreiter et al., 2015; Yue et al., 2020), which is typically based on minimizing prediction uncertainties of the model. With our proposed AL approach being based on GPR models, this research area is the most related competitor to our work.

For a standard GPR model, the prediction uncertainty is the higher the farther away we move from training inputs. In this way, *GP uncertainty sampling* samples (pseudo-)uniformly from the input space which makes up for a low-dispersion sequence (Niederreiter, 1988) (see Definition 5). Note that standard GP uncertainty sampling is an input space geometric argument since it does not depend on the regression function to infer. As already indicated in the introduction and as we will show in Sec. 5.1, input space geometric arguments feature no benefit regarding asymptotic AL performance.

Since our model is a mixture of GPR experts, it is straightforward to derive its uncertainty as a *mixture of Gaussian process uncertainties* (MoGPU) by simply weighting the predictive variances of all experts with respect to the gate output (see (32) for a definition). As opposed to GP uncertainty sampling, MoGPU can cope with structural inhomogeneities. Therefore, we consider MoGPU as a fair baseline competitor to our *superior sampling scheme* and compare both in Sec. 5.1.

## 3  Preliminaries

We will now give a formal definition of the regression task, the AL objective and LOB, and a short review of the asymptotic results on LFC and the optimal training distribution of the LPS model in Sec. 3.1 and 3.2. Then we recap asymptotic results on the optimal bandwidth of GPR in Sec. 3.3 and known models in Sec. 3.4 that will serve as building blocks of our proposed adaptive bandwidth MoE model later on.

In the following, we denote by $\boldsymbol{diag}(z) \in \mathbb{R}^{d \times d}$ the diagonal matrix with the entries of the vector $z \in \mathbb{R}^d$ on its diagonal and by $\mathcal{I}_d = \boldsymbol{diag}(\mathbb{1}_d)$ the identity matrix, where $\mathbb{1}_d$ is the vector of ones in $\mathbb{R}^d$.

### 3.1  Formal definition of the regression task and AL objective

Let $f$ be the target regression function defined on an input space $\mathcal{X} \subset \mathbb{R}^d$ that we want to infer from noisy observations $y_i = f(x_i) + \varepsilon_i$, where $x_i \in \mathcal{X}$ are the training inputs and $\varepsilon_i$ is independently drawn noise from a distribution with mean $\mathbb{E}[\varepsilon_i] = 0$ and local noise variance $\mathbb{V}[\varepsilon_i] = v(x_i)$. We denote a training set by $(\boldsymbol{X}_n, \boldsymbol{Y}_n)$, where $\boldsymbol{X}_n = (x_1, \ldots, x_n) \in \mathcal{X}^n$ and $\boldsymbol{Y}_n = (y_1, \ldots, y_n) \in \mathbb{R}^n$. For a given model class $\widehat{f}$ that returns a predictor $\widehat{f}_{\boldsymbol{X}_n, \boldsymbol{Y}_n}$ for a training set $(\boldsymbol{X}_n, \boldsymbol{Y}_n)$, we can define the pointwise *conditional mean squared error* of $\widehat{f}$ in $x \in \mathcal{X}$, given $\boldsymbol{X}_n$, by

$$\mathrm{MSE}\left(x, \widehat{f}|\boldsymbol{X}_n\right) = \mathbb{E}_{\boldsymbol{Y}_n}\left[(\widehat{f}_{\boldsymbol{X}_n, \boldsymbol{Y}_n}(x) - f(x))^2\right] = \mathbb{E}_{\boldsymbol{\varepsilon}_n}\left[(\widehat{f}_{\boldsymbol{X}_n, f(\boldsymbol{X}_n)+\boldsymbol{\varepsilon}_n}(x) - f(x))^2\right]. \tag{1}$$

Note that via marginalization the conditional mean squared error is no function of the training labels $\boldsymbol{Y}_n$. Given a test probability density $q \in \mathcal{C}^0\left(\mathcal{X}, \mathbb{R}_+\right)$ such that $\int_{\mathcal{X}} q(x)dx = 1$, the *conditional mean integrated squared error* of the model under the given training set is then defined as

$$\mathrm{MISE}\left(q, \widehat{f}|\boldsymbol{X}_n\right) = \int_{\mathcal{X}} \mathrm{MSE}\left(x, \widehat{f}|\boldsymbol{X}_n\right) q(x)dx. \tag{2}$$

With these preparations, the AL task is to construct a training dataset $(\boldsymbol{X}_n', \boldsymbol{Y}_n')$ such that

$$\boldsymbol{X}_n' \approx \arg\min_{\boldsymbol{X}_n \in \mathcal{X}^n} \mathrm{MISE}\left(q, \widehat{f}|\boldsymbol{X}_n\right). \tag{3}$$

### 3.2  Locally optimal bandwidths, function complexity, and optimal training

Let $\widehat{f}^\Sigma$ be a family of *kernel machines* which is characterized by a positive definite bandwidth matrix parameter $\Sigma \in \mathbb{S}_{++}^d$ of an RBF kernel $k^\Sigma(x, x') := |\Sigma|^{-1} \phi(\|\Sigma^{-1}(x - x')\|)$ for a monotonically decreasing function $\phi \colon \mathbb{R}_+ \to \mathbb{R}_+$. The well known Gaussian kernel is for example implemented by $\phi(z) = \exp\{-\frac{1}{2}z^2\}$.

Given a bandwidth space $\mathcal{S} \subseteq \mathbb{S}_{++}^d$ we define the LOB function of $\widehat{f}$ by

$$\Sigma_{\widehat{f}}^n(x) = \arg\min_{\Sigma \in \mathcal{S}} \mathrm{MSE}\left(x, \widehat{f}^\Sigma|\boldsymbol{X}_n\right), \tag{4}$$

assuming that this minimizer uniquely exists for all $x \in \mathcal{X}$.

Denote by $m_Q^\Sigma$ the predictor of the LPS model of order $Q$ under bandwidth $\Sigma$ and by $\Sigma_Q^n := \Sigma_{m_Q}^n$ the LOB function (4) of LPS, if it is well-defined. This is the case, e.g., for the *isotropic* bandwidths space $\mathcal{S} = \{\sigma\mathcal{I}_d \mid \sigma > 0\}$ under mild assumptions[2], where we particularly can write $\Sigma_Q^n(x) = \sigma_Q^n(x)\mathcal{I}_d$. We refer to Appendix A for details on the LPS model and asymptotic results. For the optimal predictor

$$\widehat{f}_{\mathrm{LPS}}^Q := m_Q^{\Sigma_Q^n(x)}(x) \tag{5}$$

---

[2]For LOB being well-defined in the isotropic case, we generally require a non-vanishing bias and variance in terms of a bias-variance-decomposition of the MSE of the predictor in x, for all $x \in \mathcal{X}$. See, e.g., Eq. (36) for the LPS predictor $m_Q$, or Silverman (1986); Wand & Jones (1994) in more general.

of LPS, letting $\widehat{f} = \widehat{f}_{\mathrm{LPS}}^{Q}$ in Eq. (3), Panknin et al. (2021) have shown that there exists an optimal training density $p_{\mathrm{Opt}}^{Q,n}$ that allows the optimal training inputs in Eq. (3) to be asymptotically obtained by independently and identically sampling $\boldsymbol{X}_n' \sim p_{\mathrm{Opt}}^{Q,n}$. They have also shown that this density exhibits a closed-form

$$p_{\mathrm{Opt}}^{Q,n}(x) \propto \left[\mathfrak{C}_Q^n(x)q(x)\right]^{\frac{2(Q+1)+d}{4(Q+1)+d}} v(x)^{\frac{2(Q+1)}{4(Q+1)+d}} (1+o(1)), \tag{6}$$

where for an arbitrary training dataset $(\boldsymbol{X}_n, \boldsymbol{Y}_n)$ with $\boldsymbol{X}_n \sim p$, the LFC of LPS is defined by

$$\mathfrak{C}_Q^n(x) := \left[\frac{v(x)}{p(x)n}\right]^{\frac{d}{2(Q+1)+d}} \left|\Sigma_Q^n(x)\right|^{-1} = \left[\frac{v(x)}{p(x)n}\right]^{\frac{d}{2(Q+1)+d}} \sigma_Q^n(x)^{-d}. \tag{7}$$

The LFC in (7) asymptotically solely depends on the behavior of $f$ as opposed to $p, v,$ and $n$: It scales with the local variation of $f$ in the vicinity of $x$. For example,

$$\mathfrak{C}_1^n(x) \propto \boldsymbol{trace}(D_f^2(x))^{\frac{2d}{2(Q+1)+d}}(1+o(1)) \tag{8}$$

is a function of the trace of the Hessian of $f$ (Fan et al., 1997).

The optimal density $p_{\mathrm{Opt}}^{Q,n}$ in (6) implies that we require more training data where the problem is locally more complex (large $\mathfrak{C}_Q^n$) or noisy (large $v$), or where test instances are more likely (large $q$). As already noted in the introduction, the results to LFC and the optimal training density of LPS indicate their problem intrinsic nature, as they reflect no direct dependence on the LPS model except for the order $Q$. Note that for $f \in \mathcal{C}^\alpha(\mathcal{X}, \mathbb{R})$, there is a canonical choice $Q = \lceil \alpha \rceil - 1$ of the LPS model order. When deriving LFC under this canonical-order model, we consider the dependence of the associated LFC and the optimal training density on $Q$ negligible, as its choice is driven by the problem intrinsic regularity.

In practice, we obtain $\boldsymbol{X}_n' \sim p_{\mathrm{Opt}}^{Q,n}$ by estimating Eq. (6) and (7) from $(\boldsymbol{X}_{n'}, \boldsymbol{Y}_{n'})$ with $\boldsymbol{X}_{n'} \sim p$ for an arbitrary training density $p$, where $n' < n$, followed by adding the remaining $n - n'$ inputs appropriately (see Sec. 4.2).

The construction of $p_{\mathrm{Opt}}^{Q,n}$ crucially depends on reliable estimates of LOB as the key ingredient for the estimation of LFC. While Panknin et al. (2021) provide such an estimate based on Lepski's method (Lepski, 1991; Lepski & Spokoiny, 1997), it does not scale well with increasing input space dimension $d$. This is because pointwise estimates suffer from the *curse of dimensionality* regarding robustness and computational feasibility. The goal of this work is to implement the above AL framework but based on a functional LOB estimate in the domain of GPR instead of LPS, since the GPR model class can naturally deal with high input space dimensions (Williams & Rasmussen, 1996). Relying on the model-agnosticity, we expect that LOB estimates based on LPS can be exchanged for LOB estimates based on GPR in the formulation of LFC and $p_{\mathrm{Opt}}^{Q,n}$ when matching the degree $Q$ to the smoothness of the regression function appropriately.

### 3.3 On the scaling of GPR bandwidths

The major difference between LPS and GPR is that we keep a fixed model complexity—in the sense of the number of basis functions—in the former while there is varying model complexity in the latter as we add further training instances. E.g., under the Gaussian kernel the model complexity of GPR grows infinitely. When the regularity of the kernel and the target function $f$ match, then, as soon as the training size $n$ becomes large enough, there is no need for further shrinkage of the bandwidth to reproduce $f$ with GPR in the asymptotic limit. In particular, given enough samples, there is no need for local bandwidth adaption.

However, there is a mismatch if $f \in \mathcal{C}^\alpha(\mathcal{X}, \mathbb{R})$ is $\alpha$-times continuously differentiable since the Gaussian kernel is infinitely often continuously differentiable. As shown by Van der Vaart et al. (2007; 2009), in order to obtain optimal *minimax*-convergence of the predictor (except for logarithmic factors), the associated (global) bandwidth has to follow the asymptotic law

$$\Sigma_{\mathrm{GPR}}^n \propto n^{-\frac{1}{2\alpha+d}}. \tag{9}$$

Note that for $f \in \mathcal{C}^\alpha(\mathcal{X}, \mathbb{R})$, where the theoretical results of LPS apply, the scaling factor $n^{-\frac{1}{2\alpha+d}}$ of LOB in sample size matches exactly for both classes, LPS and GPR. In our work, we will use (9) to deduce a GPR-based LFC estimate in analogy to the LPS-based LFC estimate (7) by Panknin et al. (2021).

### 3.4 Preliminaries on the applied models

We will now introduce the models that we implement in this work. For the RBF-kernel $k$, we define the kernel matrix between $X \in \mathcal{X}^n$ and $X' \in \mathcal{X}^m$ as $\boldsymbol{K}^\Sigma(X, X') = \left[k^\Sigma(x, x')\right]_{x \in X, x' \in X'}$. As a shorthand notation we furthermore define $\boldsymbol{K}^\Sigma(X) := \boldsymbol{K}^\Sigma(X, X)$.

#### 3.4.1 Sparse variational Gaussian processes

We define the sparse GPR model $\widehat{y} \sim \mathcal{SVGP}(\theta)$ (see, e.g. Williams & Rasmussen (1996); Hensman et al. (2015)) as follows: The sparse GP is described by the (hyper-) parameters $\theta = (\mu, \lambda, \widehat{v}, \Sigma, \boldsymbol{X}_\dagger, \boldsymbol{\mu}_\dagger, \boldsymbol{S}_\dagger)$, which are the global constant prior mean $\mu$, the regularization parameter $\lambda$, the label noise variance function $\widehat{v}$, the bandwidth matrix $\Sigma$ of the kernel and the prior distribution, given by the IP locations $\boldsymbol{X}_\dagger \in \mathcal{X}^m$ as well as their inducing value distribution, characterized by the moments $\boldsymbol{\mu}_\dagger$ and $\boldsymbol{S}_\dagger$. That is, for the inducing values $\boldsymbol{Y}_\dagger$ of $\boldsymbol{X}_\dagger$ we assume $\boldsymbol{Y}_\dagger = \widehat{y}(\boldsymbol{X}_\dagger) \sim \mathcal{N}(\cdot; \boldsymbol{\mu}_\dagger, \boldsymbol{S}_\dagger)$. Here, the degree of sparsity is described by $m$ IPs: This number can be fixed in advance or gradually increased with training size $n$, where the increase $m_n = o[n]$ is typically much slower than $n$. If we can assume homoscedastic noise, we let $\widehat{v}(x) \equiv \sigma_\varepsilon^2$.

The sparse GP then outputs

$$\widehat{y}(\boldsymbol{X}_*) \sim \mathcal{N}(\cdot; \boldsymbol{\mu}^*(\boldsymbol{X}_*), \boldsymbol{C}^*(\boldsymbol{X}_*) | \theta_e) \tag{10}$$

for the mean function

$$\boldsymbol{\mu}^*(\boldsymbol{X}_*) = \boldsymbol{K}_{*\dagger} \boldsymbol{K}_\dagger^{-1/2}(\widetilde{\boldsymbol{\mu}}_\dagger - \boldsymbol{K}_\dagger^{-1/2} \boldsymbol{\mu}(\boldsymbol{X}_\dagger)) + \boldsymbol{\mu}(\boldsymbol{X}_*), \tag{11}$$

and the covariance function

$$\boldsymbol{C}^*(\boldsymbol{X}_*) = \lambda \left[ \boldsymbol{K}_* + \boldsymbol{K}_{*\dagger} \boldsymbol{K}_\dagger^{-1/2}(\widetilde{\boldsymbol{S}}_\dagger - \mathcal{I}_m) \boldsymbol{K}_\dagger^{-1/2} \boldsymbol{K}_{*\dagger}^\top \right] + \boldsymbol{diag}(\widehat{v}(\boldsymbol{X}_*)), \tag{12}$$

where $\widetilde{\boldsymbol{\mu}}_\dagger = \boldsymbol{K}_\dagger^{-1/2} \boldsymbol{\mu}_\dagger$ and $\widetilde{\boldsymbol{S}}_\dagger = \boldsymbol{K}_\dagger^{-1/2} \boldsymbol{S}_\dagger \boldsymbol{K}_\dagger^{-1/2}$ are the whitened moments of the inducing value distribution (Pleiss et al. (2020), Sec. 5.1), and we have defined $\boldsymbol{K}_* = \boldsymbol{K}^\Sigma(\boldsymbol{X}_*)$, $\boldsymbol{K}_\dagger = \boldsymbol{K}^\Sigma(\boldsymbol{X}_\dagger)$ and $\boldsymbol{K}_{*\dagger} = \boldsymbol{K}^\Sigma(\boldsymbol{X}_*, \boldsymbol{X}_\dagger)$.

We choose $\boldsymbol{\mu}$ to be the constant mean function, i.e., $\boldsymbol{\mu}(X) = \mu \mathbb{1}_n$ for $X \in \mathcal{X}^n$, noting that other mean functions are possible. Note that test predictions $\widehat{f}_{\mathrm{GP}}(x) = \boldsymbol{\mu}^*(x)$ are given by Eq. (11).

**The training objective** Let $P$ denote the prior distribution of the inducing function values $\boldsymbol{Y}_\dagger$ of the IPs $\boldsymbol{X}_\dagger$ and let $Q$ denote a tractable *variational distribution* intended to approximate $P(\cdot) \approx Q(\cdot | \boldsymbol{X}_n, \boldsymbol{Y}_n)$. In *variational inference*, we want to minimize the *Kullback-Leibler divergence* $\mathcal{KL}[Q \| P]$ between $Q$ and $P$, which is equivalent to maximizing the data log-evidence $\log(P(\boldsymbol{X}_n, \boldsymbol{Y}_n))$. As a tractable approximation, we maximize the *evidence lower bound* (ELBO), given by

$$\mathbb{E}_{u \sim Q(\cdot | \boldsymbol{X}_n, \boldsymbol{Y}_n)} \log(P(\boldsymbol{X}_n, \boldsymbol{Y}_n | u)) - \mathcal{KL}[Q(\cdot | \boldsymbol{X}_n, \boldsymbol{Y}_n) \| P]$$
$$\approx \frac{1}{n} \sum_{b=1}^{n} P_b - \mathcal{KL}[Q(\cdot | \boldsymbol{X}_n, \boldsymbol{Y}_n) \| P], \tag{13}$$

where $P_b$ is the predictive log-likelihood in $x_b$, marginalized over the variational distribution $Q$, that is,

$$P_b := \mathbb{E}_{u \sim Q(\cdot | \boldsymbol{X}_n, \boldsymbol{Y}_n)} \log \int P(y_b | f) P(f | u, x_b) df. \tag{14}$$

#### 3.4.2 Sparse mixture of experts

Given a finite set of expert models $\widehat{y}_l$ that are parameterized by $\theta_{e_l}$, the MoE model is given by

$$\widehat{f}_{\mathrm{MoE}}(x) = \sum_{l=1}^{L} G(x)_l \widehat{y}_l(x), \tag{15}$$

where the *gate* $G: \mathcal{X} \to [0, 1]^L$ is a probability assignment of an input $x$ to the experts. In particular, it holds $\sum_{l=1}^{L} G(x)_l \equiv 1$ and $G(x)_i \geq 0, \forall x \in \mathcal{X}$ and $1 \leq i \leq L$.

We implement the approach of Shazeer et al. (2017) to model the gate $G$ as follows: For the *softmax* function

$$\mathbf{soft\,max}(\boldsymbol{a})_i := \exp\{\boldsymbol{a}_i\} \Big/ \sum\nolimits_{l=1}^{L} \exp\{\boldsymbol{a}_l\}, \tag{16}$$

where $\boldsymbol{a} \in \mathbb{R}^L$, Shazeer et al. (2017) propose to set

$$G(x) = \mathbf{soft\,max}(\widetilde{h}_1(x), \ldots, \widetilde{h}_L(x)), \quad \text{where} \quad \widetilde{h}_{i_l}(x) = \begin{cases} h_{i_l}(x) & , l < \kappa \\ -\infty & , l \geq \kappa + 1 \end{cases} \tag{17}$$

for an adequate permutation $(i_1, \ldots, i_L)$ of $\{1, \ldots, L\}$ such that $h_{i_l}(x) > h_{i_{l+1}}(x)$ are ordered decreasingly. Here, $h_l(x) = g_l(x) + \mathcal{N}(0, \mathfrak{s}_l^2)$ is a noisy version of single-channel gating models $g_l$ with parameters $\theta_{g_l}$. Note that these models can be chosen freely and may also deviate from the choice of expert models $\widehat{y}_l$.

The cutoff value $1 \leq \kappa \leq L$ controls the sparsity of the MoE, as it enforces the minor mixture weights to strictly equal zero. For stability reasons, during the training, we give each expert a chance to become an element of the top-$\kappa$ components by adding independent Gaussian noise $\mathcal{N}(0, \mathfrak{s}_l^2)$ before thresholding, where $\mathfrak{s} \in \mathbb{R}_{++}^L$ is another hyperparameter to set or learn. This noisy gating prevents a premature discarding of initially underperforming experts.

The overall MoE hyperparameter set is thus given by

$$\Theta = (\{\theta_{e_l}\}_{l=1}^L, \{\theta_{g_l}\}_{l=1}^L, \kappa, \mathfrak{s}). \tag{18}$$

## 4 Estimating locally optimal bandwidths via mixture of Gaussian processes

In this section, we derive our main contribution, namely the GPR-based AL framework, which we summarized in Fig. 2. We first derive our GPR-based estimates of LFC and the *superior training density* in Sec. 4.1 (Fig. 2, B). Combining this estimate with the AL framework from Panknin et al. (2021), we obtain a GPR-based, model-agnostic *superior sampling scheme* in Sec. 4.2 (Fig. 2, C). Next, we describe our GPR-based MoE model in Sec. 4.3 (Fig. 2, A) of which we obtain the required LOB estimate of GPR for the estimation of *superior training density*. The scalability of this estimate enables the application of our *superior sampling scheme* to problems of high input space dimensions. We then give details on the training of the MoE in Sec. 4.4 and finally propose an LFC-based IP selection method in Sec. 4.5. We summarize the pseudo-code of our *superior sampling scheme* in Algorithm 1.

### 4.1 GPR-based LFC and the *superior training density*

Let $\Sigma_{\mathrm{GPR}}^n(x)$ denote the LOB function (4) of GPR. Inspired by the results to LFC and the *superior training density* of LPS in Eq. (7) and (6), we are able to deduce their GPR-based analog. Here, we need to take into account that GPR adapts universally[3] to functions $f \in \mathcal{C}^\alpha(\mathcal{X}, \mathbb{R})$, as opposed to LPS, whose decay rate is determined by the specified polynomial order $Q$. The idea of the LFC estimate was to adjust LOB appropriately so that it becomes invariant under the influence of the training density, heteroscedasticity, and its global decay with respect to the training size $n$.

Combining the local effective sample size $p(x)n$ with the scaling result of the global $\Sigma_{\mathrm{GPR}}^n$ in (9) from Sec. 3.3, we propose an LFC estimate for GPR as follows (see Appendix C for proof details).

**Theorem 1** (LFC of GPR). *For $f \in \mathcal{C}^\alpha(\mathcal{X}, \mathbb{R})$, $\boldsymbol{X}_n \sim p$ and homoscedastic noise, the GPR-based LFC estimate of $f$ in $x \in \mathcal{X}$ is asymptotically given by*

$$\mathfrak{C}_{GPR}^n(x) := \left[\frac{1}{p(x)n}\right]^{\frac{d}{2\alpha+d}} |\Sigma_{GPR}^n(x)|^{-1}. \tag{19}$$

In analogy to Eq. (7), $\mathfrak{C}_{\mathrm{GPR}}^n$ measures the structural complexity of $f$, as it asymptotically does not depend on $p$, $v$ and $n$. Note that the LPS model provides no explicit way to adapt to the local noise variance $v(x)$,

---

[3]That is, the MISE decays at the minimax-rate $n^{-\frac{2\alpha}{2\alpha+d}}$ of nonparametric models.

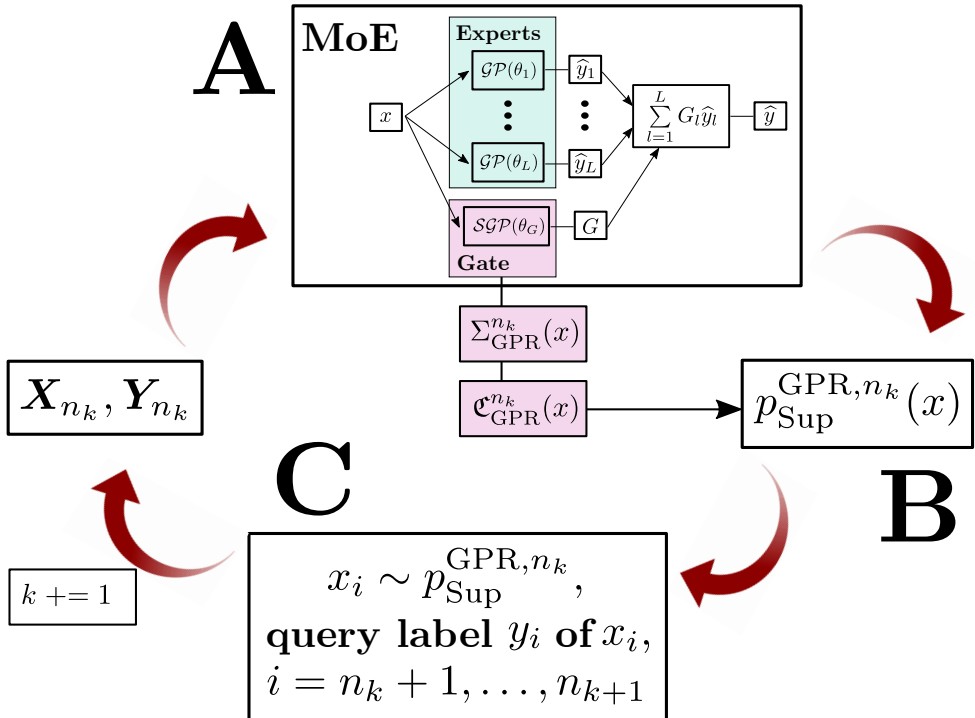

Figure 2: The proposed AL framework.

such that the LOB of LPS scales with respect to $v$ to address heteroscedasticity (see (36) in Appendix A). For GPR, we have made the restriction of homoscedastic noise in the definition of $\mathfrak{C}^n_{\mathrm{GPR}}$ in Theorem 1, since we are not aware of a theory on the scaling of GPR-based LOB with respect to heteroscedasticity. However, as opposed to LPS, a heteroscedastic GPR model provides an explicit way to adapt to the local noise variance $v(x)$ via regularization. As a result, we observe only very little influence of heteroscedasticity on LOB function, which we will demonstrate in Sec. 5.1. Thus, $\mathfrak{C}^n_{\mathrm{GPR}}$ will be sufficiently calibrated in a heteroscedastic scenario, making it a reasonable estimate of LFC in practice without further restrictions.

Now, when putting $\mathfrak{C}^n_{\mathrm{GPR}}$ into Eq. (6) with $Q = \alpha - 1$, we obtain the *superior training density*

$$p^{\mathrm{GPR},n}_{\mathrm{Sup}}(x) \propto [\mathfrak{C}^n_{\mathrm{GPR}}(x)q(x)]^{\frac{2\alpha+d}{4\alpha+d}} \, v(x)^{\frac{2\alpha}{4\alpha+d}}(1 + o(1)). \tag{20}$$

For even $\alpha \in \mathbb{N}$ with $Q = \alpha - 1$ and $\mathfrak{C}^n_{\mathrm{GPR}} \equiv \mathfrak{C}^n_Q$, $p^{\mathrm{GPR},n}_{\mathrm{Sup}}$ and $p^{Q,n}_{\mathrm{Opt}}$ coincide, which proved to be optimal for LPS. In this sense, (20) generalizes (6) to the general case of $\alpha \in \mathbb{R}_+$, where we expect that the true optimal training density for $f \in \mathcal{C}^\alpha(\mathcal{X})$ will not deviate by a lot from $p^{\mathrm{GPR},n}_{\mathrm{Sup}}$. Since LPS and GPR are related models, we furthermore expect $\mathfrak{C}^n_{\mathrm{GPR}}$ to be similar to $\mathfrak{C}^n_Q$ for the appropriate order $Q$.

Note that for $f \in \mathcal{C}^\infty(\mathcal{X})$, we let $\alpha \to \infty$ in Eq. (19) and (20) to obtain

$$\mathfrak{C}^n_{\mathrm{GPR}}(x) = |\Sigma^n_{\mathrm{GPR}}(x)|^{-1} \quad \text{and} \quad p^{\mathrm{GPR},n}_{\mathrm{Sup}}(x) \propto [\mathfrak{C}^n_{\mathrm{GPR}}(x)q(x)v(x)]^{\frac{1}{2}}. \tag{21}$$

While $\boldsymbol{X}_n \sim p^{\mathrm{GPR},n}_{\mathrm{Sup}}$ will not be optimal for our model, we expect it to be asymptotically superior to the naive *random test sampling*, i.e., $\boldsymbol{X}_n \sim q$, due to the model-agnosticity of the LPS-based result. To assess the asymptotic performance of a training density $p$ (such as $p^{\mathrm{GPR},n}_{\mathrm{Sup}}$), let us first observe the following:

For regression problems and under weak assumptions the law of the MISE does not change with respect to $p$, except for a constant multiple (Györfi et al., 2002; Willett et al., 2005). Accordingly, the number of actively selected training samples ($\sim p$) that are required to achieve the same level of accuracy of *random test sampling* is given by a constant $\varrho > 0$. Formally, we can define $\varrho$ as follows.

---

**Algorithm 1:** Superior training data process $(\boldsymbol{X}_n, \boldsymbol{Y}_n)_{n\in\mathbb{N}}$ with labels $\boldsymbol{Y}_n$ of training inputs $\boldsymbol{X}_n \xrightarrow{d} p_{\mathrm{Sup}}^{\mathrm{GPR},n}$

---

**Input**

1: Intermediate training sizes $(n_k)_{k\in\mathbb{N}_0}$ with $n_k < n_{k+1}, \forall k \in \mathbb{N}_0$ for reestimation
2: A labeled validation set $\boldsymbol{X}_{\mathrm{val}}, \boldsymbol{Y}_{\mathrm{val}}$
3: An input generating process $\boldsymbol{X}_{pool} \in \mathcal{X}^N$ with $\boldsymbol{X}_{pool} \sim p_\mathcal{X}$
4: The label oracle $\boldsymbol{y}\colon \mathcal{X} \to \mathbb{R}$
5: (optional) The test density $q$
6: (optional) The intrinsic dimension $\delta \le d$ of the input space $\mathcal{X}$
7: (optional) The regularity $\alpha$ of the target function $f \in \mathcal{C}^\alpha(\mathcal{X}, \mathbb{R})$

---

**Output**

8: (Infinite) training data process $(\boldsymbol{X}_n, \boldsymbol{Y}_n)$ with labels $\boldsymbol{Y}_n$ of training inputs $\boldsymbol{X}_n \sim p_{\mathrm{Sup}}^{\mathrm{GPR},n}$

**Procedure**

---

9: ▷ Initialization
10: Estimate pool density $\widehat{p}_\mathcal{X}$ based on $\boldsymbol{X}_{pool}$          ▷ e.g., using kernel density estimation
11: **if** $q$ is not specified **then**
12:     Set $q \leftarrow \widehat{p}_\mathcal{X}$
13: **if** $\delta$ is not specified **then**
14:     Estimate $\delta$ based on $\boldsymbol{X}_{pool}$          ▷ e.g., following the work of Facco et al. (2017)
15: **if** $\alpha$ is not specified **then**
16:     Set $\alpha \leftarrow \infty$          ▷ as discussed in Sec. 6
17: Set $p_0 \leftarrow q$
18: Draw initial training inputs $\boldsymbol{X}_{n_0} \sim p_0$
19: Query labels $\boldsymbol{Y}_{n_0} \leftarrow \boldsymbol{y}(\boldsymbol{X}_{n_0})$ from the oracle
20: Set $(\Theta_H, \Sigma_E) \leftarrow \mathrm{hyper\_init}(\boldsymbol{X}_{n_0}, \boldsymbol{Y}_{n_0}, p_0, \boldsymbol{X}_{\mathrm{val}}, \boldsymbol{Y}_{\mathrm{val}})$      ▷ see Algorithm 2 in Appendix D
21: ▷ Sample Process
22: **for** $k \in \mathbb{N}_0$ **do**
23:     **if** $k > 0$ **then**
24:        Update IP locations $\boldsymbol{X}_\dagger^E, \boldsymbol{X}_\dagger^G \in \Theta_H$, where $\boldsymbol{X}_\dagger^E, \boldsymbol{X}_\dagger^G \sim \sqrt{p_k \cdot \widehat{\mathfrak{C}}_{\mathrm{GPR}}^{n_{k-1}}}$     ▷ see (29) in Sec. 4.5
25:        **if** $k == 1$ **then**
26:          Gradually decrease $m_E = \left|\boldsymbol{X}_\dagger^E\right|$ and $m_G = \left|\boldsymbol{X}_\dagger^G\right|$ as long as the validation performance of $\widehat{f}_{\mathrm{MoE}}$ does not degrade
     as discussed in Sec. 4.4.3
27:     Train the model $\widehat{f}_{\mathrm{MoE}}$ from Sec. 4.3 with hyperparameters $\Theta_H$ on $(\boldsymbol{X}_{n_k}, \boldsymbol{Y}_{n_k})$ as described in Sec. 4.4
28:     Estimate the LOB $\widehat{\Sigma}_{\mathrm{GPR}}^{n_k}$ of GPR according to (24)
29:     Estimate the LFC $\widehat{\mathfrak{C}}_{\mathrm{GPR}}^{n_k} \leftarrow \left[1/p_k(x)\right]^{\frac{1}{2\alpha+d}} \left|\widehat{\Sigma}_{\mathrm{GPR}}^{n_k}(x)\right|^{-1}$ according to (22)
30:     Estimate the superior training density $\widehat{p}_{\mathrm{Sup}}^{\mathrm{GPR},n_k} \leftarrow \left[\widehat{\mathfrak{C}}_{\mathrm{GPR}}^{n_k}(x)q(x)\right]^{\frac{2\alpha+d}{4\alpha+d}} \widehat{v}(x)^{\frac{2\alpha}{4\alpha+d}}$ according to (23)
31:     Set $\gamma_1 = \max_{x\in\mathcal{X}} p_k(x)\big/\widehat{p}_{\mathrm{Sup}}^{\mathrm{GPR},n_k}(x)$ and $\gamma_2 = \max\left\{0, (0.5 - \gamma_1^{-1})\big/(1-\gamma_1^{-1})\right\}$    ▷ see Sec. 4.2
32:     Set $p_{k+1} \leftarrow \gamma_2 p_k + (1-\gamma_2)\widehat{p}_{\mathrm{Sup}}^{\mathrm{GPR},n_k}$
33:     Set $\widetilde{p}_{k+1} \leftarrow 2p_{k+1} - p_k$          ▷ see Sec. 4.2
34:     Draw $X_{n_k+1}, \ldots, X_{n_{k+1}} \sim \widetilde{p}_{k+1}$ via importance sampling from $\boldsymbol{X}_{pool}$ as described in Sec. 4.2
35:     Query labels $y_{n_k+1}, \ldots, y_{n_{k+1}} \leftarrow \boldsymbol{y}(X_{n_k+1}, \ldots, X_{n_{k+1}})$ from the oracle
36:     Set $\boldsymbol{X}_{n_{k+1}} \leftarrow \boldsymbol{X}_{n_k} \cup \{X_{n_k+1}, \ldots, X_{n_{k+1}}\}$ and $\boldsymbol{Y}_{n_{k+1}} \leftarrow \boldsymbol{Y}_{n_k} \cup \{y_{n_k+1}, \ldots, y_{n_{k+1}}\}$    ▷ Then $\boldsymbol{X}_{n_{k+1}} \sim p_{k+1}$

---

**Definition 2.** *Over the space of square-integrable functions $f \in \mathcal{L}^2(\mathcal{X})$, for a nonparametric regression model $\widehat{f}$ and a training density $p$, we define by $\varrho(\widehat{f}, p) > 0$ the relative required sample size such that for $n' = \varrho(\widehat{f}, p)n$, $\boldsymbol{X}'_{n'} \sim p$ and $\boldsymbol{X}_n \sim q$ it holds that*

$$MISE\left(q, \widehat{f}|\boldsymbol{X}_n\right) = MISE\left(q, \widehat{f}|\boldsymbol{X}'_{n'}\right)(1 + o(1)).$$

Thus, a training density $p$ is asymptotically superior to *random test sampling*, if $\varrho(\widehat{f}, p) < 1$, since we achieve the same performance as *random test sampling* with only a fraction of the number of training samples. In Sec. 5 we will demonstrate the superiority of the training density $p_{\mathrm{Sup}}^{\mathrm{GPR},n}$ for our GPR-based MoE model.

**Respecting the intrinsic dimension in high-dimensional input spaces** In Eq. (19), (20) and (21) we assume the input space $\mathcal{X}$ to have full degrees of freedom $d$, which in practice is particularly not the case

in high-dimensional feature spaces. For an intrinsic dimension $\delta < d$ of $\mathcal{X}$, we adjust as follows: For the space $\mathcal{S} = \{\sigma\Sigma \mid \sigma > 0\}$ of bandwidth candidates that are essentially isotropic up to a fixed, shared positive definite factor $\Sigma \in \mathbb{S}_{++}^d$ that is, e.g., calculated in a pre-processing step, let $\Sigma_{\mathrm{GPR}}^n(x) = \sigma_{\mathrm{GPR}}^n(x)\Sigma$ be the LOB function of GPR with respect to $\mathcal{S}$. Then we replace all occurrences of $d$ for $\delta$ and $|\Sigma_{\mathrm{GPR}}^n(x)|$ for $\sigma_{\mathrm{GPR}}^n(x)^\delta$ in Eq. (19), (20) and (21).

Besides being an ingredient to AL, LFC can also be used to reduce the required model complexity. For example, in an RBF-network or a sparse GPR model, we can coarsen or refine the model resolution by placing an adequate amount of basis functions or IPs, respecting LFC. We will discuss this choice in Sec. 4.5 and demonstrate its ability to reduce the overall model complexity in Sec. 5.1. Finally, LFC can be inspected to obtain deeper insights into the research field of the regression problem, which is particularly hard for high-dimensional data (see Sec. 5.2).

## 4.2 The active learning framework

Starting with an initial training set $\boldsymbol{X}_{n_0}, \boldsymbol{Y}_{n_0}$ of size $n_0$ with $\boldsymbol{X}_{n_0} \sim p_0$ for some initial training distribution such as $p_0 \equiv q$, we implement the online sampling procedure as described in Panknin et al. (2021), such that $\boldsymbol{X}_n \sim p_{\mathrm{Sup}}^{\mathrm{GPR},n}$ as $n \to \infty$. We grow the training set as follows:

Given the current training set $\boldsymbol{X}_{n_k}, \boldsymbol{Y}_{n_k}$ we estimate $\widehat{\Sigma}_{\mathrm{GPR}}^{n_k}$ as described in Sec. 4.3. Using (19), (20), it is

$$\widehat{\mathfrak{C}}_{\mathrm{GPR}}^{n_k}(x) \propto \left[1/p_k(x)\right]^{\frac{1}{2\alpha+d}} \left|\widehat{\Sigma}_{\mathrm{GPR}}^{n_k}(x)\right|^{-1}, \quad \text{and} \tag{22}$$

$$\widehat{p}_{\mathrm{Sup}}^{\mathrm{GPR},n_k}(x) \propto \left[\widehat{\mathfrak{C}}_{\mathrm{GPR}}^{n_k}(x)q(x)\right]^{\frac{2\alpha+d}{4\alpha+d}} \widehat{v}(x)^{\frac{2\alpha}{4\alpha+d}}. \tag{23}$$

Letting the next sample size be $n_{k+1} = 2n_k$, we have already drawn half the samples of $n_{k+1}$ according to a potentially different distribution $p_k$ than the new proposed $\widehat{p}_{\mathrm{Sup}}^{\mathrm{GPR},n_k}$. The closest we can get in distribution to $\widehat{p}_{\mathrm{Sup}}^{\mathrm{GPR},n_k}$ is given by $\boldsymbol{X}_{n_{k+1}} \sim p_{k+1}$, where $p_{k+1} := \gamma_2 p_k + (1-\gamma_2)\widehat{p}_{\mathrm{Sup}}^{\mathrm{GPR},n_k}$, for $\gamma_2 = \max\left\{0, \frac{0.5 - \gamma_1^{-1}}{1 - \gamma_1^{-1}}\right\} \in [0, 0.5)$ and $\gamma_1 = \max_{x \in \mathcal{X}} \dfrac{p_k(x)}{\widehat{p}_{\mathrm{Sup}}^{\mathrm{GPR},n_k}(x)}$. This is achieved by sampling $x_{n_k+1}, \dots, x_{n_{k+1}} \sim \widetilde{p}_{k+1}$ for $\widetilde{p}_{k+1} = 2p_{k+1} - p_k$, which is a valid probability density (Panknin et al., 2021).

**Adaptions in the pool-based active learning scenario** In the AL framework described above, we deal with properly normalized probability densities. But in the *pool-based* AL scenario such normalization is usually impossible since our information about the input space $\mathcal{X}$ is restricted to a large, unlabeled *pool* of samples $\boldsymbol{X}_{pool} \in \mathcal{X}^N$. This pool follows a distribution $\boldsymbol{X}_{pool} \sim p_{\mathcal{X}}$, for which it is common to assume an (unnormalized) density estimate $\widehat{p}_{\mathcal{X}}$ to be given: Unlabeled inputs are considered cheaply accessible, whereas querying labels is expensive.

For our AL framework to be applicable, it suffices to keep all considered densities such as $\widehat{p}_{\mathrm{Sup}}^{\mathrm{GPR},n_k}$ at equal norm, which we can enforce via normalizing a density $p$ by $\bar{p} = p/\mathrm{norm}(p)$, where

$$\mathrm{norm}(p) = \left|\boldsymbol{X}_{pool}\right|^{-1} \sum\nolimits_{x \in \boldsymbol{X}_{pool}} p(x)/\widehat{p}_{\mathcal{X}}(x).$$

To see this, note that first of all $\widehat{p}_{\mathcal{X}}$ is an unnormalized estimate of $p_{\mathcal{X}}$ such that we can write $\widehat{p}_{\mathcal{X}} \approx c \cdot p_{\mathcal{X}}$ for some unknown constant $c > 0$. On the one hand, it is $\int_{\mathcal{X}} \widehat{p}_{\mathcal{X}}(x)dx = c$ by definition. On the other hand, it is

$$\mathrm{norm}(p) \approx \int_{\mathcal{X}} \frac{p(x)}{\widehat{p}_{\mathcal{X}}(x)} p_{\mathcal{X}}(x)dx = \frac{1}{c}\int_{\mathcal{X}} p(x)dx,$$

such that also $\int_{\mathcal{X}} \bar{p}(x)dx = \frac{1}{\mathrm{norm}(p)}\int_{\mathcal{X}} p(x)dx \approx c$ holds for any unnormalized density $p$.

Subsequently, the required samples $x_{n_k+1}, \dots, x_{n_{k+1}} \sim \widetilde{p}_{k+1}$ are obtained via *importance sampling* from the pool with *importance weights* $\mathbb{P}(x_i = x) \propto \widetilde{p}_{k+1}(x)/[\mathrm{norm}(\widetilde{p}_{k+1})\widehat{p}_{\mathcal{X}}(x)]$ for $x \in \boldsymbol{X}_{pool}$ and $n_k + 1 \leq i \leq n_{k+1}$.

### 4.3 Sparse mixture of Gaussian processes

Recall the sparse MoE model (15) from Sec. 3.4.2, given by

$$\widehat{f}_{\mathrm{MoE}}(x) = \sum_{l=1}^{L} G(x)_l \widehat{y}_l(x),$$

where the gate $G \colon \mathcal{X} \to [0,1]^L$ is a probability assignment of an input $x$ to the expert models $\widehat{y}_l$. In particular, it holds $\sum_{l=1}^{L} G(x)_l \equiv 1$ and $G(x)_i \geq 0, \forall x \in \mathcal{X}$ and $1 \leq i \leq L$. According to (18), besides the expert and gate model parameters $\{\theta_{e_l}\}_{l=1}^{L}$ and $\{\theta_{g_l}\}_{l=1}^{L}$, this MoE approach has two hyperparameters, $\kappa$ and $\mathfrak{s}$, for controlling the sparsity of the gate and adding noise to the gate responses during the training to escape local optima.

We choose the expert models as well as the single channel gating models to be sparse variational GPs (Williams & Rasmussen, 1996; Hensman et al., 2015), that is, $\widehat{y}_l \sim \mathcal{SVGP}(\theta_{e_l})$ and $g_l \sim \mathcal{SVGP}(\theta_{g_l})$, which are parameterized by $\theta_{e_l}$ and $\theta_{g_l}$, as described in Sec. 3.4.1. The overall MoE hyperparameter set is thus given by $\Theta = (\{\theta_{e_l}\}_{l=1}^{L}, \{\theta_{g_l}\}_{l=1}^{L}, \kappa, \mathfrak{s})$.

We will keep certain hyperparameters of $\Theta$ constant after initialization, and share some hyperparameters across experts and the channels of the gate: While the covariances of the inducing value distributions $\boldsymbol{S}_{\dagger} \in \theta, \theta \in \Theta$ could be full positive definite matrices, we apply $\boldsymbol{S}_{\dagger} = 0$ throughout, giving favorable stability and computational efficiency. For the same reasons, we fix the *inducing point* (IP) locations $\boldsymbol{X}_{\dagger} \in \theta, \theta \in \Theta$ after initialization. Furthermore, we share the IP locations among the experts, respectively the gate channels, such that for $\boldsymbol{X}_{\dagger} \in \theta_{e_l}$ we apply $\boldsymbol{X}_{\dagger} = \boldsymbol{X}_{\dagger}^E$ and for $\boldsymbol{X}_{\dagger} \in \theta_{g_l}$ we apply $\boldsymbol{X}_{\dagger} = \boldsymbol{X}_{\dagger}^G$, for all $1 \leq l \leq L$.

In this work, our goal is to fit a single, coherent regression problem by a MoE approach. Therefore, we propose to share all the parameters across the experts that characterize the regression function rather than the expert model. That is, we share the mean $\mu_E$, the regularization parameter $\lambda_E$, and the noise variance function $\widehat{v}$, respectively the global noise variance $\sigma_{\varepsilon}^2$ with $\widehat{v}(x) \equiv \sigma_{\varepsilon}^2$ in case of homoscedasticity. Furthermore, we apply a fixed, logarithmically spaced set of individual expert bandwidth scaling factors $\sigma_1 < \ldots < \sigma_L$ that are multiplied by a fixed, shared bandwidth matrix $\Sigma_E$. Our expert parameters therefore reduce to

$$\theta_{e_l} = (\mu_E, \lambda_E, \widehat{v}, \sigma_l \Sigma_E, \boldsymbol{X}_{\dagger}^E, \boldsymbol{\mu}_{\dagger}^{e_l}, 0).$$

**Remark 3.** *Recall from Sec. 2 that it is possible to replace the variational GPR expert models for full as well as sparse analytic GPR formulations (see Appendix B). With slight abuse of notation, these cases are subsumed by setting $\boldsymbol{\mu}_{\dagger}^{e_l} = \emptyset$ or $\boldsymbol{X}_{\dagger}^E = \boldsymbol{\mu}_{\dagger}^{e_l} = \emptyset$ for sparse, respectively full analytic GPR.*

Since our objective does not incorporate any likelihood about the gate's output, there is no noise function to fit for the gate, such that we set $\widehat{v} \equiv 0$ for $\widehat{v} \in \theta_{g_l}$ and all $1 \leq l \leq L$. Each output channel of the gate poses its own classification problem, which is why we do not share the means. Yet, we share the regularization parameter and the bandwidth, as the individual channels should be structurally similar. Our gate parameters therefore reduce to

$$\theta_{g_l} = (\mu_{g_l}, \lambda_G, 0, \sigma_G \mathcal{I}_d, \boldsymbol{X}_{\dagger}^G, \boldsymbol{\mu}_{\dagger}^{g_l}, 0).$$

After training as described in Sec. 4.4, this MoE can cope with a varying structural complexity through the individual bandwidth scaling factors $\sigma_l$ of the experts and heteroscedastic noise through the adaptive regularization. Additionally, we can now use the gate of our MoE to propose an LOB estimate of GPR.

**A GPR-based LOB estimate**   After training of the MoE, we use the learned gate $G$ from (15) to predict $\Sigma_{\mathrm{GPR}}^n(x)$ as

$$\widehat{\Sigma}_{\mathrm{GPR}}^n(x) = \widehat{\sigma}_{\mathrm{GPR}}^n(x) \Sigma_E, \quad \text{where} \quad \widehat{\sigma}_{\mathrm{GPR}}^n(x) = \exp \left\{ \sum_{l=1}^{L} G(x)_l \log(\sigma_l) \right\}. \qquad (24)$$

Due to the finite candidate set $\sigma_1, \ldots, \sigma_L$ we are limited to measure a quantization of $\Sigma_{\mathrm{GPR}}^n(x)$ through $G(x)_l = \mathbb{P}(\Sigma_{\mathrm{GPR}}^n(x) = \sigma_l \Sigma_E)$. If, in fact, $\Sigma_{\mathrm{GPR}}^n \in \{\sigma_1 \Sigma_E, \ldots, \sigma_L \Sigma_E\}$ holds true, then there exists an index function $j(x) \in \{1, \ldots, L\}$ such that $\Sigma_{\mathrm{GPR}}^n(x) = \sigma_{j(x)} \Sigma_E$. In this case, we are able to exactly recover LOB

with $G(x)_l = \begin{cases} 1, & l = j(x) \\ 0, & \text{else} \end{cases}$. In any other case, the estimate (24) of LOB is a reasonable interpolation, which deviation from $\Sigma_{\text{GPR}}^n$ can be controlled by the number of bandwidth candidates of the MoE.

## 4.4 Model training

This section is devoted to the training of the model described in Sec. 4.3. We first set up the training objective in Sec. 4.4.1 and describe the training procedure of our model in Sec. 4.4.2, where we identify hyperparameters of the approach. Then, we discuss how to choose the essential hyperparameters systematically on the initial training dataset in Sec. 4.4.3.

### 4.4.1 The training objective

First, we will set up the objective function for training our MoE model in *batch mode.*

**The main objective**   Denote by $\varnothing \subsetneq \mathcal{B} \subseteq \{1, \ldots, n\}$ the indices of a batch, and let $w_{\mathcal{B}} = \sum_{b \in \mathcal{B}} w(x_b)$ for the training importance weight function $w \propto q/p$. Let $P_l$ be the prior distribution of the inducing function values of the l-th expert and $Q_l$ the corresponding variational distribution as defined in Sec. 3.4.1. We choose the (through the gate G) weighted sum of the individual expert negative ELBO objectives (13), denoted by

$$\text{Obj}(\boldsymbol{X}_n, \boldsymbol{Y}_n, \mathcal{B}, w, \Theta) = -\sum_{l=1}^{L} \left[ w_{\mathcal{B}}^{-1} \sum_{b \in \mathcal{B}} v_l(x_b) P_{b,l} - \frac{1}{n_l} \mathcal{KL} \left[ Q_l(\cdot | \boldsymbol{X}_n, \boldsymbol{Y}_n) \| P_l \right] \right],$$

as our main objective, where $n_l = n w_{\mathcal{B}} / \nu_{\mathcal{B},l}$ for $\nu_{\mathcal{B},l} = \sum_{b \in \mathcal{B}} v_l(x_b)$ with $v_l(x) = G(x)_l w(x)$ and

$$P_{b,l} = \mathbb{E}_{u \sim Q_l(\cdot | \boldsymbol{X}_n, \boldsymbol{Y}_n)} \log \int P_l(y_b|f) P_l(f|u, x_b) df$$

is the predictive log-likelihood (14) of the l-th expert in $x_b$, marginalized over its variational distribution $Q_l$.

**A penalty on small bandwidth choices**   In the spirit of Lepski's method (Lepski, 1991; Lepski & Spokoiny, 1997), we prefer the largest choice of bandwidth out of all candidates that perform comparably well. In order to enforce this, we penalize smaller bandwidth choices by adding the following term:

$$\text{pen}_\sigma(\boldsymbol{X}_n, \boldsymbol{Y}_n, \mathcal{B}, w, \Theta) = \frac{2}{(L-1)} \sum_{l=1}^{L} \nu_{\mathcal{B},l}(L-l) \Big/ \sum_{l=1}^{L} \nu_{\mathcal{B},l}. \tag{25}$$

Note that $\text{pen}_\sigma(\boldsymbol{X}_n, \boldsymbol{Y}_n, \mathcal{B}, w, \Theta) = 1$ if $\nu_{\mathcal{B},1} = \ldots = \nu_{\mathcal{B},L}$. Our total objective then amounts to

$$\text{Obj}(\boldsymbol{X}_n, \boldsymbol{Y}_n, \mathcal{B}, w, \Theta) = \text{Obj}(\boldsymbol{X}_n, \boldsymbol{Y}_n, \mathcal{B}, w, \Theta) + \vartheta_\sigma \text{pen}_\sigma(\boldsymbol{X}_n, \boldsymbol{Y}_n, \mathcal{B}, w, \Theta). \tag{26}$$

**Remark 4.** *If we assume our problem to be (almost) noise-free, we replace the Obj in our objective* (26) *for the* mean squared error

$$MSE(\boldsymbol{X}_n, \boldsymbol{Y}_n, \mathcal{B}, w, \Theta) = w_{\mathcal{B}}^{-1} \sum_{b \in \mathcal{B}} w(x_b) \|y_b - \widehat{y}(x_b)\|^2.$$

### 4.4.2 Training procedure

We implement our model in *PyTorch* (Paszke et al., 2019), using the *GPyTorch*-package (Gardner et al., 2018). Given the training set $(\boldsymbol{X}_n, \boldsymbol{Y}_n)$, we minimize the objective described in Sec. 4.4.1 via ADAM-optimization (Kingma & Ba, 2015). It remains to identify those variables of the MoE that will be adapted as parameters during the training. Then, the remaining variables are hyperparameters that need to be specified or tuned through an external validation step.

Recall from Sec. 4.3 that the MoE has two further hyperparameters, $\kappa$ for enforcing sparse gate responses and a noise term on the gate responses during the training, which is controlled by $\mathfrak{s}$. Instead of learning $\mathfrak{s}$ as a parameter during the training—like proposed by Shazeer et al. (2017)—we propose to shrink $\mathfrak{s} \leftarrow \mathfrak{s} \eta_{\mathfrak{s}}$ after

each training epoch, for a multiplicative factor $\eta_\mathfrak{s} < 1$ and an initial value $\mathfrak{s} := \mathfrak{s}_0$ as hyperparameters. We discuss this heuristic choice in Appendix F.

We require appropriate learning rates for the optimization of the parameters and tunable hyperparameters of the model. Generally, we suggest applying an adaptive base learning rate $\eta$, where we shrink $\eta \leftarrow \eta_i := \frac{1}{2}\eta_{i-1}$ for an initial base learning rate $\eta_0$ during the training as soon as the validation performance gets stuck until $\eta_i$ crosses a lower threshold, e.g., $\eta_i < \eta_0/1000$. Note that, relative to the base learning rate, good learning rates for the individual types of tunable parameters should be deployed: Within a GP component $\mathcal{SVGP}(\theta)$ with $\theta = (\mu, \lambda, \widehat{v}, \Sigma, \boldsymbol{X}_\dagger, \boldsymbol{\mu}_\dagger, \boldsymbol{S}_\dagger)$, the hyperparameters $(\mu, \lambda, \widehat{v})$ must be updated on a smaller scale than the inducing value distribution, given by $(\boldsymbol{\mu}_\dagger, \boldsymbol{S}_\dagger)$. In this regard, let $\eta_H \leq 1$ be the factor such that, if we update $\boldsymbol{\mu}_\dagger$ at rate $\eta$, then we update $(\mu, \lambda, \widehat{v})$ at rate $\eta_H \eta$.

Similarly, we need to update the gate parameters $\theta_{g_l}$ on a smaller scale than the expert parameters $\theta_{e_l}$. In this regard, let $\eta_G \leq 1$ be the factor such that, if we update $\theta_{e_l}$ at rate $\eta$, then we update $\theta_{g_l}$ at rate $\eta_G \eta$.

The set of hyperparameters that require off-training selection (e.g., via cross-validation) is thus given by

$$\Theta_H = (B, \kappa, \{\sigma_l\}_{l=1}^L, \sigma_G, \lambda_G, \boldsymbol{X}_\dagger^E, \boldsymbol{X}_\dagger^G, \mathfrak{s}_0, \eta_\mathfrak{s}, \vartheta_\sigma, \eta_0, \eta_H, \eta_G), \tag{27}$$

whereas the overall set of parameters that get tuned while training is given by

$$\Theta_T = (\mu_E, \lambda_E, \widehat{v}, \Sigma_E, \boldsymbol{\mu}_\dagger^E, \mu_G, \boldsymbol{\mu}_\dagger^G). \tag{28}$$

We provide further details on the design choices for our MoE model in Appendix F.

### 4.4.3 Choosing the hyperparameters

Since our MoE approach is based on known building blocks (Williams & Rasmussen, 1996; Hensman et al., 2015; Shazeer et al., 2017) we can train our model using well-established software libraries (Kingma & Ba, 2015; Paszke et al., 2019; Gardner et al., 2018), with the hyperparameters chosen by following best practice. While the set of hyperparameters (27) appears to be large, most of them can be tuned in advance on the initial training dataset of moderate size and held fixed in the subsequent training data refinement process.

Note that some hyperparameters impact the computational complexity rather than the model performance. Thus, as long as they are not underestimated, their tuning is optional and will therefore be postponed:

- Since our MoE is robust concerning unnecessarily large choices of the gate output sparsity $\kappa$, we initialize $\kappa \equiv L$ while choosing the remaining hyperparameters, followed by tuning $\kappa$ as the last hyperparameter, where we successively reduce $\kappa$ until we observe a significant loss of performance of the MoE.

- The numbers $m_E = \left| \boldsymbol{X}_\dagger^E \right|$, $m_G = \left| \boldsymbol{X}_\dagger^G \right|$ of IPs of the expert and the gate are the main driver of the computational complexity of our MoE. While unnecessarily large numbers will not hurt the model performance, they should therefore be set to the smallest value that leads to no significant loss of performance to keep the computational complexity of the model moderate at larger training sizes. In the initial iteration, we use $m_E = n_0$ for the experts and $m_G = \frac{n_0}{4}$, where the locations of the IPs $\boldsymbol{X}_\dagger^E, \boldsymbol{X}_\dagger^G \sim p$ are chosen *diverse* as described in Appendix E. In the second iteration, where we have first estimates of LFC, $\boldsymbol{X}_\dagger^E$ and $\boldsymbol{X}_\dagger^G$ are drawn as discussed in Sec. 4.5. Here, we gradually decrease $m_E$ and $m_G$ until we observe a significant loss of validation performance of the MoE. We hold $m_E$ and $m_G$ fixed in subsequent iterations. Note that the IP locations are not subject to optimization.

The initial base learning rate and the expert's internal hyperparameters learning rate $(\eta_0, \eta_H)$ and the batch size $B$ that are related to a single GPR expert rather than the whole MoE model:

- First, we hold $\eta_H = 0.2$, $B = n_0$ fixed at reasonable initial values and perform line search over $\eta_0$ according to the resulting validation performance of a single, isotropic, sparse GPR expert. Here, too small values of $\eta_0$ result in slow convergence of the objective, in which case we interrupt the training immediately and increase $\eta_0$ as long as the first objective updates are consistently decreasing.

- Next, we choose $\eta_H$ according to the resulting validation performance of a single, isotropic, sparse GPR expert, where we gradually decrease $\eta_H$, starting from $\eta_H = 1$. Again, we interrupt the training for too large choices of $\eta_H$, where the training objective will diverge.

- Finally, we choose $B$ according to the resulting validation performance of a single, isotropic, sparse GPR expert, where we gradually decrease $B$, starting from $B = n_0$.

Next, we have to deal with the remaining hyperparameters that are related to the MoE model. For this, we first initialize the less crucial hyperparameters at reasonable values, tuning them afterward: We apply a set of experts with $\{\sigma_l\}_{l=1}^L$, where $L = 7$, and $\sigma_l = 2^{\frac{l-4}{\delta}}$, which are logarithmically spaced around the global bandwidth estimate $\Sigma_E$ of the best-performing model that we obtained from the above tuning of the hyperparameters related to a single GPR expert. The noise added to the gate $(\mathfrak{s}_0, \eta_\mathfrak{s})$ as well as the regularization $\vartheta_\sigma$ of the bandwidth function are about fine-tuning of the model. We set them to $\mathfrak{s}_0 = 0.1, \eta_\mathfrak{s} = 1/\sqrt{2}$ and $\vartheta_\sigma = 0.01$. We suggest to keep $\eta_\mathfrak{s} = 1/\sqrt{2}$ throughout without further tuning.

- Via grid search, we choose $\sigma_G, \lambda_G$ according to the resulting validation performance of the MoE model, where we gradually decrease the gate learning rate from $\eta_G = 1$.

As a last step, we choose the hyperparameters for fine-tuning of the MoE model:

- First, we perform line search over $\vartheta_\sigma$ according to the resulting validation performance of the MoE.

- Next, we tune $\{\sigma_l\}_{l=1}^L$: We observe that unreasonable bandwidths will be automatically dropped during the training. Therefore, if the minimal or maximal candidate associated with $\sigma_1, \sigma_L$ is not chosen during the training, we remove the respective expert and retrain the MoE. Vice versa, we expand the bandwidth candidate range beyond $\sigma_1, \sigma_L$ with a factor of $2^{\mp\frac{1}{\delta}}$ as long as the boundary candidates are not dropped during the training.

- Finally, we perform line search over $\mathfrak{s}_0$ according to the resulting validation performance of the MoE.

## 4.5 Initializing the IP locations

Since the numbers of IPs of the gate $\boldsymbol{X}_\dagger^G$, as well as the experts $\boldsymbol{X}_\dagger^E$ are the computational bottleneck of our model, they should be chosen advisedly. We can interpret the choice of IPs as a nested AL task at small sample size. In the small sample size regime, input space geometric arguments have proven to be robust and superior in comparison to naive approaches like random sub-sampling from the training inputs (Teytaud et al., 2007; Yu & Kim, 2010; Wu, 2019; Liu et al., 2021). They are representative, respecting the training distribution, and diverse (with *low-dispersion*) so that they achieve an acceptable representation of the dataset at the smallest possible number of IPs. By low-dispersion, we resort to the following definition:

**Definition 5.** *The* dispersion*, given by* $\sup_{x \in \mathcal{X}} \min_{1 \leq i \leq n} \|x - x_i\|$ *(Niederreiter, 1988) is a measure of how well spread out the training sample is. We say that a sequence has low-dispersion if its dispersion is lower than the dispersion of random uniform sampling.*

Indeed, by sampling the IPs in this manner, we can reduce the distance of an evaluation point $x$ to its closest neighbor in $\boldsymbol{X}_\dagger^E$, which is known to reduce the reconstruction error of a full kernel matrix by a sparse representation (Zhang et al., 2008).

In addition, recall that our derived LFC measure of local structural complexity quantifies the local variation of the target function. Intuitively, we require more IPs to sense and reconstruct the target function where this local variation is higher. In summary, we therefore propose to choose the IPs

$$\boldsymbol{X}_\dagger^E, \boldsymbol{X}_\dagger^G \sim \sqrt{p \cdot \mathfrak{C}_{\text{GPR}}^n} \tag{29}$$

as the *geometric mean* of LFC and the training density $\boldsymbol{X}_n \sim p$ in a diverse way. Here, we ensure diversity by implementing distribution preserving clustering or particle repulsion as described in Appendix E.

## 5 Experiments

In this section, we will first analyze our approach on toy-data, regarding the MoE model, LFC, and the *superior training density*. Then, we apply our approach to a high-dimensional MD simulation dataset from quantum chemistry, by which we can deduce deeper insights into this regression problem.

We denote the *root mean squared error* (RMSE) and the *maximum absolute error* (max AE) of a model $\widehat{f}$ for a test set $\boldsymbol{X}_{\mathrm{T}} \in \mathcal{X}^N$ with $\boldsymbol{X}_{\mathrm{T}} \sim q$ by

$$\mathrm{RMSE}(\widehat{f}, \boldsymbol{X}_n, \boldsymbol{Y}_n) = \left[ \frac{1}{N} \sum_{x \in \boldsymbol{X}_{\mathrm{T}}} \left| f(x) - \widehat{f}_{\boldsymbol{X}_n, \boldsymbol{Y}_n}(x) \right|^2 \right]^{\frac{1}{2}} \quad \text{and} \quad \max \mathrm{AE}(\widehat{f}, \boldsymbol{X}_n, \boldsymbol{Y}_n) = \max_{x \in \boldsymbol{X}_{\mathrm{T}}} \left| f(x) - \widehat{f}_{\boldsymbol{X}_n, \boldsymbol{Y}_n}(x) \right|.$$

As already discussed in Sec. 4.1, the learning rate is invariant under change of the training density $\boldsymbol{X}_n \sim p$ in the considered scenario. For our MoE model and $f \in \mathcal{C}^\alpha(\mathcal{X}, \mathbb{R})$ we, thus, can write

$$\mathrm{RMSE}(\widehat{f}_{\mathrm{MoE}}, \boldsymbol{X}_n, \boldsymbol{Y}_n) = C_p n^{-\tau}(1 + o(1)) \;, \text{ with } \quad \tau = \begin{cases} \frac{\alpha}{2\alpha + d}, & \alpha < \infty \\ 1/2, & \alpha = \infty \end{cases}, \tag{30}$$

where $C_p > 0$ is a constant depending on the training density $p$. Note that we can theoretically bound the asymptotic RMSE from below by $C^* n^{-\tau}$, where we have defined $C^* := C_{p^*}$ with $\boldsymbol{X}'_n \sim p^*$ being the optimal training set from (3). Unfortunately, since $p^*$ is unknown—even when given the ground truth—we are not able to estimate $C^*$ and, thus, provide a lower bound of the RMSE beyond the known learning rate $n^{-\tau}$.

As an AL performance measure, we use the relative required sample size from Definition 2 which can be estimated for a GPR-based model such as our MoE and $f \in \mathcal{C}^\alpha(\mathcal{X}, \mathbb{R})$ according to

$$\varrho(\widehat{f}_{\mathrm{MoE}}, p) \approx \left[ \frac{\mathrm{RMSE}(\widehat{f}_{\mathrm{MoE}}, \boldsymbol{X}'_n, \boldsymbol{Y}'_n)}{\mathrm{RMSE}(\widehat{f}_{\mathrm{MoE}}, \boldsymbol{X}_n, \boldsymbol{Y}_n)} \right]^{\frac{1}{\tau}} \tag{31}$$

where it is $\boldsymbol{X}'_n \sim p$ and $\boldsymbol{X}_n \sim q$ with respective labels $\boldsymbol{Y}'_n$ and $\boldsymbol{Y}_n$. Using (31), we can compare the asymptotic AL performance of different AL sampling schemes in the following experiments. For example, we can quantify the AL performance of our proposed AL framework by sampling $\boldsymbol{X}'_n \sim \widehat{p}_{\mathrm{Sup}}^{\mathrm{GPR}, n}$.

### 5.1 Doppler function

We will first demonstrate our approach on the *Doppler* function (see, for example, Donoho & Johnstone (1994)), which was also discussed in related work that deals with inhomogeneous complexity (Panknin et al., 2021; Bull et al., 2013). For $x \in \mathcal{X} = [0, 1]$, let

$$\mathbb{P}(y|x) = \mathcal{N}(y; f(x), 1), \quad f(x) = C\sqrt{x(1-x)} \sin\left(2\pi(1+\epsilon)/(x+\epsilon)\right),$$

where $\epsilon = 0.05$, $C$ is chosen such that $\|f\|_2 = 7$ and $\mathcal{N}(\cdot; \mu, \sigma^2)$ denotes the Gaussian distribution with mean $\mu$ and variance $\sigma^2$. We assume a uniform test distribution $q \sim \mathcal{U}(\mathcal{X})$ in all Doppler function experiments.

This one-dimensional, homoscedastic toy-example allows for an easy and intuitive visualization. Fig. 3 shows an example dataset as blue dots and the true function $f$ to infer in black. Due to the strong variation of structural complexity, a single-scale GPR model does not cope well with the Doppler function (see Appendix G.1 for a comparison of single-scale to multi-scale GPR).

We implement our proposed MoE model as described in Sec. 4.3 with sparse GPs as the expert and gate models and using the Gaussian kernel $k$. We apply 512, respectively 128 IPs for the experts and the gate, which are chosen via SVGD (see Appendix E). Furthermore we apply $\sigma_j = 10^{(j-10)/3}, 1 \leq j \leq 7$, as the expert bandwidths, $\lambda_E = 20$ as the initial expert regularization, and $\sigma_G = 0.05$ and $\lambda_G = 10$ for the gate. For the training, we apply a batch size of $B = 512$, a terminal expert sparsity $\kappa = 2$, a penalty factor of $\vartheta_\sigma = 0.5$ for small bandwidth choices, gate noise parameters $\mathfrak{s}_0 = 0.1$ and $\eta_{\mathfrak{s}} = 1/\sqrt{2}$, and learning rate parameters $\eta = 0.01$, $\eta_H = 0.2$, $\eta_G = 1$.

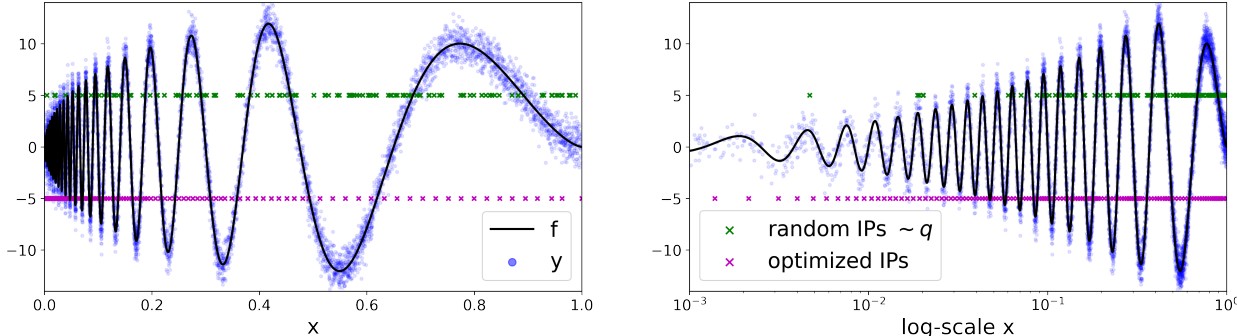

Figure 3: The Doppler experiment: An exemplary dataset and the locations of 128 IPs, once sampled most naively—that is, random according to the test distribution—and once optimized regarding diversity as well as structural complexity and representativeness as described in Sec. 4.5, shown on natural x-scale (left) and on logarithmic x-scale (right).

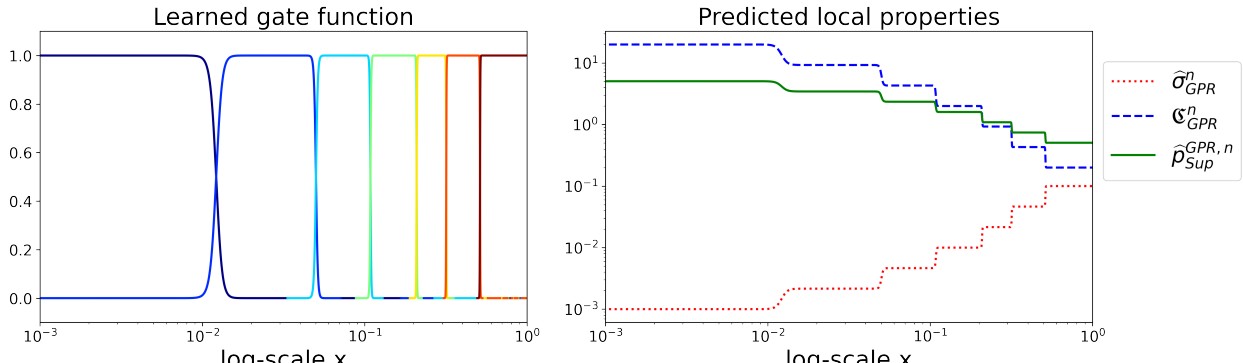

Figure 4: The Doppler experiment: The gate function (left) and the associated estimates of LOB, LFC and *superior training density* (right) trained on the dataset from Fig. 3 and shown on a logarithmic x-scale.

Fig. 4 shows the gate function after training of the MoE model, as described in Sec. 4.4.1, and the associated estimates of LOB, LFC and the *superior training density*, calculated according to (24) and (21).

**Comparing the active learning framework in the LPS and GPR domain** Since $f \in \mathcal{C}^\infty(\mathcal{X}, \mathbb{R})$, our deduced *superior training density* estimate is given by Eq. (21). In Fig. 5 we plot our estimates of LOB and the *superior training density* in comparison to the LPS-based results for polynomial degrees of order $Q = 1, 3$, and with implementation and hyperparameters as described in Panknin et al. (2021). Here, we can observe the qualitative similarity of the LPS- and GPR-based estimates of LOB.

When conducting the proposed GPR-based active sampling scheme as described in Sec. 4.2, we additionally observe quantitative benefits in Fig. 6 over *random test sampling*—quite similar to the LPS-based result for $Q = 3$: When estimating the relative sample size (31) we require to achieve the same RMSE via active sampling compared to *random test sampling*, we obtain $\varrho(\widehat{f}_{\text{MoE}}, \widehat{p}_{\text{Sup}}^{\text{GPR},n}) = 0.58 \pm 0.04$. This means that we save about 42% of samples via our active sampling scheme.

This provides evidence for the effectiveness of our *superior sampling scheme*, combining the theoretical foundation of the LPS domain with the efficient access to LOB estimates in the GPR domain.

**Comparing *random test sampling* to *equidistant sampling*** In the introduction, we indicated that the advantage of the robust and model-agnostic input space geometric arguments (Teytaud et al., 2007; Yu & Kim, 2010; Wu, 2019; Liu et al., 2021) diminishes as the training size grows. We can substantiate this claim by comparing *random test sampling* to *equidistant sampling* on the Doppler dataset. By *equidistant sampling* over $\mathcal{X} = [0, 1]$ we mean the deterministic construction of the training inputs, where for $n$ given training samples the subsequent $n$ training inputs get placed halfway between all nearest neighbors of the former $n$

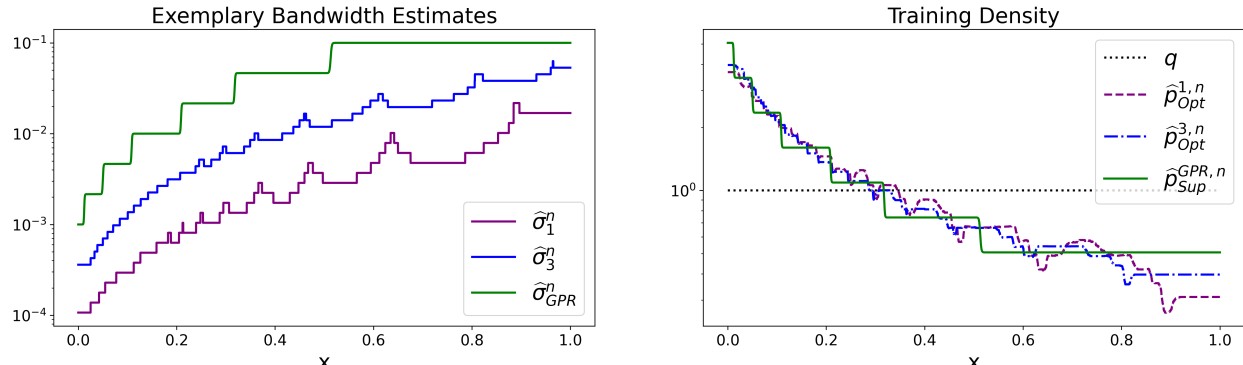

Figure 5: The Doppler experiment: The LOB estimates (left) and the resulting *superior training density* of our proposed GPR-based approach in comparison to the LPS-based approach of order $Q = 1, 3$ (right). The results are averaged over 20 repetitions.

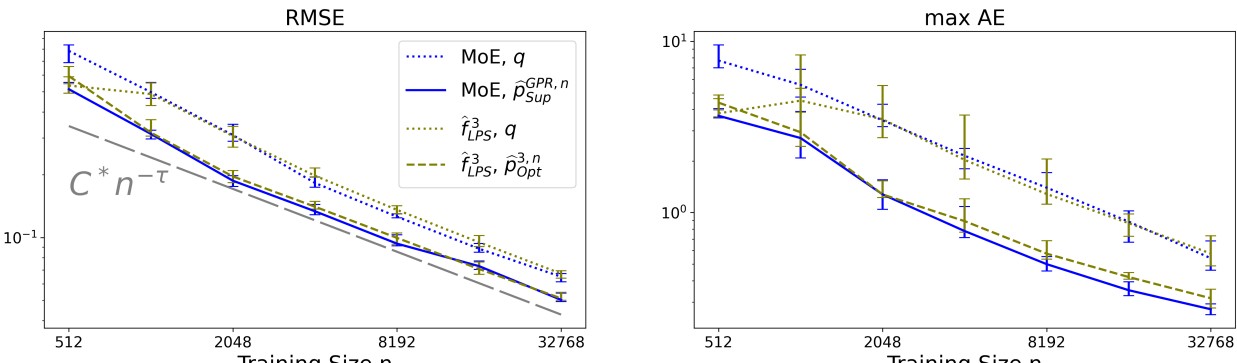

Figure 6: The Doppler experiment: The RMSE (left) and the max AE (right) of our proposed GPR-based approach in comparison to the LPS-based approach of order $Q = 3$ (see Eq. (5) and (6)), once using the respective AL scheme and once, applying *random test sampling*. The results are averaged over 20 repetitions. The long-dashed, gray line is for illustration of the optimal learning rate $\tau$ from (30), where the offset $C^*$ is imaginary. It shall therefore not be confused with a true lower bound.

samples. In this way, the training input inter-distances are halved exactly with each iteration. We regard this construction as the optimal input space geometric choice, which result will subsume all AL competitors of this type. We now observe in Fig. 7 that, indeed, *equidistant sampling* is superior to *random test sampling* at small training sizes. As claimed, however, with growing training size, this advantage gradually diminishes until it has vanished completely at $n = 2^{15}$ training samples.

**On Gaussian process uncertainty** In Sec. 2 we mentioned that GP uncertainty sampling is the most related approach to our *superior sampling scheme* since both build on GPR models. As also discussed therein, we can regard standard GP uncertainty sampling as an input space geometric argument, whose performance we can subsume by equidistant sampling in the Doppler experiment. Particularly this implies that standard GP uncertainty sampling provides no benefits regarding asymptotic AL performance.

Instead—given the gate function of our MoE from the previous part of this experiment, which was obtained for $2^{15}$ training samples and which we now keep fixed—we define the uncertainty estimate of our model MoGPU as a straightforward extension of GP uncertainty sampling which takes the inhomogeneous complexity of data into account: By simply weighting the predictive variances of all experts in some input $x$ with respect to the gate values $G(x)$ from (15), we derive

$$\text{MoGPU}(x) = \sum_{l=1}^{L} G(x)_l \boldsymbol{C}_{\theta_l}^*(x), \tag{32}$$

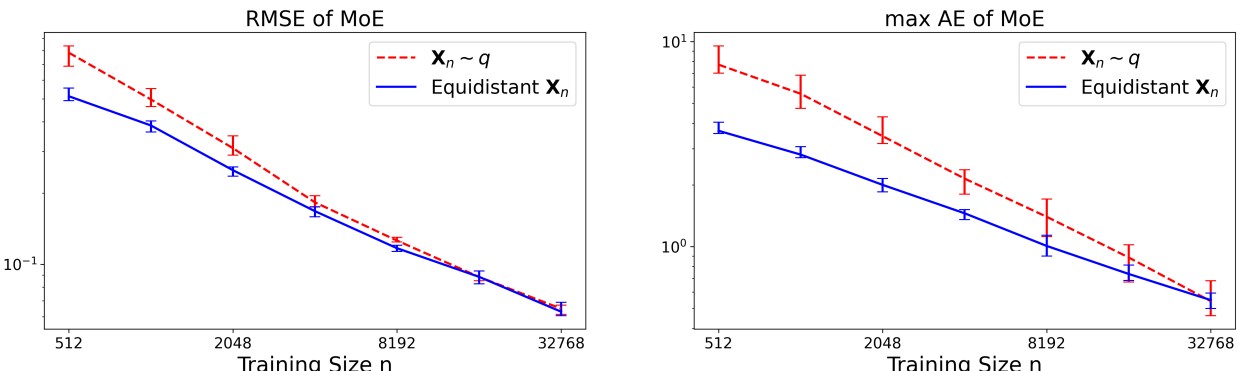

Figure 7: The Doppler experiment: The RMSE (left) and the max AE (right) of our proposed MoE model when comparing *random test sampling* to *equidistant sampling*. The results are averaged over 20 repetitions.

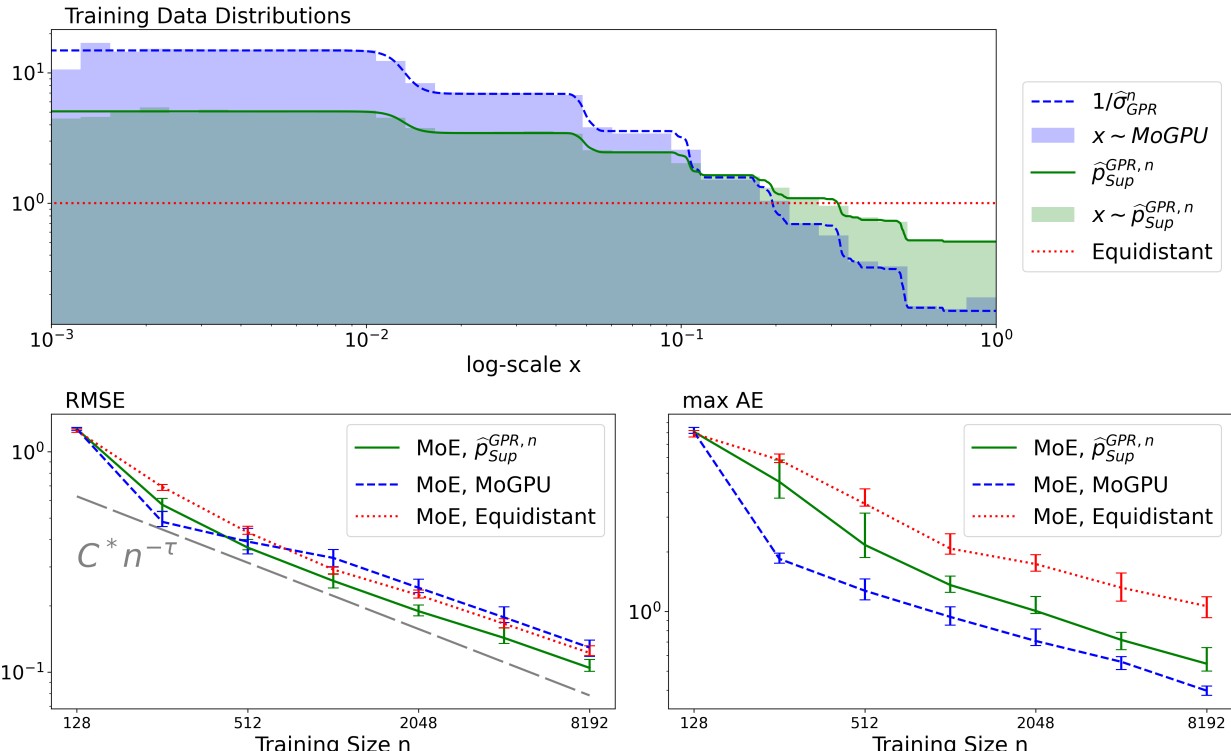

Figure 8: A comparison of the mixture of Gaussian process uncertainty and the *equidistant sampling* baselines to our proposed active sampling scheme for the Doppler experiment: (Top) The training data histograms after $2^{13}$ samples, contrasted with functions of $\sigma_{\mathrm{GPR}}^n$, and the RMSE (bottom left) and the max AE (bottom right) at several training sizes of the compared schemes. The results are averaged over 20 repetitions. The long-dashed, gray line is for illustration of the optimal learning rate $\tau$ from (30), where the offset $C^*$ is imaginary. It shall therefore not be confused with a true lower bound.

where $\boldsymbol{C}_{\theta_l}^*$ is the predictive variance of the l-th expert (see (12)). Note that we consider MoGPU as a baseline competitor to our *superior sampling scheme*.

Intuitively, the uncertainty estimate in $x \in \mathcal{X}$ increases as the applied bandwidth $\sigma_{\mathrm{GPR}}^n(x)$ decreases. Now, in order to equalize uncertainty over the input space, MoGPU will sample more in regions where $\sigma_{\mathrm{GPR}}^n$ is smaller. For $\boldsymbol{X}_n$ drawn according to MoGPU, we expect $\boldsymbol{X}_n \sim [\sigma_{\mathrm{GPR}}^n]^{-d}$. This expectation holds as can be seen at the top in Fig. 8.

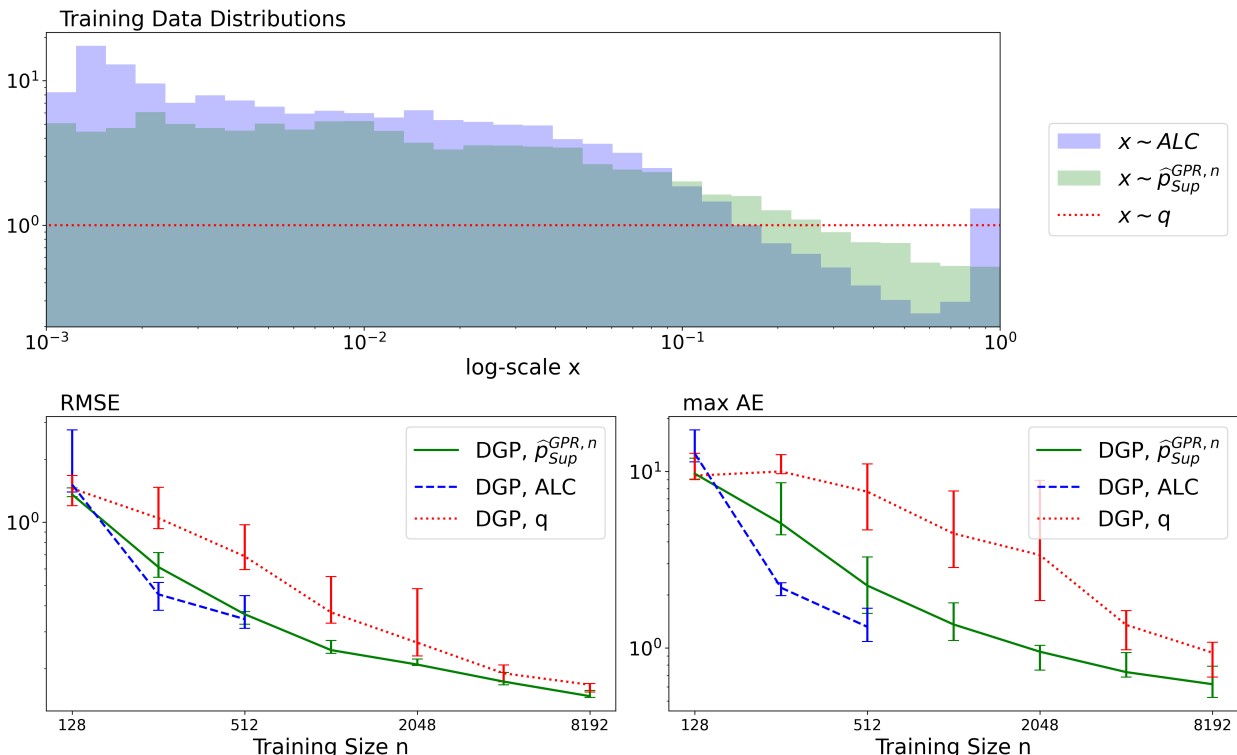

Figure 9: A comparison of the DGP model performance, using the ALC criterion for DGP, the *random test sampling* baseline, and our proposed active sampling scheme for the Doppler experiment: (Top) The training data histograms after 512 samples, and the RMSE (bottom left) and the max AE (bottom right) at several training sizes of the compared schemes. The results are averaged over 5 repetitions.

For evaluation, we combine the fixed gate function with full GPR experts and compare our proposed sampling scheme with MoGPU (both determined through the gate). In all error measures the beneficial effect of the low-dispersion property of $\boldsymbol{X}_n$ drawn according to MoGPU has already vanished for about 512 training samples, from where the asymptotic law dominates. As expected, our approach is superior to MoGPU when comparing RMSE. Interestingly, MoGPU is superior to our approach regarding the max AE, suggesting that $\boldsymbol{X}_n \sim [\sigma^n_{\mathrm{GPR}}]^{-d}$ is the preferable training distribution under the supremum-norm.

**AL Performance on deep Gaussian processes**   To demonstrate the model-agnosticity of our AL approach, we deploy the DGP model of Sauer et al. (2023b) using the CRAN package *deepgp*[4]. This package also implements an aggregate variance-based AL criterion (Cohn, 1994), which they named *active learning Cohn* (ALC) after the originator. We deploy a 3-layer DGP model, using the Gaussian kernel. For test evaluation, We train the model using *Vecchia-approximation* (Sauer et al., 2023a) with a total of 10,000 Gibbs-sampling steps, burning the initial 8000, and thinning the remaining steps to 1,000.

Beginning with 128 equidistant samples, we refine the training data of the DGP model using the ALC criterion, *random test sampling*, and our proposed superior training scheme. The resulting training data distributions and the performance of the DGP model are plotted in Fig. 9. As expected, our superior training scheme performs superior to *random test sampling*. While the ALC criterion that is particularly designed for the DGP model performs best, we observe only very little difference at 512 training samples. Note that sampling according to the ALC criterion becomes computationally challenging already at this point since the DGP model has to be re-trained after each new sample. In contrast, sampling $\boldsymbol{X}_n \sim p^{\mathrm{GPR},n}_{\mathrm{Sup}}$ can be performed in batch mode. This result emphasizes the complementary nature of our asymptotic work to the classic *bottom-up* AL literature.

---

[4]See `https://cran.r-project.org/web/packages/deepgp/index.html`

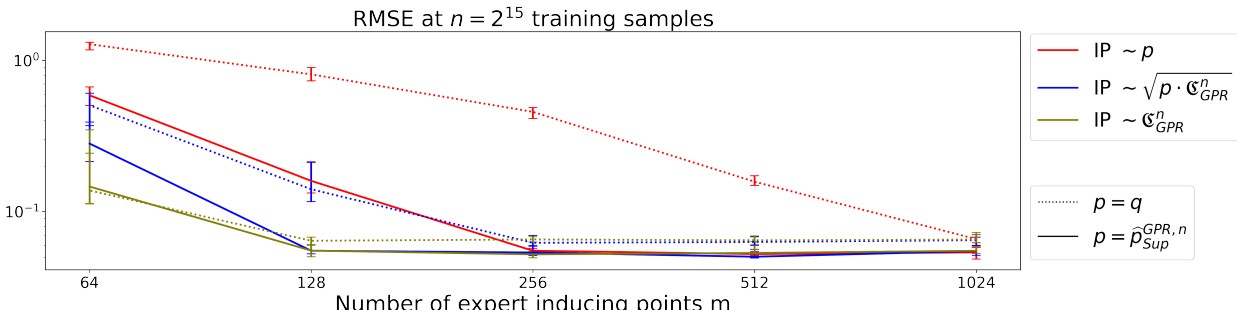

Figure 10: The Doppler experiment: The curves show the RMSE at training size $n = 2^{15}$ for a varying number of expert IPs $m$. The colors correspond to different IP distributions, whereas the line styles correspond to the underlying training distribution. The results are averaged over 20 repetitions.

**Necessity of the small bandwidth penalty**   We impose a penalty on small bandwidth choices through the factor $\vartheta_\sigma = 0.5$ to regularize the bandwidth function and prevent overfitting, as described in Appendix F. We demonstrate this overfitting issue in Appendix G.2 that results from applying no regularization ($\vartheta_\sigma = 0$).

**Parsimonious modeling using LFC**   In Sec. 4.1 we mentioned that LFC can also be used to coarsen or refine the model resolution adequately to reduce the overall complexity of the model. While we fixed the IPs to reasonable numbers in the other parts of the Doppler experiment, that is, $m = 512$ and $m = 1024$ IPs under active, respectively *random test sampling*, we here investigate the influence of the number of IPs and their distribution on the capability to resemble the Doppler function. Recall from Sec. 4.5 that we interpret the choice of the IPs as a nested AL task at small sample size ($m \ll n$), where it is reasonable for them to be sampled in a diverse way, respecting the training distribution but also the structural complexity of the target function. In Fig. 3, we show a naive choice and our optimized choice of IPs.

In Fig. 10, we compare the RMSE for the fixed training size $n = 2^{15}$ for both settings, active and passive, when sampling the IPs according to the training density $p$, the LFC and their geometric mean (29). First of all, we observe that we generally require less IPs with active sampling compared to *random test sampling*, which originates from the fact that the *superior training density* $\widehat{p}_{\mathrm{Sup}}^{\mathrm{GPR},n}$ already respects LFC to some degree. Next, we observe that the geometric mean of the training density and LFC performs best, provided that the number of IPs $m$ is large enough. Finally, we observe that, non-surprisingly, we can shrink $m$ the most under the LFC distribution, namely to $m = 128$, before the performance of the model degrades substantially.

In summary, we are able to shrink the model complexity up to a factor of 8 for the Doppler function without a significant loss of performance, when respecting LFC in the model design.

**Comparing our proposed IP selection method to a greedy fast forward selection**   In Sec. 2, we discussed other IP selection approaches. For comparison, we have implemented the *greedy fast forward* (GFF) IP selection method of Seeger et al. (2003), in which, beginning from scratch, the most informative training inputs are gradually added to the set of IPs as a means to approximate the full $\mathcal{GP}(\theta)$ distribution. Here, the information of an IP candidate $x_i \in \boldsymbol{X}_n \setminus \boldsymbol{X}_\dagger$ is measures by

$$J(x_i) = \mathcal{KL}\left[Q_{\boldsymbol{X}_\dagger \cup \{x_i\}} \| Q_{\boldsymbol{X}_\dagger}\right],$$

which is the Kullback-Leibler divergence between the posterior distributions based on the IPs $\boldsymbol{X}_\dagger \cup \{x_i\}$ and $\boldsymbol{X}_\dagger$. Accordingly, the updated set of IPs is given by $\boldsymbol{X}_\dagger \leftarrow \boldsymbol{X}_\dagger \cup \{x_i^*\}$, where

$$x_i^* = \mathbf{arg\,max}_{x_i \in \boldsymbol{X}_n \setminus \boldsymbol{X}_\dagger} J(x_i).$$

The procedure converges, when the remaining IP candidates carry no further information, that is, $J(x_i^*) < \varepsilon_{\mathcal{J}}$, up to a specified threshold $\varepsilon_{\mathcal{J}} \geq 0$.

At a given threshold $\varepsilon_{\mathcal{J}}$, we observe that the number of selected IPs is very small for the experts with a large bandwidth, while it increases drastically ($\propto \sigma_i^{-1}$) for experts with a small bandwidth. Now that the overall

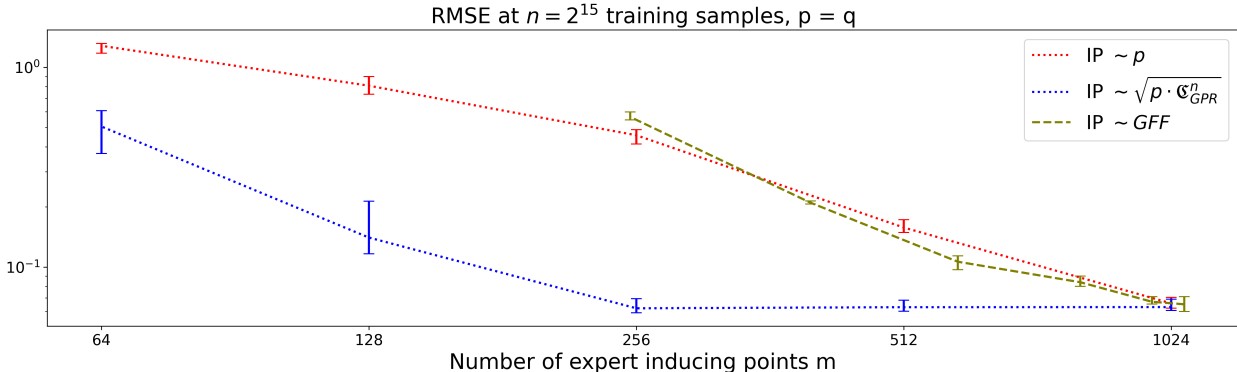

Figure 11: The Doppler experiment: The curves show the RMSE at training size $n = 2^{15}$ at a varying number of expert IPs $m$, where the IPs are either chosen at random, according to our proposed selection method, and via GFF. The results are averaged over 20 repetitions.

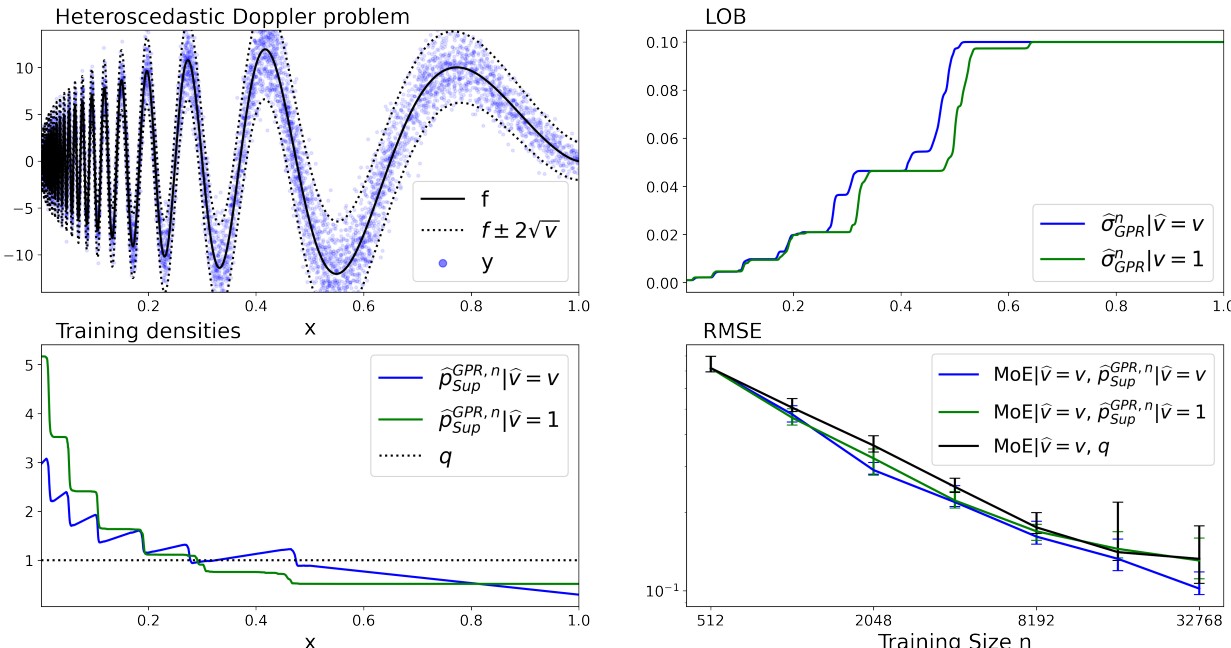

Figure 12: The Doppler experiment under heteroscedastic noise: (Top left) An exemplary dataset; (Top right) The LOB estimates, when comparing the homoscedastic to the heteroscedastic Doppler experiment; (Bottom left) The training densities of *random test sampling*, $p_{\mathrm{Sup}}^{\mathrm{GPR},n}$ and $p_{\mathrm{Sup}}^{\mathrm{GPR},n}$ when wrongly assuming homoscedasticity; (Bottom right) The RMSE at several training sizes of the compared sampling schemes. The results are averaged over 20 repetitions.

complexity of the MoE is dominated by the expert with the most IPs, it is fair to compare the number of IPs of the expert at bandwidth $\sigma_1$ with our statically specified number $m$ of IPs in Fig. 10. Here, we will vary the threshold $\varepsilon_{\mathcal{J}}$ to obtain a curve that maps the associated number of IPs to the achieved RMSE. The results in Fig. 11 show that with GFF almost no IPs can be saved for this inhomogeneously complex problem, as opposed to our proposed IP selection method.

In any case, even under training according to $p_{\mathrm{Sup}}^{\mathrm{GPR},n}$, the selected IPs by Seeger et al. (2003) are uniformly distributed. In particular, the need for less IPs of the experts at smaller bandwidths to the right of $\mathcal{X}$ is not recognized and, thus, we observe no IP savings at an acceptable performance over a random IP selection at all for this inhomogeneously complex problem.

**Heteroscedastic noise treatment**   While the treatment of heteroscedastic noise is not the main focus of this work, we will now demonstrate our approach on a heteroscedastic version of the Doppler experiment. For this, we let $v(x) = (3 - 4\,|x - 0.5|))^2 \in [1, 9]$, which we plot in Fig. 12 (top left) together with the resulting dataset. Here, we assume the local noise variance (or an estimate of it) to be provided externally, again, since its estimation is out of the scope of this work. However, note that the estimation of $v$ is well-studied in the literature, especially for GPs (Kersting et al., 2007; Cawley et al., 2006; Tresp, 2001).

In Fig. 12 (top right) we compare the LOB estimates obtained from the homoscedastic dataset and the heteroscedastic version. As we have suggested in Sec. 4.1, the influence of $v$ on the LOB estimates obtained from heteroscedastic GPR experts is relatively small. Likewise, we proceed with the evaluation of our proposed AL scheme under heteroscedasticity. Here, we compare to *random test sampling* but also to $p_{\text{Sup}}^{\text{GPR},n}$ under the wrong assumption of homoscedastic noise. Note that we use the heteroscedastic MoE in all cases since the wrong assumption of homoscedastic noise in the experts makes the MoE very volatile. The respective training densities and RMSE learning curves can be seen in the bottom row in Fig. 12. Due to the stronger noise (compared to the homoscedastic experiment), the asymptotic behavior begins to materialize later from $n = 2^{13}$ training samples. Until this point, sampling only with respect to the structural complexity looks also promising. However, as soon as the training size becomes large enough to roughly resemble the target function, respecting the inhomogeneity in the noise level becomes crucial to achieve a homogeneous pointwise convergence and, thus, maintaining asymptotic superiority.

## 5.2   Force field reconstruction

We now turn our attention to a real-world example in which we predict the *potential energy surface* (PES) and corresponding *force field* (FF) of a molecule from first-principles calculations. The PES function links the geometry $x = [R_1, \ldots, R_{\boldsymbol{a}}] \in \mathbb{R}^{3 \times \boldsymbol{a}}$ of a molecule to its potential energy $E \in \mathbb{R}$, where $R_i$ are the Cartesian positions of the $\boldsymbol{a}$ atoms of the molecule. In ab initio computations, this mapping is achieved by solving the time-independent Schrödinger equation. The PES encodes essential information on the properties of a molecule. Due to thermal and quantum effects, molecules are never perfectly rigid but assume different configurations. The distribution of these configurations is determined by the shape of the PES. For example, the minima of the PES will be sampled more frequently than other regions and correspond to stable structures. This has practical implications since many experimental techniques measure an expectation value over molecular distributions. In order to achieve a meaningful comparison, sampling needs to be taken into account in theoretical simulations as well. One of the most successful approaches to sample molecular distributions is MD simulation. They model the time evolution of the atomic positions, sampling the PES by integrating Newton's equations of motion. To this end, energy-conserving forces acting on each atom are required. These forces are the negative derivative of the PES with respect to the atomic positions $F \in \mathbb{R}^{3 \times \boldsymbol{a}}$.

This type of proxy for the prohibitively expensive ab initio quantum mechanical calculations is commonly used to enable long-timescale MD simulations that consist of millions of steps, each requiring the evaluation of the PES and FF for a new geometry. Converged MD trajectories give unique insights into the dynamic behavior and structure-function relationships of physical systems at atomic scale. They are widely used in molecular biology research and play a crucial role in applications such as protein folding and drug discovery. ML has the potential to profoundly advance this field, as it bears the promise of offering a unique cost-accuracy trade-off that is not achievable with traditional methods (Noé et al., 2020; von Lilienfeld et al., 2020; Unke et al., 2021b; Keith et al., 2021). However, some commonly deployed ML-based FFs rely on rather naive exhaustive sampling schemes to gather training data, which stands in the way of scaling to larger system sizes, both, from a data acquisition cost and training perspective. Here, we demonstrate how our method can be used to construct smaller, yet more effective training datasets.

In this experiment, we reconstruct a FF for the molecule malonaldehyde, which has $\boldsymbol{a} = 9$ atoms and the chemical formula $C_3H_4O_2$ (see Fig. 13 (A)). Formally, we try to infer the high-dimensional target function $f : \mathcal{X} \to \mathcal{Y}, R \mapsto [E, F]$, where $\mathcal{X} = \mathbb{R}^{3\boldsymbol{a}}$ and $\mathcal{Y} = \mathbb{R}^{1+3\boldsymbol{a}}$. For visualization purposes, we only show a two-dimensional subspace of the PES, which is characterized by the two main features of this molecule, its two rotors (aldehyde groups) (Chmiela et al., 2018; Sauceda et al., 2020). Their relative orientation is the dominant driver of the potential energy in this case and therefore most descriptive. Each point on the surface depicted in Fig. 13 (B) is generated by fixing the rotor pair at a particular angle and relaxing all remaining

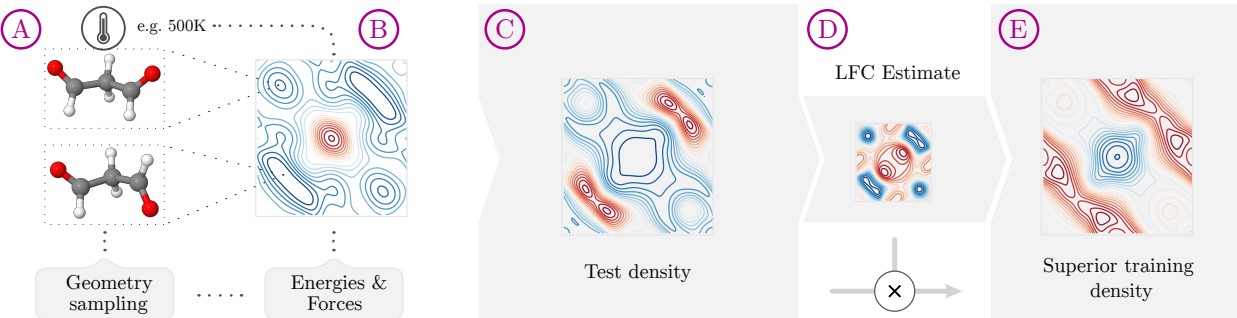

Figure 13: Reconstructing ML-based FFs using our MoE approach: (A-B) The inputs and outputs of the regression task are the geometries and energies (including forces, i.e., energy gradients) of malonaldehyde. As an example, we highlight the geometries of the two energetically stable states found in the local minima of the energy surface. (C) The density estimate of the true MD geometry distribution. (E) The *superior training density* estimate (21) based on our approach. All properties are evaluated at the relaxed malonaldehyde configurations and plotted with respect to the angles of the two aldehyde rotors of malonaldehyde (see Chmiela et al. (2018); Sauceda et al. (2020)).

degrees of freedom to obtain a minimal energy configuration. We will refer to these geometries as the *relaxed configurations* in the following.

To reconstruct the FF, we consider the broadly adopted *symmetric gradient-domain machine learning* (sGDML) method (Chmiela et al., 2018; 2019), which is a GPR model that takes energy and force labels and also roto-translational and permutational invariances of the geometries into account (see Appendix H.1 for details). We anticipate that sGDML will benefit from our MoE approach, where we deploy sGDML as the expert model, since the transition paths along the PES vary in complexity, due to the interplay between distinct atom types with different characteristic interaction length scales. Our AL approach can only improve training efficiency if there are inhomogeneities in the data. Using our LFC estimate, we therefore first verify our intuition that the PES of malonaldehyde varies in complexity. Based on this, we derive the *superior training density*, which we finally input into our AL framework to refine the training dataset in a superior way.

**Experimental setup**   All experiments use an extensive pre-computed reference trajectory (almost a million data points $(\boldsymbol{X}_{pool}, \boldsymbol{Y}_{pool})$) as ground truth, as opposed to generating new data points on demand. This test setup allows a post-hoc verification of the training distribution generated by our AL approach, while still providing ample redundancy and therefore sampling freedom.

Recall from Sec. 4.2 that we require an unnormalized density estimate of the trajectory $\boldsymbol{X}_{pool} \sim p_{\mathcal{X}}$ since we are dealing with a pool-based AL scenario. We estimate $\widehat{p}_{\mathcal{X}}$ by standard *kernel density estimation*, based on the energy-to-energy entry of the sGDML kernel $\widetilde{\boldsymbol{k}}$ from (46) at $\sigma = 0.03$. Fig. 13 (C) shows the density estimate of the relaxed configurations, where we observe that $p_{\mathcal{X}}$ is very unbalanced, with a strong concentration of mass near the stable configurations.

We implement our MoE approach, using the sGDML kernel $\widetilde{\boldsymbol{k}}$ from (46) with a Gaussian base kernel function $k$. While we sample the training data randomly (with appropriate weights) from the pool, we will draw sub-samples (i.e., for choosing the IPs of sparse expert and gate models) via symmetrized *distributional clustering* (DC) with distributional k-means++ initialization (see Appendix E).

Since this dataset comes with practically noise-free labels (we consider the first principle calculations as ground truth), we tune the experts (and MoE model) with respect to MSE rather than the Obj objective. For stability, we will apply $\widehat{v}(x) = 10^{-9}$ even though we assume no noise.

**Anisotropic bandwidths**   sGDML operates on $\boldsymbol{d} = \boldsymbol{a}(\boldsymbol{a} - 1)/2 = 36$ features that are based on the interatomic distances of the molecule. In contrast to the work of Chmiela et al. who restrict themselves to an isotropic bandwidth $\Sigma_E = \sigma_E \mathcal{I}_{\boldsymbol{d}}$, our implementation of sGDML in GPyTorch naturally enables us to tune an anisotropic bandwidth $\Sigma_E = \boldsymbol{diag}(\sigma_1, \dots, \sigma_{\boldsymbol{d}})$ in the preprocessing step.

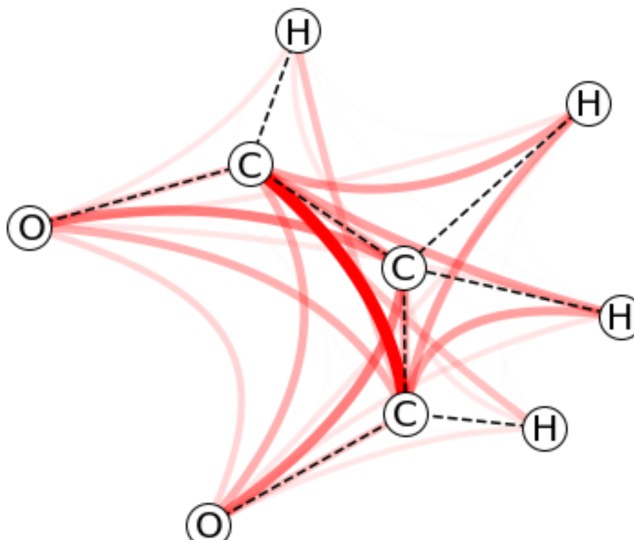

Figure 14: A visualization of the individual feature importance of malonaldehyde in the anisotropic case: The structural formula of the molecule is plotted in black. The importance of the individual interatomic distances is reciprocal to $\Sigma_E$, which is the bandwidth estimate obtained by training the anisotropic sGDML model. Hence, we express the importance of each interatomic distance of the molecule in red, where the importance corresponds to the line saturation

$$\gamma = \left[ \frac{-\log(\Sigma_E) - \min(-\log(\Sigma_E))}{\max(-\log(\Sigma_E)) - \min(-\log(\Sigma_E))} \right]^2 \in [0, 1].$$

We partially offset the increased memory footprint of the model due to the tunable $\Sigma_E$ by implementing the sparse GPR model from Sec. 3.4.1 under the sGDML kernel $\widetilde{\boldsymbol{k}}$ from (46) and limiting the number of IPs to $m = 128$ configurations. Since all our features are of the same type—pairwise interatomic distances—they are inherently calibrated in terms of scale. Hence, the reciprocal entries of $\Sigma_E$ directly translate into the importance of the features, which we display in Fig. 14.

We observe, that the importance assigned to some pairs of atoms agrees with chemical intuition, e.g., interactions with light hydrogen atoms are generally weaker. Furthermore, the important role of the opposing aldehyde groups in malonaldehyde emerges in the form of a heavily weighted path that connects the O-C-C-C-O backbone of the molecule.

In Fig. 15 we see that our anisotropic variant of sGDML performs consistently better than the original isotropic sGDML model. Similar to the calculation of the relative sample size in (31) we can compare two models of equal asymptotic MSE law. When comparing anisotropic to isotropic sGDML, both under *random test sampling*, we can save about 10% of samples.

**Setting up the MoE model** After having trained $\Sigma_E$, we apply dense sGDML experts with $\Sigma_j = \sigma_j \Sigma_E$, where $\sigma_j = 2^{-5/4+j/2}, 1 \leq j \leq 8$ as the individual expert bandwidths, $\lambda_E = 1$ as the initial expert regularization, and $\sigma_G = 0.1$ and $\lambda_G = 10^4$ for the sparse gate with 1024 IPs. For the training, we apply a batch size of $B = 1024$, a terminal expert sparsity $\kappa = 8$, a penalty factor of $\vartheta_\sigma = 0.01$ for small bandwidth choices, gate noise parameters $\mathfrak{s}_0 = 0.01$ and $\eta_\mathfrak{s} = 1/\sqrt{2}$, and learning rate parameters $\eta = 0.005$, $\eta_H = 0.05$, $\eta_G = 0.1$. As we discuss in Appendix F, for tuning the MoE with dense (sGDML) experts, we either require an additional gate training set, which is independent of the training set for the experts, or we could provide leave-one-out (LOO) responses of the experts for the training of the gate. In our experiment, we use an additional gate training set $\boldsymbol{X}_{n_G}^G$ of fixed size $n_G = 2^{14}$. The anisotropic MoE model performs consistently better than anisotropic sGDML, as can be seen in Fig. 15. When comparing the anisotropic MoE model to isotropic sGDML, both under *random test sampling*, we can save about 21% of samples.

**Active learning** We assume an intrinsic dimension of $\delta = 2$ (the two aldehyde rotor angles, the most salient features of malonaldehyde) and a smooth target function $f \in \mathcal{C}^\infty(\mathcal{X}, \mathcal{Y})$. The test distribution is given by the MD trajectory such that $q = p_\mathcal{X}$. Prior to the AL procedure, we separate the validation samples $\boldsymbol{X}_{\mathrm{val}}$ and test samples $\boldsymbol{X}_{\mathrm{T}}$ at random from the pool $\boldsymbol{X}_{pool}$. We apply an initial expert training size of $n_0 = 2^9$, doubling the sample size with each iteration of the AL procedure. The initial expert training set $\boldsymbol{X}_{n_0}$ and the gate training set $\boldsymbol{X}_{n_G}^G$ are drawn via importance sampling from the remaining pool with weights $\widehat{p}_\mathcal{X}^{-1/2}(\boldsymbol{X}_{pool} \setminus (\boldsymbol{X}_{\mathrm{val}} \cup \boldsymbol{X}_{\mathrm{T}}))$. By this it is $\boldsymbol{X}_{n_0} \sim q^{1/2}$, which is more in alignment with the *superior training density* (21) than sampling $\boldsymbol{X}_{n_0} \sim q$.

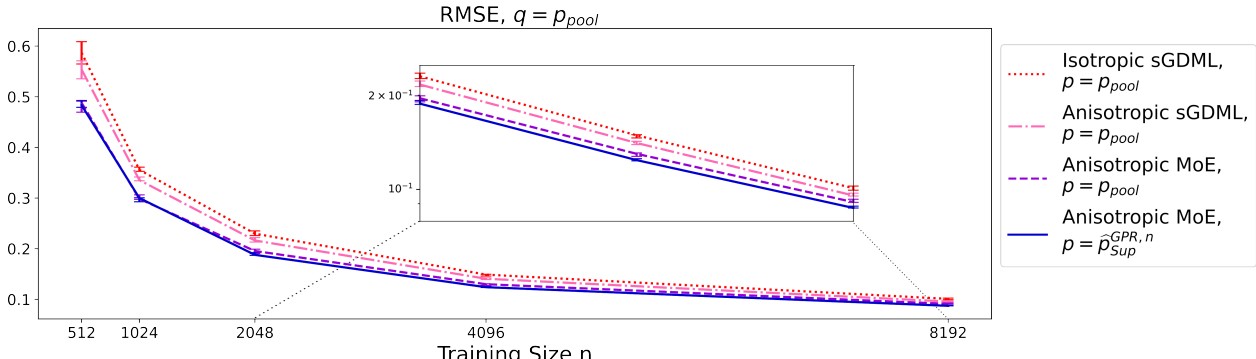

Figure 15: The RMSE under the true MD trajectory test distribution for different variants of sGDML and training distribution at varying training size: The performance is given for passive sampling, using the original isotropic sGDML (dotted), anisotropic sGDML (dash-dotted) and our MoE model with anisotropic sGDML experts (dashed), and for the MoE model, applying the proposed *superior sampling scheme* (solid). The results are averaged over 5 repetitions.

In Fig. 13 (D, E) we show the estimates of LFC and the *superior training density* under the pool test distribution, evaluated on the relaxed configurations of malonaldehyde. The LFC estimates confirm our expectation that the transition areas are more complex to model than the regions near the stable configurations. Subsequently, our active sampling scheme shifts sample mass away from the stable regimes in favor of the transition areas.

We have plotted the error curves of passive and active sampling schemes in Fig. 15. When estimating the relative sample size (31) that we require to achieve the same RMSE via active sampling compared to *random test sampling*, we obtain $\varrho(\widehat{f}_{\mathrm{MoE}}, \widehat{p}_{\mathrm{Sup}}^{\mathrm{GPR}, n}) = 0.920 \pm 0.013$. This means that we save about 8% of samples under the MoE model with our active sampling scheme compared to *random test sampling*. In total, when comparing our actively trained MoE approach to the passively trained, original sGDML model, we can save about 31% of samples. Notably, DFT level calculations (Perdew et al., 1996; Blum et al., 2009; Tkatchenko & Scheffler, 2009) for the studied system require minutes to hours of computation *per sample*, CCSD(T) level computations even require days of computation per sample. So in the field of quantum chemistry saving roughly a third of computing power is of practical importance.

## 6    Discussion

**Active learning**   Recall that in this work we have restricted ourselves to the scenario of model-agnostic AL with persistent performance at large training size. This scenario is complementary to the more common small sample size regime. And while both cases are important, a lot of AL related work (as discussed in Sec. 2) does not apply to our AL scenario. We also discussed and demonstrated in our experiments that input space geometric arguments which are model-free asymptotically come with no benefit over *random test sampling*. Since our proposed model is GPR-based, we also analyzed uncertainty sampling (MoGPU) for our MoE model to show that our proposed *superior sampling scheme* differs from uncertainty sampling. Moreover, our *superior sampling scheme* showed to be superior to uncertainty sampling. Finally, in the regime of our AL scenario, the approach by Panknin et al. (2021) recently has demonstrated state-of-the-art performance to recent, sophisticated, model-agnostic AL approaches. This was done by training different models on the actively constructed training sets of their and other AL approaches and assessing their performance. In particular, they compared favorably to Goetz et al. (2018)—a random tree-based AL approach—in a heteroscedastic setting, using a regression forest model and to Bull et al. (2013)—a wavelet-based AL approach—in a setting of inhomogeneous complexity, using an RBF-network. This demonstrates the flexibility of this AL approach in terms of learning problem specifications as well as model choices. Now that our work builds on the previous work of Panknin et al. (2021), state-of-the-art performance of our work is implied.

**Interpretability of LFC**   Due to model-agnosticity, we consider LFC to be an intrinsic, *interpretable* property of the regression problem, which can be used as an analysis tool by domain experts:

When looking at a single point in a high-dimensional input space, a visual assessment of the local structural complexity (e.g., a human can visually detect more complexity to the left of the Doppler function) is challenging. Here, the scalar LFC value gives a human assessable, quantitative description. The LFC (as a scalar-valued function) then even allows for an easy visualization of the local structural complexity in high-dimensional input spaces, if the input space features a reasonable low-dimensional projection. This benefit was demonstrated in the high-dimensional FF reconstruction experiment, where the two-dimensional visualization of LFC provides new insights into the regression problem. Note that the LFC function cannot be visualized in the absence of a low-dimensional projection.

**Parsimonious modeling**   We proposed a novel, model-agnostic approach to select the IPs of GPR, sampling them in a diverse way from a distribution that is representative for the training data and respects the LFC. In the experiments, we have seen that for problems of inhomogeneous complexity, our approach sustains the expressive power of the model at a considerably smaller number of IPs, compared to the GFF IP selection method of Seeger et al. (2003).

**Heteroscedasticity**   While inhomogeneities in noise are not the focus of our work, note that both, our model as well as the original AL framework upon which we built our approach naturally deal with heteroscedasticity. It, therefore, suffices to complement our work with an estimate of the noise variance function $v$ as, e.g., given in Kersting et al. (2007); Cawley et al. (2006). The only aspect left open is to elaborate on the impact of $v$ on the LOB of GPR and, thus, the adequate adjustment with respect to $v$ in the derivation of the LFC. As we argued, GPR already treats heteroscedasticity through local adaptions of the regularization. Hence, we assume the influence of $v$ on the LOB to be negligible to not existent. This is opposed to the LOB of LPS whose only way to deal with heteroscedasticity is through adaption of its LOB.

**Intrinsic dimension and smoothness of the problem**   In our derivation of LFC and the *superior training density*, we assumed the intrinsic dimension $\delta \leq d$ and the smoothness $\alpha \in (0, \infty]$ of $f \in \mathcal{C}^\alpha(\mathcal{X}, \mathbb{R})$ to be given through domain knowledge.

If $\delta$ is unknown, we can estimate it from unlabeled input instances, such as $\boldsymbol{X}_{pool}$ in a pre-processing step. However, this is beyond the scope of our work and we refer to the approach of Facco et al. (2017)[5] for an estimate of the $\delta$ for unbalanced input distributions in high-dimensional input spaces $\mathcal{X}$.

If we have no ground truth knowledge about the smoothness $\alpha$ of the target function $f \in \mathcal{C}^\alpha(\mathcal{X}, \mathbb{R})$, we resort to $\widehat{\alpha} = \infty$ as default in practice. This assumption is justified, as long as $f$ happens to be rougher in at most finitely many locations of the input space. Since, asymptotically, the violation of $\alpha = \infty$ affects only a set of measure zero, the influence on our AL setting that addresses large training sizes is marginal. We consider the restriction to target functions that are at most rough on a finite set of input space locations a weak assumption which, thus, comes with no practical limitations.

One way to deal with an unknown smoothness $\alpha$ is to deploy the *Matérn* kernel

$$k_\nu(x, x') := \frac{2^{1-\nu}}{\Gamma(\nu)} \left( \|x - x'\|/\sigma \right)^\nu \widetilde{k}_\nu \left( \|x - x'\|/\sigma \right),$$

where $\Gamma$ is the *gamma function* and $\widetilde{k}_\nu$ is the modified Bessel function of the second kind or order $\nu$. After finding the best fitting $\nu^*$, we obtain by $\widehat{\alpha} := \lceil \nu^* \rceil - 1$ a reasonable estimate to $\alpha$. We defer this idea to future work as it is beyond the scope of this work.

**Dimensional scaling**   As opposed to nonstationary GP approaches (e.g., the tree-based or local GP by Gramacy & Lee (2008); Gramacy & Apley (2015) that suffer from the curse of dimensionality through input space localization, we segment the input space into a fixed number $L$ of patches, given by the $L$ experts of our MoE. Thus, if we were to instantiate our MoE with dense GPR models, our approach scales well concerning the input space dimension $d$. However, in real-world applications, we typically deal with training sizes that are too large for dense modeling. In this regime, sparse GPR representations scale poorly in $d$ as their IPs

---

[5]For an implementation in Python see `https://scikit-dimension.readthedocs.io/en/latest/index.html`

must be space-filling (Binois & Wycoff, 2022). Likewise, our density-based AL approach encounters decaying power for large $d$. A low intrinsic dimension ($\delta < d$) of the regression problem is therefore crucial for our work to apply.

**LFC and the *superior sampling scheme* as a ML concept** Recall that we consider LFC to be a problem intrinsic property. Here, the problem is characterized by features $x \in \mathcal{X}$ with labels $f(x) \in \mathbb{R}$, a hypothesis space of *locally adaptive models*, and the MISE as the loss function (or rather the pointwise MSE from (1) due to the localization of LFC). In this sense, LFC is formally a property of the combination of model and loss according to the characterization of (Jung (2022), Chapter 2). Similarly, the *superior training density* is a property of the combination of model and loss with respect to the same hypothesis space and MISE instead of the pointwise MSE as loss function.

**On a realistic implementation in ab initio FF reconstruction** In our FF reconstruction experiment, we assumed a large unlabeled reference trajectory $\boldsymbol{X}_{pool}$ to be given that already follows the true molecular distribution. This will not be given in practice, since building the input trajectory already requires the computationally expensive estimation of the respective labels. At this point, the actual task behind the regression problem would already be solved. We outline a realistic ab initio FF reconstruction scenario in Appendix H.3.

## 7 Conclusion

Standard ML tasks implicitly assume a certain homogeneity in the data scales. However, in practice this structural property of the learning problem may not be fulfilled, e.g., in multiscale problems from the sciences such as turbulence (Brunton et al., 2020) or quantum chemistry (Noé et al., 2020; von Lilienfeld et al., 2020; Unke et al., 2021b; Keith et al., 2021).

In this work, we aimed to identify local inhomogeneities in regression tasks, which can be used to construct better models and training datasets and for domain interpretation. To this end, we combined recent results on model-agnostic LFC estimates and asymptotically optimal sampling, which are founded in the domain of LPS, with estimates of LOB, which are derived in the GPR domain. By this, we benefit from both sides, having a theoretically sound superior sampling scheme on the one hand, and having access to the required estimates from a model that naturally can cope with high input space dimensions on the other hand. Furthermore, we have shown how respecting LFC in the selection of IPs contributes to parsimonious modeling.

On synthetic data, we showcased and validated our approach, where we analyzed similarities with the LPS-based analog but also compared to the most related GP uncertainty sampling concepts for AL. To show the full potential of our approach, we studied a real-world, high-dimensional force field reconstruction task. Our approach not only gave access to an interpretable visualization of the inhomogeneous structural complexity but also guided the sampling process in a way that takes the structural changes into account, enhancing the quality of the training data. Here, we additionally identified the multi-scale structure of the individual atomic interactions, whose treatment also results in a substantial performance gain of the broadly adopted method sGDML.

**Future work** In Sec. 4.1 we conjecture that the LOB of heteroscedastic GPR is invariant or scales at most weakly with respect to the local noise level $v(x)$. This claim should be supported by further theoretical investigation. While we deployed our estimates of LFC and the superior training density, using $\widehat{\alpha} = \infty$, if the smoothness of the target function $f \in \mathcal{C}^{\alpha}(\mathcal{X}, \mathbb{R})$ is unknown, it is possible to (re-)estimate $\widehat{\alpha}$, e.g., by tuning the regularity of the Matérn kernel of a GPR model after the acquisition of each new training data batch. While we have compared to baseline IP selection methods, a thorough comparison to more sophisticated approaches remains open. A promising idea is also to combine our LFC estimate with the IP selection approach by Moss et al. (2023) to obtain informative and diverse IPs in GPR. Finally, we will focus on the application of our approach to real-world problems from chemistry, physics, and further domains also applying techniques from *explainable AI* (e.g. Samek et al. (2021); Letzgus et al. (2022)). In particular, recent advances on sGDML regarding the scalability by Chmiela et al. (2023) will enable the application of our approach to large molecular systems.

**Acknowledgments**

D. Panknin, S. Chmiela, S. Nakajima, and K.-R. Müller were funded by the German Ministry for Education and Research as BIFOLD - Berlin Institute for the Foundations of Learning and Data (ref. BIFOLD23B). D. Panknin was also supported by the BMBF project ALICE III, Autonomous Learning in Complex Environments (01IS18049B). K.-R. Müller was also supported by the BMBF Grants 01GQ1115 and 01GQ0850, under the Grants 01IS14013A-E, 031L0207A-D; DFG under Grant Math+, EXC 2046/1, Project ID 390685689 and by the Institute of Information & Communications Technology Planning & Evaluation (IITP) grants funded by the Korea Government (No. 2017-0-00451, Development of BCI based Brain and Cognitive Computing Technology for Recognizing User's Intentions using Deep Learning) and funded by the Korea Government (No. 2019-0-00079, Artificial Intelligence Graduate School Program, Korea University).

All funding sources were not involved in the process of writing and submitting this work.

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

# A   Asymptotic results for local polynomial smoothing

In this section, we will review the theory of Panknin et al. (2021).

The prediction of the LPS model of order $Q$ under the bandwidth $\Sigma \in \mathbb{S}_{++}^d$ in $x \in \mathcal{X}$ can be understood as follows: First, the regression problem is localized around $x$ according to weights $k^\Sigma(\cdot, x)$ that decrease with growing distance to $x$. Then we search for the polynomial up to order $Q$ that fits the localized regression problem best. Finally, the evaluation of this polynomial in $x$ is returned as the prediction. Formally, it is

$$m_Q^\Sigma(x) = \mathfrak{p}_{Q,\Sigma,x}^*(0), \text{ where} \tag{33}$$
$$\mathfrak{p}_{Q,\Sigma,x}^* = \underset{\mathfrak{p} \in \mathcal{P}_Q(\mathbb{R}^d)}{\textbf{arg\,min}} \sum_{i=1}^n k^\Sigma(x_i, x) \left(y_i - \mathfrak{p}(x_i - x)\right)^2,$$

and $\mathcal{P}_Q(\mathbb{R}^d)$ is the space of the real polynomial mappings $\mathfrak{p}\colon \mathbb{R}^d \to \mathbb{R}$ up to order $Q$.

The localization is controlled by $\Sigma$ through the kernel weights $\boldsymbol{K}^\Sigma(x, x_i)$ for $x_i \in \boldsymbol{X}_n$: For an RBF-kernel, $k^\Sigma(x, x')$ decays monotonically with growing distance of $x'$ to $x$. This decay is dampened or amplified as $\Sigma$ increases or decreases, respectively (in the sense of the Loewner order).

For readability, since $\Sigma$ will be replaced by terms with more involved notation, we redefine (1) by

$$\text{MSE}\left(x, \widehat{f}, \Sigma | \boldsymbol{X}_n\right) := \text{MSE}\left(x, \widehat{f}^\Sigma | \boldsymbol{X}_n\right). \tag{34}$$

For a bandwidth space $\mathcal{S} \subseteq \mathbb{S}_{++}^d$, Panknin et al. (2021) proposed to minimize the AL objective

$$\text{MISE}\left(q, \widehat{f} | \boldsymbol{X}_n\right) = \int_\mathcal{X} \inf_{\Sigma \in \mathcal{S}} \text{MSE}\left(x, \widehat{f}, \Sigma | \boldsymbol{X}_n\right) q(x) dx, \tag{35}$$

which is the optimal MISE, obtained by predictions that are based on locally optimal chosen bandwidths. If these locally optimal bandwidth choices are well-defined, that is, if for all $x \in \mathcal{X}$ there exists a unique $\Sigma' \in \mathcal{S}$ such that

$$\text{MSE}\left(x, \widehat{f}, \Sigma' | \boldsymbol{X}_n\right) = \inf_{\Sigma \in \mathcal{S}} \text{MSE}\left(x, \widehat{f}, \Sigma | \boldsymbol{X}_n\right),$$

we are able to define the LOB function

$$\Sigma^n(x) = \textbf{arg\,min}_{\Sigma \in \mathcal{S}} \text{MSE}\left(x, \widehat{f}, \Sigma | \boldsymbol{X}_n\right).$$

This function exists, for example, in the *isotropic* case $\mathcal{S} = \{\sigma \mathcal{I}_d \mid \sigma > 0\}$ for LPS under mild conditions, where we denote $\Sigma^n(x) = \sigma^n(x)\mathcal{I}_d$ (see, e.g., Masry (1996; 1997); Fan et al. (1997) or Panknin et al. (2021) for an overview).

Assuming the isotropic bandwidths candidate space $\mathcal{S} = \{\sigma \mathcal{I}_d \mid \sigma > 0\}$, the LOB as in Eq. (4) is an asymptotically well-defined function under mild assumptions[6]: Denoting the LOB of LPS of order $Q$ by $\Sigma_Q^n(x) = \sigma_Q^n(x)\mathcal{I}_d$ such that

$$\sigma_Q^n(x) = \textbf{arg\,min}_{\sigma > 0} \text{MSE}\left(x, m_Q, \sigma \mathcal{I}_d | \boldsymbol{X}_n\right), \tag{36}$$

asymptotically it holds

$$\sigma_Q^n(x) = C_Q \left[\frac{v(x)}{p(x)n}\right]^{\frac{1}{2(Q+1)+d}} \text{b}_Q\left[x, \mathcal{I}_d\right]^{-\frac{2}{2(Q+1)+d}} + o_p\left[n^{-\frac{1}{2(Q+1)+d}}\right], \tag{37}$$

where $C_Q$ is a constant, and $\text{b}_Q\left[x, \mathcal{I}_d\right]$ is a function of $x$ taken from the asymptotic *conditional bias* $f(x) - \mathbb{E}\left[m_Q^{h_n \mathcal{I}_d}(x) \Big| \boldsymbol{X}_n\right]$ of LPS (Masry, 1996; 1997). That is, for a sequence $h_n \to 0$ as $n \to \infty$ we can write the conditional bias, which is of order $Q+1$, as

$$f(x) - \mathbb{E}\left[m_Q^{h_n \mathcal{I}_d}(x) \Big| \boldsymbol{X}_n\right] = h_n^{Q+1} \text{b}_Q\left[x, \mathcal{I}_d\right] + o_p\left[h_n^{Q+1}\right]. \tag{38}$$

---

[6]We require non-vanishing leading bias- and variance-terms of $m_Q(x)$, which is guaranteed if $\forall x \in \mathcal{X}$ it holds that $\text{b}_Q\left[x, \mathcal{I}_d\right] \neq 0$ from Eq. (38) and $v(x) > 0$.

Eq. (37) shows how LOB scales asymptotically with respect to the training size $n$, the local noise level function $v(x)$ and the training density $p(x)$. The remaining bias component depends on the local structural complexity, which can be characterized by the derivatives of $f$ in a non-trivial way. Therefore it encodes the local structural complexity of $f$. Given all other properties and LOB itself, we are able to formulate LFC in a closed form.

**Definition 6** (Panknin et al. (2021)). *For LPS of order $Q$, the LFC of $f$ in $x \in \mathcal{X}$ is asymptotically given by*

$$\mathfrak{C}_Q^n(x) = \left[ \frac{v(x)}{p(x)n} \right]^{\frac{d}{2(Q+1)+d}} \left| \Sigma_Q^n(x) \right|^{-1} = \left[ \frac{v(x)}{p(x)n} \right]^{\frac{d}{2(Q+1)+d}} \sigma_Q^n(x)^{-d}.$$

As already mentioned in Eq. (35), given a test density $q$, the AL task is to minimize $\mathrm{MISE}\left(q, m_Q | \boldsymbol{X}_n\right)$. Now, if LOB is well-defined, we can rewrite

$$\mathrm{MISE}\left(q, m_Q | \boldsymbol{X}_n\right) = \int_{\mathcal{X}} \inf_{\Sigma \in \mathcal{S}} \mathrm{MSE}\left(x, m_Q, \Sigma | \boldsymbol{X}_n\right) q(x) dx$$

$$= \int_{\mathcal{X}} \mathrm{MSE}\left(x, m_Q, \Sigma_Q^n(x) | \boldsymbol{X}_n\right) q(x) dx.$$

Finally, when solving for the optimal training dataset

$$\boldsymbol{X}_n' \approx \mathbf{argmin}_{\boldsymbol{X}_n \in \mathcal{X}^n} \mathrm{MISE}\left(q, m_Q | \boldsymbol{X}_n\right),$$

as in Eq. (3), the optimal training inputs $\boldsymbol{X}_n'$ can be written asymptotically as an independent and identically distributed sample from the optimal training distribution, whose density $p_{\mathrm{Opt}}^{Q,n}$ possesses an asymptotic closed form.

**Theorem 7** (Panknin et al. (2021)). *Let $v, q \in \mathcal{C}^0\left(\mathcal{X}, \mathbb{R}_+\right)$ for a compact input space $\mathcal{X}$, where $q$ is a test probability density. Additionally, assume that $v$ and $q$ are bounded away from zero. I.e., $v, q \geq \epsilon$ for some $\epsilon > 0$. Let $k$ be a RBF-kernel with bandwidth parameter space $\mathcal{S} = \{\sigma \mathcal{I}_d \mid \sigma > 0\}$. Let $Q \in \mathbb{N}$ be odd and $f \in \mathcal{C}^{Q+1}\left(\mathcal{X}\right)$ such that the bias of order $Q+1$ does not vanish almost everywhere. Then the optimal training density for LPS of order $Q$ is asymptotically given by*

$$p_{Opt}^{Q,n}(x) \propto \left[\mathfrak{C}_Q^n(x)q(x)\right]^{\frac{2(Q+1)+d}{4(Q+1)+d}} v(x)^{\frac{2(Q+1)}{4(Q+1)+d}} \left(1 + o(1)\right).$$

We will use this optimal distribution to sample $\boldsymbol{X}_n' \sim p_{\mathrm{Opt}}^{Q,n}$ with a proposed estimator for $\mathfrak{C}_Q^n$ that is scalable with respect to the input space dimension.

For LPS with $\boldsymbol{X}_n' \sim p$ and $\boldsymbol{X}_n \sim q$, we can asymptotically calculate the relative required sample size from Definition 2 in Sec. 4.1 by

$$\varrho(m_Q, p) = \left[ \frac{\mathrm{MISE}\left(q, m_Q | \boldsymbol{X}_n'\right)}{\mathrm{MISE}\left(q, m_Q | \boldsymbol{X}_n\right)} \right]^{\frac{2(Q+1)+d}{2(Q+1)}}. \tag{39}$$

## B  Analytic GPR formulations

### B.1  Classical Gaussian process regression

The GPR model $\widehat{y} \sim \mathcal{GP}(\theta)$ (see, e.g. Williams & Rasmussen (1996)) is defined as follows: The GP is described by the hyperparameters $\theta = (\mu, \lambda, \widehat{v}, \Sigma)$, which are the global constant prior mean $\mu$, the regularization parameter $\lambda$, the label noise variance function $\widehat{v}$ and the bandwidth matrix $\Sigma$ of the kernel. If we can assume homoscedastic noise, we let $\widehat{v}(x) \equiv \sigma_\varepsilon^2$.

The GP prior then assumes the labels $\boldsymbol{Y}_n$ of $\boldsymbol{X}_n$ to be distributed according to $\boldsymbol{Y}_n = \widehat{y}(\boldsymbol{X}_n) \sim \mathcal{N}(\cdot; \boldsymbol{\mu}(\boldsymbol{X}_n), \boldsymbol{C}(\boldsymbol{X}_n) | \theta)$, for the constant mean function $\boldsymbol{\mu}(\boldsymbol{X}_n) = \mu \mathbb{1}_n$, and the covariance function

$$\boldsymbol{C}(\boldsymbol{X}_n) = \lambda \boldsymbol{K}_n + \boldsymbol{diag}(\widehat{v}(\boldsymbol{X}_n)),$$

where $\boldsymbol{K}_n = \boldsymbol{K}^{\Sigma}(\boldsymbol{X}_n)$ is the kernel matrix of $\boldsymbol{X}_n$.

For test inputs $\boldsymbol{X}_*$, the posterior predictive distribution of $\boldsymbol{Y}_*$ is then given by

$$\widehat{y}(\boldsymbol{X}_*) \sim \mathcal{N}(\cdot\,; \boldsymbol{\mu}^*(\boldsymbol{X}_*), \boldsymbol{C}^*(\boldsymbol{X}_*)|\theta),$$

where the predictive mean and covariance are given by

$$\boldsymbol{\mu}^*(\boldsymbol{X}_*) = \boldsymbol{\mu}(\boldsymbol{X}_*) + \boldsymbol{C}_{*n}\boldsymbol{C}_n^{-1}(\boldsymbol{Y}_n - \boldsymbol{\mu}(\boldsymbol{X}_n)), \tag{40}$$
$$\boldsymbol{C}^*(\boldsymbol{X}_*) = \boldsymbol{C}_* - \boldsymbol{C}_{*n}\boldsymbol{C}_n^{-1}\boldsymbol{C}_{*n}^{\top}, \tag{41}$$

and we have defined

$$\boldsymbol{C}(\boldsymbol{X}_n \cup \boldsymbol{X}_*) = \begin{bmatrix} \boldsymbol{C}_n & \boldsymbol{C}_{*n}^{\top} \\ \boldsymbol{C}_{*n} & \boldsymbol{C}_* \end{bmatrix}.$$

## B.2 Analytic sparse Gaussian processes

We define the sparse GPR model $\widehat{y} \sim \mathcal{SGP}(\theta)$ as follows, following Snelson & Ghahramani (2005): The sparse GP is described by the (hyper-) parameters $\theta = (\mu, \lambda, \widehat{v}, \Sigma, \boldsymbol{X}_{\dagger})$, which are the global constant prior mean $\mu$, the regularization parameter $\lambda$, the label noise variance function $\widehat{v}$, the bandwidth matrix $\Sigma$ of the kernel and the prior distribution, given by the IP locations $\boldsymbol{X}_{\dagger} \in \mathcal{X}^m$.

Here, the degree of sparsity is described by $m$ IPs: This number can be fixed in advance or gradually increased with training size $n$, where the increase $m_n = o[n]$ is typically much slower than $n$. If we can assume homoscedastic noise, we let $\widehat{v}(x) \equiv \sigma_{\varepsilon}^2$.

The sparse GP then outputs

$$\widehat{y}(\boldsymbol{X}_*) \sim \mathcal{N}(\cdot\,; \boldsymbol{\mu}^*(\boldsymbol{X}_*), \boldsymbol{C}^*(\boldsymbol{X}_*)|\theta_e)$$

for the mean function

$$\boldsymbol{\mu}^*(\boldsymbol{X}_*) = \boldsymbol{K}_{*\dagger}\boldsymbol{Q}_{\dagger}^{-1}\boldsymbol{K}_{n\dagger}^{\top}(\Lambda + \boldsymbol{diag}(\widehat{v}(\boldsymbol{X}_n)))^{-1}(\boldsymbol{Y}_n - \boldsymbol{\mu}(\boldsymbol{X}_n))$$

and the covariance function

$$\boldsymbol{C}^*(\boldsymbol{X}_*) = \boldsymbol{K}_* - \boldsymbol{K}_{*\dagger}(\boldsymbol{K}_{\dagger}^{-1} - \boldsymbol{Q}_{\dagger}^{-1})\boldsymbol{K}_{*\dagger}^{\top} + \boldsymbol{diag}(\widehat{v}(\boldsymbol{X}_*))$$

where we have defined $\boldsymbol{K}_{\dagger} = \boldsymbol{K}^{\Sigma}(\boldsymbol{X}_{\dagger})$, $\boldsymbol{K}_n = \boldsymbol{K}^{\Sigma}(\boldsymbol{X}_n)$, $\boldsymbol{K}_{*\dagger} = \boldsymbol{K}^{\Sigma}(\boldsymbol{X}_*, \boldsymbol{X}_{\dagger})$, $\boldsymbol{K}_{n\dagger} = \boldsymbol{K}^{\Sigma}(\boldsymbol{X}_n, \boldsymbol{X}_{\dagger})$, $\boldsymbol{Q}_{\dagger} = \boldsymbol{K}_{\dagger} + \boldsymbol{K}_{n\dagger}^{\top}(\Lambda + \boldsymbol{diag}(\widehat{v}(\boldsymbol{X}_n)))^{-1}\boldsymbol{K}_{n\dagger}$, and $\Lambda = \boldsymbol{diag}(\boldsymbol{\lambda})$ with $\boldsymbol{\lambda} = \boldsymbol{diag}(\boldsymbol{K}_n + \boldsymbol{K}_{n\dagger}\boldsymbol{K}_{\dagger}^{-1}\boldsymbol{K}_{n\dagger}^{\top})$.

We choose $\boldsymbol{\mu}$ to be the constant mean function, i.e., $\boldsymbol{\mu}(X) = \mu\mathbb{1}_n$ for $X \in \mathcal{X}^n$, noting that other mean functions are possible.

# C  LFC of GPR

**Theorem 1** (LFC of GPR). *For $f \in \mathcal{C}^{\alpha}(\mathcal{X}, \mathbb{R})$, $\boldsymbol{X}_n \sim p$ and homoscedastic noise, the GPR-based LFC estimate of $f$ in $x \in \mathcal{X}$ is asymptotically given by*

$$\mathfrak{C}_{GPR}^n(x) := \left[\frac{1}{p(x)n}\right]^{\frac{d}{2\alpha+d}} |\Sigma_{GPR}^n(x)|^{-1}. \tag{19}$$

*Proof.* Let $\mathcal{X} = \biguplus_{i=1}^k \mathcal{X}_i^k$ be a segmentation of the input space with non-empty interiors $(\mathcal{X}_1^k)^{\circ}, \dots, (\mathcal{X}_k^k)^{\circ} \neq \emptyset$, over which we can define the restricted bandwidth function search space

$$\boldsymbol{S}_k = \left\{ \Sigma(x) = \sum_{i=1}^k \mathbb{1}_{\mathcal{X}_i^k}(x)\Sigma_i \,\middle|\, \Sigma_1, \dots, \Sigma_k \in \mathcal{S} \right\}.$$

Here, $\mathbb{1}_A(z)$ is the indicator function, returning 1 for $z \in A$ and 0, else. Furthermore let $\Sigma^{k,n} \in \boldsymbol{S}_k$ be the minimizer of the MISE over $\boldsymbol{S}_k$ with $\Sigma^{k,n}(x) = \sum_{i=1}^{k} \mathbb{1}_{\mathcal{X}_i^k}(x)\Sigma_i^{k,n}$ such that

$$\int_{\mathcal{X}} \mathrm{MSE}\left(x, \widehat{f}^{\Sigma^{k,n}(x)}|\boldsymbol{X}_n\right) q(x)dx = \min_{\Sigma \in \boldsymbol{S}_k} \int_{\mathcal{X}} \mathrm{MSE}\left(x, \widehat{f}^{\Sigma(x)}|\boldsymbol{X}_n\right) q(x)dx.$$

Recall from (9) that $\Sigma_{\mathrm{GPR}}^n \propto n^{-\frac{1}{2\alpha+d}}$ generally holds for arbitrary input spaces. Due to this, asymptotically, $\widehat{f}_{|\mathcal{X}_i^k}$ does not depend on training samples outside $\mathcal{X}_i^k$. Hence, letting $\boldsymbol{X}_{i,n} := \{x \in \boldsymbol{X}_n \mid x_i \in \mathcal{X}_i^k\}$, the individual $\Sigma_i^{k,n}$ are asymptotically found by solving the isolated segments of the objective

$$\int_{\mathcal{X}_i^k} \mathrm{MSE}\left(x, \widehat{f}^{\Sigma_i^{k,n}}|\boldsymbol{X}_{i,n}\right) q(x)dx = \min_{\Sigma \in \mathcal{S}} \int_{\mathcal{X}_i^k} \mathrm{MSE}\left(x, \widehat{f}^{\Sigma}|\boldsymbol{X}_{i,n}\right) q(x)dx.$$

First of all, it is $\mathbb{E}\,\boldsymbol{X}_{i,n} = p(\mathcal{X}_i^k)n$, where $p(A) := \int_A p(x)dx$ is the probability for a training sample to fall into $A \subset \mathcal{X}$. In addition, we need to account for the expanse of $\mathcal{X}_i^{k_n}$, which we measure by $\boldsymbol{Vol}(\mathcal{X}_i^{k_n})$. Here, $\boldsymbol{Vol}(A) := \int_A dx$ is the volume of $A \subset \mathcal{X}$. Again with (9), it is therefore

$$\Sigma_i^{k,n} \propto \left[p(\mathcal{X}_i^{k_n})/\boldsymbol{Vol}(\mathcal{X}_i^{k_n})n\right]^{-\frac{1}{2\alpha+d}}.$$

Subsequently, we can slowly refine the segmentation $\mathcal{X} = \biguplus_{i=1}^{k_n} \mathcal{X}_i^{k_n}$, where $\max_{1 \le i \le k_n} \boldsymbol{Vol}(\mathcal{X}_i^{k_n}) \to 0$ for $k_n \to \infty$ slow enough (with $k_n = o(n)$). Then, for almost every $x \in \mathcal{X}$, there exists a sequence $(i_{k,x})_{k \in \mathbb{N}}$ with $x \in \mathcal{X}_{i_{k,x}}^k$ for all $k \in \mathbb{N}$ such that

$$\Sigma_{i_{k_n,x}}^{k_n,n} = p(x)n(1 + o_p[1]).$$

By construction, it is $\Sigma_{i_{k_n,x}}^{k_n,n} = \Sigma_{\mathrm{GPR}}^n(x)(1 + o_p[1])$. It follows $\Sigma_{\mathrm{GPR}}^n(x) = p(x)n(1 + o_p[1])$ such that $|\Sigma_{\mathrm{GPR}}^n(x)| = [p(x)n]^{\frac{d}{2\alpha+d}}(1 + o_p[1])$. Therefore, asymptotically, $\mathfrak{C}_{\mathrm{GPR}}^n(x) := \left[\frac{1}{p(x)n}\right]^{\frac{d}{2\alpha+d}}|\Sigma_{\mathrm{GPR}}^n(x)|^{-1}$ does not depend on $n$ and $p$. Under homoscedasticity, asymptotically, $\mathfrak{C}_{\mathrm{GPR}}^n$ is necessarily a function that only depends on $f$, which justifies its use as a measure of LFC. $\blacksquare$

## D  Algorithmic summary of the proposed AL framework

---

$$\text{Algorithm 2: } (\Theta_H, \Sigma_E) \leftarrow \text{hyper\_init}(\boldsymbol{X}_{n_0}, \boldsymbol{Y}_{n_0}, p_0, \boldsymbol{X}_{\text{val}}, \boldsymbol{Y}_{\text{val}})$$

---

**Input**

1: Initial training data $(\boldsymbol{X}_{n_0}, \boldsymbol{Y}_{n_0})$
2: Training data density $p_0$
3: A labeled validation set $(\boldsymbol{X}_{\text{val}}, \boldsymbol{Y}_{\text{val}})$

**Output**

4: Initial hyperparameters $\Theta_H = (B, \kappa, \{\sigma_l\}_{l=1}^L, \sigma_G, \lambda_G, \boldsymbol{X}_\dagger^E, \boldsymbol{X}_\dagger^G, \mathfrak{s}_0, \eta_{\mathfrak{s}}, \vartheta_\sigma, \eta_0, \eta_H, \eta_G)$
5: Global (anisotropic) expert bandwidth $\Sigma_E \in \Theta_T$

**Procedure**

6: ▷ Initialize secondary hyperparameters related to computational complexity
7: Identify $\kappa \equiv L$
8: Set $m_E \leftarrow n_0$ and $m_G \leftarrow \frac{n_0}{4}$                ▷ Recall $m_E = \left|\boldsymbol{X}_\dagger^E\right|$, $m_G = \left|\boldsymbol{X}_\dagger^G\right|$ are the number of IPs
9: Draw IP locations $\boldsymbol{X}_\dagger^E, \boldsymbol{X}_\dagger^G \sim p_0$ as described in Appendix E
10: ▷ Tune expert-related hyperparameters
11: Choose $(B, \eta_0, \eta_H)$ as described in Sec. 4.4.3 according to the validation performance of $\mathcal{SVGP}(\theta)$, where $\mu, \lambda, \widehat{v}, \Sigma_E, \boldsymbol{\mu}_\dagger \in \theta$ are learned with respect to $(\boldsymbol{X}_{n_0}, \boldsymbol{Y}_{n_0})$ and $(B, \eta_0, \eta_H)$
12: Set $\Sigma_E \in \Theta_T$, where we choose $\Sigma_E \in \theta$ from the best performing $\mathcal{SVGP}(\theta)$ of the previous step
13: ▷ Initialize secondary hyperparameters related to fine-tuning
14: Set $L \leftarrow 7$ and $\sigma_l \leftarrow 2^{\frac{l-4}{\delta}}$ for $1 \leq l \leq L$ as described in Sec. 4.4.3
15: Set $\mathfrak{s}_0 \leftarrow 0.1, \eta_{\mathfrak{s}} \leftarrow 1/\sqrt{2}$ and $\vartheta_\sigma \leftarrow 0.01$ as described in Sec. 4.4.3
16: ▷ Tune MoE related hyperparameters
17: Choose $(\sigma_G, \lambda_G, \eta_G)$ as described in Sec. 4.4.3 according to the validation performance of $\widehat{f}_{\text{MoE}}$ from (15), where the model parameters $\Theta_T \setminus \{\Sigma_E\}$ from (28) are learned with respect to $(\boldsymbol{X}_{n_0}, \boldsymbol{Y}_{n_0})$ and the model hyperparameters $\Theta_H \setminus \{\sigma_G, \lambda_G, \eta_G\}$ from (27) are fixed
18: Choose $(\vartheta_\sigma, \mathfrak{s}_0, L, \{\sigma_l\}_{l=1}^L)$ as described in Sec. 4.4.3 according to the validation performance of $\widehat{f}_{\text{MoE}}$, where the model parameters $\Theta_T \setminus \{\Sigma_E\}$ from (28) are learned with respect to $(\boldsymbol{X}_{n_0}, \boldsymbol{Y}_{n_0})$ and the model hyperparameters $\Theta_H \setminus \{\vartheta_\sigma, \mathfrak{s}_0, \{\sigma_l\}_{l=1}^L\}$ from (27) are fixed
19: Decrease $\kappa \in \Theta_H$ (beginning from $\kappa = L$) as long as the validation performance of $\widehat{f}_{\text{MoE}}$ does not degrade

---

## E  Finding diverse IP locations

In order to obtain diverse IP locations with a certain distribution, we consider two approaches, *Stein variational gradient descent* (SVGD) (Liu & Wang, 2016; Han & Liu, 2018) and *distributional clustering* (DC) (Krishna et al., 2019).

**Stein Variational Gradient Descent**  SVGD takes a particle swarm and tries to align the empirical distribution of the particles with a target distribution, of which we require the density, as well as its derivative (Liu & Wang, 2016). In addition, the individual particles repel each other, such that we have both diversity and representativeness. In our scenario we have no access to this derivative, such that we resort to the work of Han & Liu (2018) that is solely based on the density. Since the particles move freely in the input space and we have to evaluate the target density a considerable number of times, we suggest applying SVGD, when we deal with well-behaved input spaces and target densities that are easy to evaluate. If the input space is only given through high-dimensional features from a finite set of samples, SVGD might move particles into regions far apart from the data manifold.

**Distributional Clustering**  DC is similar to the known *k-means clustering* (Gan et al., 2020) but solves a different *inertia* objective, that is modified such that asymptotically, as the number of cluster centers $\left|\boldsymbol{X}_\dagger\right| \to \infty$, the distribution of the training data is preserved (Krishna et al., 2019). Under the standard k-means clustering objective, we would observe $\boldsymbol{X}_\dagger \sim p^{\frac{d}{2+d}}$ (Graf & Luschgy, 2007), where it was $\boldsymbol{X}_n \sim p$. Since we intend to use clustering for sub-sampling rather than identifying a fixed number of true cluster centers, we deal with a comparably large number of cluster centers, here. Thus, we will use DC so as to obtain a representative set of IPs. Due to very mild assumptions on the problem, DC is specifically easy to perform in higher dimensions.

**Dealing with local optima of DC**  The inertia objective of DC is given by

$$\text{inertia}_{\text{DC}}[\boldsymbol{c}|\boldsymbol{X}_n] = \sum_{c \in \boldsymbol{c}} \sum_{x \in I_c} \mathbb{1}_{x \neq c} \log \|x - c\|, \tag{42}$$

where

$$I_c = \left\{ x \in \boldsymbol{X}_n \mid \|x - c\| \leq \|x - c'\|, \forall c' \in \boldsymbol{c} \right\} \tag{43}$$

are those elements in $\boldsymbol{X}_n$ that are closest to the center $c$.

In the classical Lloyd-step the centers are updated so as to minimize the intra-cluster inertia, which is given in the case of DC by

$$c^* = \arg\min_{z \in I_c} \sum_{x \in I_c} \mathbb{1}_{x \neq z} \log \|x - z\|. \tag{44}$$

It is a known problem that k-means-related inertia objectives suffer from local optima (Arthur & Vassilvitskii, 2007): The converged solution of cluster centers will typically lie close to their initialization. One way to tackle this issue in practice is to run multiple repetitions of the procedure, followed by choosing the solution with minimal inertia. Unfortunately, the amount of local optima increases with the number of cluster centers. In our case, where we use DC for sub-sampling rather than clustering in its usual sense, we deal with a large number of clusters such that this strategy becomes computationally tedious.

Complementary to running multiple repetitions of k-means, we will extend the state-of-the-art method *k-means++* for choosing the initial set of clusters in a more sophisticated way, where we additionally account for the training distribution. Given the inertia objective

$$\text{inertia}[\boldsymbol{c}|\boldsymbol{X}_n] = \sum_{i=1}^{n} \min_{c \in \boldsymbol{c}} \|x_i - c\|^2$$

of the cluster centers $\boldsymbol{c}$, the *k-means++* procedure builds the set of initial cluster centers as follows: Draw the first center $c_1$ randomly from $\boldsymbol{X}_n$. Then keep track of the current closest squared distance

$$d_i^m = \min_{j \in \{1, \dots, m\}} \|x_i - c_j\|^2 \tag{45}$$

of each element $x_i \in \boldsymbol{X}_n$ to the so far drawn centers $c_1, \dots, c_m$ and sample the next center $c_{m+1}$ with probability $\propto (d_i^m)_{i=1}^n$ from $\boldsymbol{X}_n$. This procedure is repeated until the desired number of cluster centers is reached.

The advantage of k-means++ is that the initial centers are more diverse than if they were sampled at random from $\boldsymbol{X}_n$. However, in its standard form, the centers initialized by k-means++ are themselves distributed flatter than $\boldsymbol{X}_n$. And so, in the case of DC, we propose the following adjustment for a *distributional k-means++*:

We sample with probability $\propto \left(d_i^m p(x_i)^{2/d}\right)_{i=1}^n$ from $\boldsymbol{X}_n$, where $\boldsymbol{X}_n \sim p$.

**Symmetrized DC for molecules**  Since any symmetric molecule has multiple equivalent representations, care must be taken when measuring distances in DC. The key idea is to always compare the two configurations in its closest representation. Using the notation from Appendix H.1, let

$$d(z, z') = \min_{1 \leq s \leq \boldsymbol{s}} \|\Phi(z) - \Phi(\pi_s z')\|$$

be the symmetrized distance between two molecule representations. The symmetrized DC algorithm is then obtained by replacing all occurrences of $\|z - z'\|$ with $d(z, z')$ in the cluster assignments $I_c$, the objective $\text{inertia}_{\text{DC}}[\boldsymbol{c}|\boldsymbol{X}_n]$, the cluster updates $c^*$ and closest distances $d_i^m$ from Equations 43, 42, 44 and 45.

# F  Design choices of the sparse MoE model

In Sec. 4.3 we have made several design choices with computational feasibility in mind. We will discuss these summarized in this section.

**The gate model**  While in Sec. 4.3 we have chosen the gate $g_l \sim \mathcal{SVGP}(\theta_{g_l})$ to be a GPR model, note that any choice of model with sufficient flexibility would have been possible. GPR features universal approximation properties, which makes it a favorable choice.

Furthermore, the gate should come with a small degree of freedom to prevent compared to the experts to prevent those from overfitting during the training of the MoE. For this reason, and the fact that we have no ground truth labels for the training of the gate anyhow, we choose our GPR-based gate to be sparse.

Finally, note that we share the set of gate IP locations $\boldsymbol{X}_\ddagger^G$ across all gate channels. While this is not necessary, it simplifies our method without costs as the MoE is rather insensitive concerning the gate IP locations, as long as these are well-spread.

**The expert models**  While we made clear why we use GPR experts in our work, we left open in Sec. 4.3, whether these experts should be sparse or dense. Here, the deciding factor is the amount of $n$ training samples that we have to deal with: When $n$ goes beyond a few thousand, we suggest switching to sparse GPR experts for computational reasons. Note that after training of the MoE, it is also possible to switch back to full GPR experts, if one aims for a high accuracy predictor. For the purpose of AL, this is not necessary.

Similar to the gate, we share the IP locations $\boldsymbol{X}_\ddagger^E$ across all experts, which simplifies our model. In contrast to the gate situation, the MoE is sensitive to the choice of expert IP locations. Now, if we were to allow individual IP locations for each expert, an elsewise locally underperforming expert might work better than the remaining experts due to a lucky choice of its individual IPs. Subsequently, this would result in a sub-optimal gate and, hence, ultimately in a wrong *superior training density* estimate.

For better generalization, if our MoE model comprises dense GPR experts, we will either have to rely on individual training sets for the experts and the gate, or we use *leave-one-out* expert responses on a shared training set.

In the sparse expert case, it is necessary to learn reasonable inducing values $\boldsymbol{\mu}_\ddagger$ prior to the actual learning procedure of the MoE to not get stuck in a spurious solution. Therefore, there should be a short pre-training phase for each individual expert.

In addition—whether or not the experts are sparse—the shared expert parameters $\mu_E, \lambda_E, \widehat{v}, \Sigma_E$ should be initialized reasonably. In this regard, we suggest training a single, global expert model before the (pre-)training of the actual experts to obtain those initial parameter estimates for which we have no prior knowledge. If we assume isotropic bandwidths to be sufficient, we can simply set $\Sigma_E = \sigma_E \mathcal{I}_d$ and learn the scalar $\sigma_E > 0$ instead. Note that, from practice, the training of the MoE suffers tremendously from online changes of the expert bandwidths. Thus, we suggest to keep $\Sigma_E$ fixed after initialization.

Finally, note that, if we stick with sparse experts after training of the MoE, it can be beneficial for the prediction accuracy to re-train the MoE, where we keep the gate fixed. In this post-processing step, we would like to apply larger learning rates on the experts to escape local optima. However, larger learning rates also lead to underperforming intermediate steps, in which an actively trained gate might reject the best fitting expert at random—therefore pushing the gate towards a local optimum. Keeping the pre-trained gate fixed at this point prevents this undesired behavior.

**The IPs**  Recall that we have set the covariance $\boldsymbol{S}_\ddagger = 0$ of the inducing value distribution to zero, whereas it could have also been a diagonal or positive definite matrix. Playing around with this parameter, we have seen no significant improvement that would justify the considerable amount of additional model parameters from a computational point-of-view.

In our approach we suggest keeping the IP locations fixed, which is also for reasons of computational feasibility, but, more importantly, adaptive IP locations come along with heavy prediction instabilities during the training.

We found it necessary and sufficient to initialize the IP locations by state-of-the-art methods, as described in Appendix E.

**The MoE objective**  For the training of our MoE in Sec. 4.4.1, we added a penalty on small bandwidth choices. As described in (Lepski, 1991; Lepski & Spokoiny, 1997), the optimal bandwidth choice is the largest

one that is capable of modeling the function. Now that we are able to model a comparably flat function by small bandwidths, as long as we have got enough training support, it can occur that, with no regularization, we choose a too-small bandwidth for such a flat region. A too-small bandwidth choice might cause overfitting. But even worse, in the subsequent AL loop the flat region is falsely identified as complex, leading to more training queries in this location, which then allow for even smaller bandwidths to model this flat region. We will demonstrate this pathological behavior for the unregularized case on toy-data in Sec. 5.1.

**The gate noise 𝔰**    Like already mentioned in Sec. 4.4.1, it is possible to tune 𝔰 in the training process:

**Remark 8.** *Shazeer et al. (2017) proposed to learn the 𝔰 parameter by adding a penalty term to the main objective that penalizes the imbalance of how likely training inputs are assigned to each expert: Let $\pi_b \in [0,1]^L$ be the expert assignment probabilities of $x_b$ and define $\pi_{\mathcal{B}} = \sum_{b \in \mathcal{B}} \pi_b$. Then they add a penalty $\mathbb{V}[\pi_{\mathcal{B}}]/[\mathbb{E}\,\pi_{\mathcal{B}}]^2$ to the objective, which is the* squared coefficient of variation—*a coefficient that accounts for the non-uniformity of a set of positive variables.*

We justify our simple heuristic to shrink 𝔰 in a static way as follows: Recall from Sec. 4.3 that 𝔰 prevents premature commitment to a spurious solution. When treating 𝔰 as a trainable parameter, it does not decay towards zero. Maintaining the noise then prevents the locally best-performing experts from converging by randomly withholding training samples. For this reason, we find that 𝔰 behaves best when decaying towards zero as the training progresses.

## G    Supplemental results on the Doppler experiment

### G.1    The single-scale GPR model

When training a single-scale GPR model on the Doppler dataset, the tuned bandwidth parameter will typically take an intermediate value, trying to compromise between more complex and simpler regions. This is reflected in the predictions in Fig. 16, where the single-scale GPR model suffers from the inhomogeneous structure, underfitting the complex region to the left while simultaneously overfitting the simple region to the right.

In Fig. 17 we compare the performance of our multi-scale MoE approach to the single-scale GPR model. The consistently inferior performance of the single-scale GPR model shows that the issue above persists even for large training sizes.

### G.2    Necessity of the small bandwidth penalty

In Appendix F we discuss overfitting issues with too small local bandwidth estimates as a consequence of inadequate regularization of LOB. To address this issue, we have proposed to penalize such small bandwidth choices by $\vartheta_\sigma \mathrm{pen}_\sigma(\boldsymbol{X}_n, \boldsymbol{Y}_n, \mathcal{B}, w, \Theta)$ with the penalty term $\mathrm{pen}_\sigma$ from (25) and a scaling factor $\vartheta_\sigma \geq 0$.

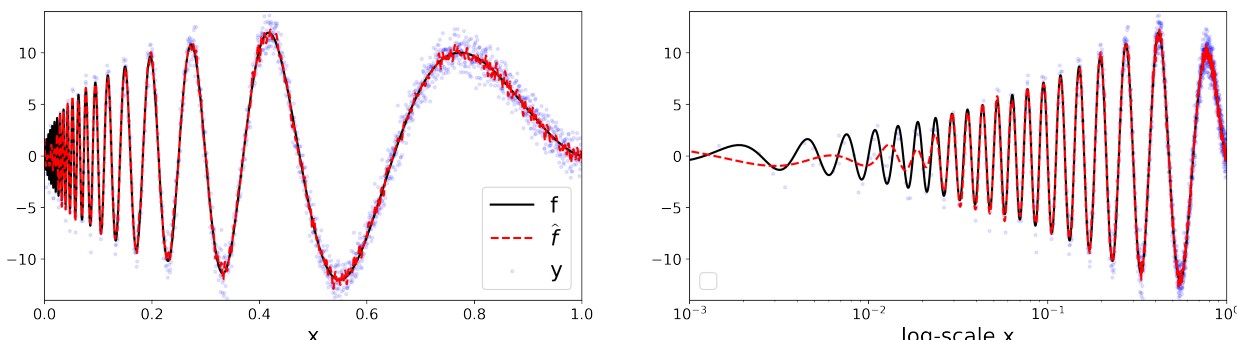

Figure 16: The Doppler experiment: An exemplary dataset and the predictions of a global GPR model, shown on natural x-scale (left) and on logarithmic x-scale (right).

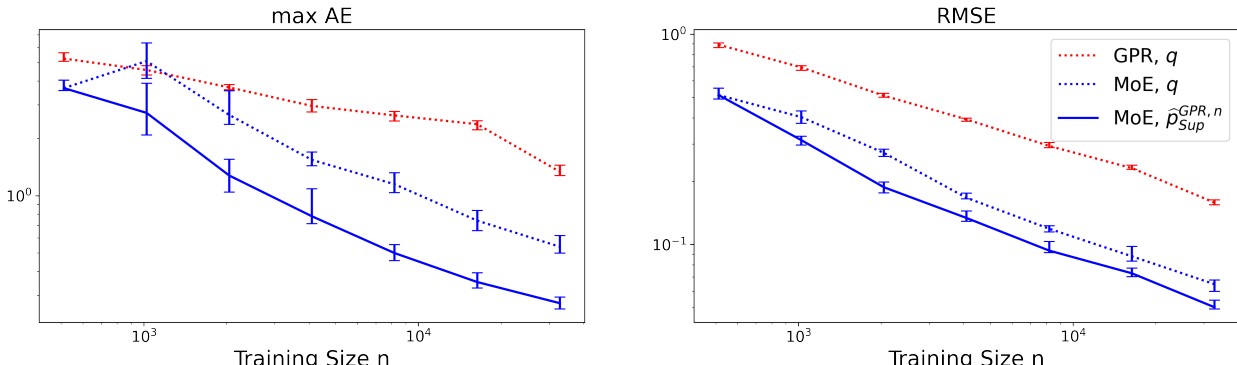

Figure 17: The Doppler experiment: The max AE (left) and the RMSE (right) of our proposed MoE model in comparison to a single-scale GPR model. The results are averaged over 20 repetitions.

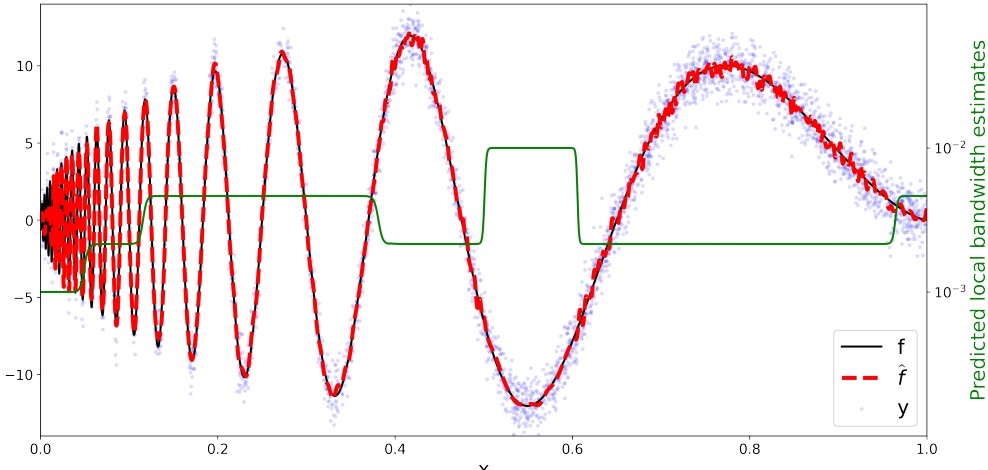

Figure 18: The Doppler experiment: An actively sampled dataset (top) with our MoE fit at $n = 2^{12}$ training samples without small bandwidth penalty ($\vartheta_\sigma = 0$), and the associated LOB estimate (bottom).

Now, while the LOB estimate with $\vartheta_\sigma = 0.5$ (see Fig. 4) consistently behaves as expected, we show for comparison a typical LOB estimate in Fig. 18 that results from applying no regularization ($\vartheta_\sigma = 0$). By chance—here, the flat region of the Doppler function to the right—the trained model suffers from massive overfitting by too small LOB estimates. These falsely obtained small LOB estimates then lead to overestimation of LFC, which subsequently results in a detrimental oversampling of these locations by the AL procedure.

# H Supplemental on the malonaldehyde MD simulation experiment

## H.1 The sGDML model

The GDML model by Chmiela et al. (2017) represents the geometry $x = [R_1, \ldots, R_{\boldsymbol{a}}] \in \mathbb{R}^{3 \times \boldsymbol{a}}$ of each molecule in terms of the reciprocal distances $\Phi(x)_{kl} = \|R_k - R_l\|^{-1}$ of all atom-pairings to achieve roto-translational invariance of the input. This representation gives us a total $\boldsymbol{d} = \boldsymbol{a}(\boldsymbol{a} - 1)/2$ input features. The similarity of

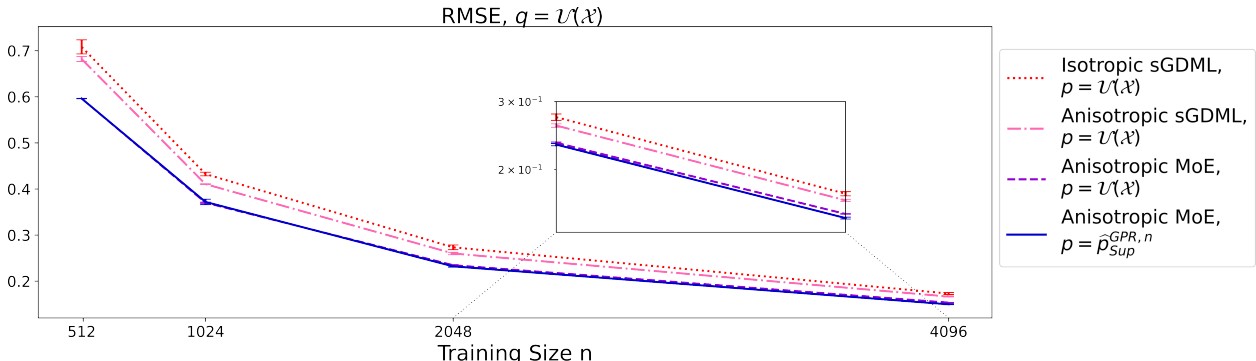

Figure 19: The RMSE under the uniform test distribution for different variants of sGDML and training distribution at varying training size: The performance is given for passive sampling, using the original isotropic sGDML (dotted), anisotropic sGDML (dash-dotted) and our MoE model with anisotropic sGDML experts (dashed), and for the MoE model, applying the proposed *superior sampling scheme* (solid). The results are averaged over 2 repetitions.

a pair of configurations $(z, E, F)$ and $(z', E', F')$ is then given by the extended covariance function

$$\boldsymbol{Cov}(E, E') = k(\Phi(z), \Phi(z')),$$
$$\boldsymbol{Cov}(E, F') = \frac{dk(\Phi(z), \Phi(z'))}{d\Phi'} \frac{d\Phi(z')}{dx},$$
$$\boldsymbol{Cov}(F, F') = \left[\frac{d\Phi(z)}{dx}\right]^{\top} \frac{dk(\Phi(z), \Phi(z'))}{d\Phi d\Phi'} \frac{d\Phi(z')}{dx}.$$

Hence, we denote the overall kernel function of two configurations by

$$\boldsymbol{k}(z, z') = \boldsymbol{Cov}((E, F), (E', F')) \in \mathbb{R}^{(3\boldsymbol{a}+1) \times (3\boldsymbol{a}+1)}.$$

Atoms of the same type are physically identical and therefore exchangeable, albeit only a small subset of such symmetries is exercised at a given (low) MD simulation temperature. Full permutational invariance is only needed when enough energy is put into the system for all bonds to break and all atoms to disassociate.

The symmetric extension sGDML (Chmiela et al., 2018; 2019) automatically identifies all accessed atom permutations from the training set and adds this symmetric prior to the covariance function. Formally, let $(\pi_s)_{s=1}^{\boldsymbol{s}}$ be atomic permutations that lead to an equivalent molecular representation. Then, the extended symmetric kernel of sGDML is given by

$$\widetilde{\boldsymbol{k}}(z, z') = \sum_{s=1}^{\boldsymbol{s}} \sum_{t=1}^{\boldsymbol{s}} \boldsymbol{k}(\pi_s z, \pi_t z'). \tag{46}$$

Malonaldehyde possesses $\boldsymbol{s} = 4$ such permutations.

**Remark 9.** *The identified set of permutations is transitively closed to form a group. Under isotropy, it suffices to permute only one of the two configurations given to the kernel: Permuting both entries (as in (46)) equals permuting one entry and multiplying by the constant $\boldsymbol{s}$. However, if the applied bandwidth is not of the form $\Sigma = \sigma \mathcal{I}_{\boldsymbol{d}}$, this property does not hold.*

### H.2 The malonaldehyde MD simulation experiment under a uniform test distribution

In this scenario, we assume a uniform test density $q = \mathcal{U}(\mathcal{X})$. Accordingly, we weight the validation and test MSE by the importance weights $1/\widehat{p}_{\mathcal{X}}(\boldsymbol{X}_{\text{val}})$ and $1/\widehat{p}_{\mathcal{X}}(\boldsymbol{X}_{\text{T}})$. We draw the initial expert training set $\boldsymbol{X}_n$ of size $n = 2^9$ and the gate training set $\boldsymbol{X}_{n_G}^G$ via importance sampling from the remaining pool with weights $1/\widehat{p}_{\mathcal{X}}(\boldsymbol{X}_{pool} \setminus (\boldsymbol{X}_{\text{val}} \cup \boldsymbol{X}_{\text{T}}))$. By this it is $\boldsymbol{X}_n \sim \mathcal{U}(\mathcal{X})$.

In Fig. 20 we show the estimates of LOB, LFC, and the *superior training density* under the pool test distribution, evaluated on the relaxed configurations of malonaldehyde. The LFC estimates in Fig. 20 confirm

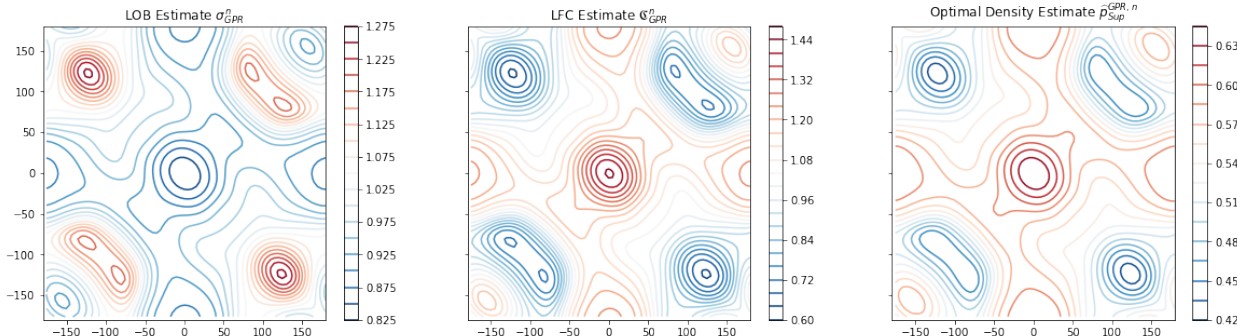

Figure 20: Estimates of LOB (left), LFC (middle) and the *superior training density* (right) under the pool test distribution $q = \mathcal{U}(\mathcal{X})$, evaluated at the relaxed malonaldehyde configurations, plotted with respect to the angles of the two aldehyde rotors of malonaldehyde.

our expectation that the transition areas are more complex to model than the regions near the stable configurations. Subsequently, our active sampling scheme shifts sample mass away from the stable regimes in favor of the transition areas.

We have plotted the error curves of passive and active sampling schemes in Fig. 19. When estimating the relative sample size (31) that we require to achieve the same RMSE via active sampling compared to *random test sampling*, we obtain $\varrho(\widehat{f}_{\mathrm{MoE}}, \widehat{p}_{\mathrm{Sup}}^{\mathrm{GPR},n}) = 0.965 \pm 0.009$. This means that we save about $3.5\%$ of samples with our active sampling scheme. With similar calculations, we save about $27\%$, when comparing the original sGDML approach with passive sampling to our MoE model with active sampling.

## H.3 A realistic MD simulation AL scenario

In the realistic ab initio FF reconstruction AL scenario, we begin by sampling the initial training set $(\boldsymbol{X}_{n_0}, \boldsymbol{Y}_{n_0})$ as well as the validation set $(\boldsymbol{X}_{\mathrm{val}}, \boldsymbol{Y}_{\mathrm{val}})$ by simulating the true MD trajectory solving the computationally expensive Schrödinger equation. Estimate the initial MD density $p_{\mathcal{X},0}$ based on $(\boldsymbol{X}_{n_0} \cup \boldsymbol{X}_{\mathrm{val}})$.

For $k \in \mathbb{N}_0$ :

- Set $q^k \leftarrow p_{\mathcal{X},k}^{1-\frac{1}{k+1}}$ to encourage exploration in early iterations and exploitation in later iterations

- Estimate the model $\widehat{f}_k := \widehat{f}_{\mathrm{MoE}}$ based on $(\boldsymbol{X}_{n_k}, \boldsymbol{Y}_{n_k})$

- Estimate $p_{\mathrm{Sup}}^{\mathrm{GPR},n_k}$ based on $q^k$ and $\widehat{f}_k$

- Sample a large pool $(\boldsymbol{X}_{pool,k+1}, \widehat{\boldsymbol{Y}}_{pool,k+1})$ of, e.g., 100,000 candidates by simulating the approximate MD trajectory using the computationally cheap model $\widehat{f}_k$. While simulation, avoid unreliable out-of-distribution predictions, e.g., by resetting the trajectory in some $x$ whenever $p_{\mathcal{X},k}(x) < \epsilon$ drops below a reasonable threshold.

- Estimate the trajectory density $p_{\mathcal{X},k+1}$ of $\boldsymbol{X}_{pool,k+1}$

- Update the training set $(\boldsymbol{X}_{n_{k+1}}, \boldsymbol{Y}_{n_{k+1}})$ by selecting input candidates from the pool $\boldsymbol{X}_{pool,k+1}$ with distribution $p_{\mathrm{Sup}}^{\mathrm{GPR},n_k}$ and estimating the respective labels solving the Schrödinger equation

