# OpenReview forum: "Local Function Complexity for Active Learning via Mixture of Gaussian Processes"
_TMLR — Accepted by TMLR_

### Review · Reviewer_WPPN · 2023-09-19

**Summary Of Contributions:**

This paper deals with selecting training data in a more efficient way than random sampling. It is based on a sequential active learning strategy where a surrogate model is used to approximate the underlying unknown function while new points are added using a criterion based on local function complexity. The surrogate is a mixture of sparse Gaussian process (GP) regression models. As for the criterion, it relies on the derivation of proxy for the local function complexity. A 1d toy example is presented, then a 32d real world one.

**Audience:**

Yes

**Claims And Evidence:**

Yes

**Requested Changes:**

(critical) *Clarity*. To help enhance the clarity of the paper, it may be worth separating the sparse GP aspect from the main subject. Most derivations are presented for isotropic kernels, but then are used on the anisotropic sGDML model.
Then, the considered problem is only formally presented on page six.

(critical) On the selection of inducing points selection, the state of the art to compare with can be found in more recent articles, say starting from Moss, H. B., Ober, S. W., & Picheny, V. (2023, April). Inducing point allocation for sparse gaussian processes in high-throughput bayesian optimisation. In International Conference on Artificial Intelligence and Statistics (pp. 5213-5230). PMLR.

(critical) The modeling of non-stationary functions is a recurrent topic with GPs, hence the use of mixtures and sparse inputs is only one option among several, see e.g.,
- Gramacy, R. B., & Lee, H. K. H. (2008). Bayesian treed Gaussian process models with an application to computer modeling. Journal of the American Statistical Association, 103(483), 1119-1130.
- Gramacy, R. B., & Apley, D. W. (2015). Local Gaussian process approximation for large computer experiments. Journal of Computational and Graphical Statistics, 24(2), 561-578.
- Roininen, L., Girolami, M., Lasanen, S., & Markkanen, M. (2018). Hyperpriors for Matérn fields with applications in Bayesian inversion. Inverse Problems and Imaging, 13(1), 1-29.
- Rullière, D., Durrande, N., Bachoc, F., & Chevalier, C. (2018). Nested Kriging predictions for datasets with a large number of observations. Statistics and Computing, 28, 849-867.
- Cutajar, K., Bonilla, E. V., Michiardi, P., & Filippone, M. (2017, July). Random feature expansions for deep Gaussian processes. In International Conference on Machine Learning (pp. 884-893). PMLR.

When using these methods, the typical criteria would not be the Gaussian process uncertainty but the integrated mean squared prediction error, or related criteria. See for instance Sauer, A., Gramacy, R. B., & Higdon, D. (2023). Active learning for deep Gaussian process surrogates. Technometrics, 65(1), 4-18.
P19: why not sampling where the predictive variance is the largest?

(critical) A discussion on the poor scaling of GPs with the dimension is needed.


Some additional related works may include:
- Lederer, A., Umlauft, J., & Hirche, S. (2019). Uniform error bounds for gaussian process regression with application to safe control. Advances in Neural Information Processing Systems, 32.
- Kanagawa, M., Hennig, P., Sejdinovic, D., & Sriperumbudur, B. K. (2018). Gaussian processes and kernel methods: A review on connections and equivalences. arXiv preprint arXiv:1807.02582.
- Mak, S., & Joseph, V. R. (2018). Support points. The Annals of Statistics, 46(6A), 2562-2592.

The regularity of $f$ is discussed several times, which for GPs is typically handled with Matérn kernels.

The link between (4) and (5) with the introduction of Q is, in my opinion, not straightforward.

(17) is said to not depend on p, v and n while it appears in (17).

A full pseudo code of the approach would be appreciated, rather than Figures.

Figs. 6 and 7: perhaps use consistent x-axis

Fig. 12: Perhaps use letters rather than positions

A moderate toy dimensional example would be welcomed, to see the scaling in 2 or 3d when strong baselines can still be used.

**Strengths And Weaknesses:**

Strengths:
- the expression for the local function complexity of a GP is derived.
- the paper combines ideas from several different fields (local polynomial smoothing, Gaussian processes, active learning, etc)

Weaknesses:
- the paper is long and sometimes hard to follow.
- the main theoretical paper inspiring this work is a preprint.
- the state of the art on GP models is relatively outdated.
- GP are well known for scaling poorly with the dimension, while this is the reason to use them here.
- the baseline for the experimental part, uncertainty sampling, is weak.

---

> ### Author Response · Authors · 2023-10-23
> **Response to Reviewer WPPN (part 1)**
>
> Dear reviewer, thank you for your suggestions to further improve our manuscript. We addressed your concerns as listed below and uploaded a revised manuscript, where the essential changes are highlighted with blue text color.
>
> # Requested Changes
> * (critical) Clarity. To help enhance the clarity of the paper, it may be worth separating the sparse GP aspect from the main subject. Most derivations are presented for isotropic kernels, but then are used on the anisotropic sGDML model. Then, the considered problem is only formally presented on page six.
>
> **Response**
> When deploying a non-variational GP as the gate of our MoE, we would require pseudo-labels (built from the expert likelihoods) and an expectation maximization loop for its training (see Sec. 2). Our deployed variational GP model for the gate is an elegant way around this problem, which is inherently sparse. From this point of view, we assumed a consequent use of this model to be the most concise way to write down our work.
>
> The derivations of LFC and superior training are given for general (e.g., anisotropic) bandwidths in Sec. 4.1 and 4.2. In practice, we then have limited the anisotropic treatment through a shared positive bandwidth matrix $\Sigma_E$ (see Sec. 4.3), which is tuned in a pre-processing step before setting up the MoE (see Sec. 4.4.3). The MoE is constituted of experts of individual bandwidths $\sigma_l\Sigma_E$ such that the locality of the GP only varies through an isotropic factor throughout our work. With the pseudo-code of our proposed \AL framework in the revised manuscript, the role of $\Sigma_E$ should become clearer.
>
> * (critical) On the selection of inducing points, the state of the art to compare with can be found in more recent articles, say starting from Moss, H. B., Ober, S. W., \& Picheny, V. (2023, April). Inducing point allocation for sparse gaussian processes in high-throughput bayesian optimisation. In International Conference on Artificial Intelligence and Statistics (pp. 5213-5230). PMLR.
>
> **Response**
> Thank you for pointing us to this related work. While the authors state that any quality function for incorporating informativeness into the inducing point selection process, they exercise a quality function proportional to the label, which is informative in their Bayesian optimization setting rather than regression.
> Note that LFC would be a possible candidate for their quality function in the regression setting. In this regard, this related work is rather complementary to our work than a state-of-the-art competitor.
> We discuss this related work in Sec. 2 and add the idea of combining this approach with our LFC estimate as future work in Sec. 7.
>
> * (critical) The modeling of non-stationary functions is a recurrent topic with GPs, hence the use of mixtures and sparse inputs is only one option among several, see e.g., ...
>
> **Response**
> Thank you for suggesting this very relevant related work. We have added a discussion on alternative nonstationary GP approaches in Sec. 2. Furthermore, we add a small experiment on the deep GP approach by (Sauer, A., Gramacy, R. B., \& Higdon, D. (2023)) to Sec. 5.1, using their easily available 'deepgp' CRAN package.
>
> * When using these methods, the typical criteria would not be the Gaussian process uncertainty but the integrated mean squared prediction error, or related criteria. See for instance Sauer, A., Gramacy, R. B., \& Higdon, D. (2023). Active learning for deep Gaussian process surrogates. Technometrics, 65(1), 4-18. P19: why not sampling where the predictive variance is the largest?
>
> **Response**
> As hinted above, we add this deep GP model and its associated active learning scheme to our toy experiment in Sec. 5.1. Here, we will combine our proposed model-agnostic superior training data process with the deep GP model. As expected, the performance of our active learning framework lies between the passive design and the active learning scheme that is tailored to this deep GP model. Here, the ability of our framework to sample batches of arbitrary size without model reestimation becomes particularly advantageous at large sample sizes compared to the pointwise active learning design of the deep GP model.

---

> > ### Author Response · Authors · 2023-10-23
> > **Response to Reviewer WPPN (part 2)**
> >
> > * (critical) A discussion on the poor scaling of GPs with the dimension is needed.
> >
> > **Response**
> > As opposed to nonstationary GP approaches (e.g., the tree-based or local GP by (Gramacy et.~al 2008) or (Gramacy et.~al 2015)) that suffer from the curse of dimensionality through input space localization, we segment the input space into a fixed number $L$ of patches, given by the $L$ experts of our MoE. If we were to instantiate our MoE with full GP models (inducing points equal to training points), our approach would not scale poorly concerning the input space dimension $d$.
> >
> > Suppose the reviewer's perspective is however from the large training size point-of-view, where we require sparse GP formulations. In that case, we admit that there is poor scaling in the number of inducing points in $d$ as well as a decaying power of nonparametric active learning over passive learning. In our work, we assume a low intrinsic dimension $\delta < d$ of the input space, which we did not discuss as a limitation of our work. We add this limitation to the discussion (Sec. 6) in the revised manuscript.
> >
> > * The regularity of f is discussed several times, which for GPs is typically handled with Matérn kernels.
> >
> > **Response**
> > We will gladly incorporate this suggestion in our future work and add a comment in the discussion in Sec. 6.
> >
> > * The link between (4) and (5) with the introduction of Q is, in my opinion, not straightforward.
> >
> > **Response**
> > We improved the sloppy notation between (4) and (5) to be mathematically rigorous in the revised manuscript.
> >
> > * (17) is said to not depend on p, v and n while it appears in (17).
> >
> > **Response**
> > This is intended: The LOB of GPR scales implicitly with the training density $p$ as well as the training size $n$. This implicit scaling is asymptotically canceled out by the prefactor that contains $p$ and $n$ such that ultimately 'prefactor times LOB' does not depend on $p$ and $n$.
> >
> > * A full pseudo code of the approach would be appreciated, rather than Figures.
> >
> > **Response**
> > We added the requested pseudo-code in the revised manuscript in Appendix D.
> >
> > * Figs. 6 and 7: perhaps use consistent x-axis, Fig. 12: Perhaps use letters rather than positions
> >
> > **Response**
> > OK
> >
> > * A moderate toy dimensional example would be welcomed, to see the scaling in 2 or 3d when strong baselines can still be used.
> >
> > **Response**
> > We apologize for not being able to provide this add-on in the short time window.

---

> > > ### Comment · Reviewer_WPPN · 2023-10-24
> > >
> > > I would to thank the authors for their detailed point by point response. Most of my comments have been appropriately addressed and I just have a few remaining comments:
> > >
> > > On the scaling, perhaps the discussion should be better compartmentalized between: scaling in terms of $n$ (training size) and $d$ (input dimension) both in terms of prediction accuracy (especially for $d$) as well as computational efficiency. This is not completely clear at this point.
> > >
> > > The full pseudo-code is helpful. It should be more detailed, with reference to equations and/or discussion in the text as they are missing on some lines (say, lines 28, 32, 34 for Algorithm 1, or lines 14-15 in Algorithm 2, etc).

---

> > > > ### Author Response · Authors · 2023-10-26
> > > > **Response to Reviewer WPPN**
> > > >
> > > > Dear reviewer,
> > > >
> > > > we added more details to the pseudo-code, including the suggested lines. On request of another reviewer, we moved the main Algorithm 1 to the main part of the paper in Sec. 4.

---

### Review · Reviewer_ZcVs · 2023-09-29

**Summary Of Contributions:**

The paper present as its main contribution an active learning framework for GPR-based methods.

**Audience:**

Yes

**Claims And Evidence:**

Yes

**Requested Changes:**

address the weaknesses

**Strengths And Weaknesses:**

Strengths:

* The authors put forward a principled approach for the important problem of choosing optimal training sets. Active learning techniques are needed in many important fields such as multi-task learning or federated learning.

* Extensive numerical experiments illustrate the proposed methods.

Weaknesses:

* It would be helpful to illustrate the key concepts in the most simple ML setting. In particular it should be made more explicit how the different concepts such as LFC correspond to the key components of ML applications, i.e., is LFC a property of a data (distribution) or a specific hypothesis (target function) or a  model (hypothesis space) or a combination of model/loss function ( see Ch. 2 of [Ref1] for a discussion of these three main components).

* avoid the excessive use of the word "framework". try to use more specific words if you mean, e.g., an algorithm or a probabilistic model or a "complexity measure" for a model.

* The main contribution should be formulated as an algorithm pseudo-code and not only as Figure 2 . This algorithm description must clearly indicate the required input data, hyper-parameter and delivered output. You must then also clearly explain how you applied that algorithm in your numerical experiments.

* There should be more discussion about the (sub-) optimality of your active learning methods. How close is your method to fundamental limits (see e.g. Sec. 6.4 of [Ref1] for fundamental lower bounds on the sample size required by plain old linear regression)

* The use of language could be sharpened (avoiding jargon), e.g., "The scalability of this estimate makes...", "such that the optimal training inputs in Eq. (3) can asymptotically be obtained..", "Let us consider a family ...of kernel machines..", "...of the hypothetically optimal training set of a model in the asymptotic limit of the sample size."

[Ref1] A. Jung, "Machine Learning: The Basics," Springer, 2022.

---

> ### Author Response · Authors · 2023-10-23
> **Response to Reviewer ZcVs**
>
> Dear reviewer, thank you for your suggestions to further improve our manuscript. We addressed your concerns as listed below and uploaded a revised manuscript, where the essential changes are highlighted with blue text color.
>
> # Weaknesses
> * It would be helpful to illustrate the key concepts in the most simple ML setting. In particular it should be made more explicit how the different concepts such as LFC correspond to the key components of ML applications, i.e., is LFC a property of a data (distribution) or a specific hypothesis (target function) or a model (hypothesis space) or a combination of model/loss function (see Ch. 2 of (A. Jung, "Machine Learning: The Basics," Springer, 2022.) for a discussion of these three main components).
>
> **Response**
> LFC and the superior training density are both properties of the combination of model/loss function, where the hypothesis space consists of "locally adaptive models" and the loss function is the conditional mean squared error for LFC and the mean integrated squared error for the superior training density. We added a characterization in Sec. 6 in the revised manuscript.
>
> * avoid the excessive use of the word "framework". try to use more specific words if you mean, e.g., an algorithm or a probabilistic model or a "complexity measure" for a model.
>
> **Response**
> We tried to be more precise about this in the revised manuscript.
>
> * The main contribution should be formulated as an algorithm pseudo-code and not only as Figure 2. This algorithm description must clearly indicate the required input data, hyper-parameter and delivered output. You must then also clearly explain how you applied that algorithm in your numerical experiments.
>
> **Response**
> We added the requested pseudo-code in the revised manuscript in Appendix D.
>
> * There should be more discussion about the (sub-) optimality of your active learning methods. How close is your method to fundamental limits (see e.g. Sec. 6.4 of (A. Jung, "Machine Learning: The Basics," Springer, 2022.) for fundamental lower bounds on the sample size required by plain old linear regression)
>
> **Response**
> Unfortunately, we do not understand the idea the reviewer has in mind. After inspection of Sec. 6.4 of (A. Jung, "Machine Learning: The Basics," Springer, 2022.), it appears to us that this section discusses model complexity rather than sample size. In particular, we see no fundamental lower bounds on the required sample size of linear regression. What we understand as fundamental limits of nonparametric active learning is discussed in (László Györfi, Michael Kohler, Adam Krzyżak, and Harro Walk. A distribution-free theory of nonparametric
> regression, volume 1. Springer, 2002.), which we mention in Sec 4.1 of our manuscript. If this bound does not suffice, we would like to ask the reviewer to elaborate more on their idea.
>
> * The use of language could be sharpened (avoiding jargon), e.g., "The scalability of this estimate makes...", "such that the optimal training inputs in Eq. (3) can asymptotically be obtained..", "Let us consider a family ...of kernel machines..", "...of the hypothetically optimal training set of a model in the asymptotic limit of the sample size."
>
> **Response**
> We tried to improve our writing in the revised manuscript.

---

> > ### Comment · Reviewer_ZcVs · 2023-10-23
> > **response**
> >
> > The authors have tried to address all my concerns. However, I still believe that the main contribution should be formulated as an algorithm (pseudocode) in the main part of the paper.
> >
> > I am still missing a comparison of the empirical performance of the proposed method with fundamental lower bounds on the RMSE (e.g. in Fig. 6 and 8).

---

> > > ### Author Response · Authors · 2023-10-23
> > > **Response to Reviewer ZcVs**
> > >
> > > Dear reviewer,
> > >
> > > thank you for your fast response.
> > >
> > > Regarding the pseudo-code, we are fine with moving it from Appendix D to the main part of the paper.
> > >
> > > Regarding the request for a fundamental lower bound on the RMSE in the toy experiment, we are still clueless. Maybe there is a misunderstanding: While we are training (and validating) on noisy labels, we evaluate prediction performance against the ground truth function f. So the fundamental lower bound on the RMSE, here, is given by 0. (If we were to test on noisy data, this bound would be 1.) If this is what the reviewer meant, we will clarify in the experimental description that we test against true f (which is currently not stated explicitly).

---

### Review · Reviewer_kvMm · 2023-10-09

**Summary Of Contributions:**

The present work introduces an active learning framework with a mixture of expert Gaussian Processes (GP) trained variational. The proposed AL strategy follows the steps of Pankin et al . 2021, formulated for local polynomial smoothing (LPS).
First, the authors adapt the former work’s framework to GPR. They give an expression for the asymptotic local function complexity under the GPR framework. They deduce from it an expression for a superior training density to be used in active learning.

To be implemented in practice the scheme requires knowing the test distribution, which is not usually the case, instead they rely on an unnormalised estimate and using an importance sampling estimate of the density. The assumption is that one has access to a large amount of unlabelled points that form a pool from which the dataset will be constructed. One the optimal training density has been derived, self-normalised importance sampling is used to estimate the probability of each point in the pool.

The authors specify their parametrisation of the sparse mixture of Gaussian process as a mixture of experts of fixed logarithmically space bandwidths. After training they can also propose an estimation of the locally optimal bandwidth as a weighted geometric average of the experts’ bandwidths, which in turn can be used to fix the sparsity inducing parameter of the model.

The authors offer a discussion on their hyperparameter choices, and their way of choosing how to initialise the IPs in a diverse way to minimise the distance to an evaluation point x.

Experiments:

In a 1d Doppler effect experiment, the authors show marginal improvement over the LPS, even with optimal training density and clear improvement over the random test sampling for all competing methods. They also use the simple example to test their model under heteroscedastic noise.
Next the authors turn to a force field reconstruction problem for a 9 atoms molecules and find improvements brought by their methods attributed both to the mixture of experts model and the superior training density over common symmetric gradient-domain machine learning.

**Audience:**

Yes

**Claims And Evidence:**

Yes

**Requested Changes:**

I have the following questions:

1) I do not understand the formulations “top-down" and “bottom up" in the following sentences:
“they then propose to sample training data in a top-down manner from this very distribution, “middle of page 2
“which refine training data bottom-up in a greedy way” bottom of page 4

2) p7 can read: “As already noted in the introduction, the results to LFC and the optimal training density of LPS indicate their problem intrinsic nature, as they reflect no direct dependence on the LPS model.’’  — Except for the order Q correct?

3) Adding more precision on how the training objective is derived would be helpful. The $q_l$ is the marginal distribution of the $l$-th component of the MoE ? (Section 4.4.1) It is not straightforward to understand also the definition of the $P_{b,l}$. How should one interpret the marginalised $f$? In the definition of $w \propto q/p$ the appearing $q$ is the test distribution and the appearing $p$?

4) Could a simpler model for the gate functions be employed? With which loss on performance?

5) Could the authors clarify how to adapt their work to “on the fly" sampling, that is when there is not a $X_{\rm pool}$ available from the start? That is crucial for a realistic implementation in ab initio force field as producing the unlabelled data (the configurations) as a pool a priori is requires to compute the forces to run the MD simulation and is therefore not cheaper than getting also the labels.

6) It is arguably not fair to say that "current ML-based FFs typically rely on rather naive exhaustive sampling schemes to gather training data" page 22,  cf for instance "Section 3.3.3. Adaptive Sampling" of the cited review by Keith et al 2022.

7) The interpretability of LFC (discussion section) for high-dimensional input spaces should be relativized, as one cannot visualise this function expect in low-dimension, or using relevant, previously known, low-dimensional projections. Or maybe the authors want to point out that one can locate high LFC points from the pool? In which case the precision would be useful.

Minor comments:
- The acronym MISE is not introduced p2.
- From an editing point of view, keeping figures to the top of pages would improve readability by avoiding lost sentences/paragraph title between figures.

**Strengths And Weaknesses:**

Strengths:
- Related works discussion appears satisfactory.
- Presentation is relatively clear, although some reptition could be avoided to improve readability.
- A toy experiments is treated in-depth to illustrate the method.
- A real world experiment is also show-cased.

Weaknesses:
- The method is rather involved. I am too far from the field to asses if all the choices are necessary: (cf question below, is a MoE of GP for the gate functions also necessary?)

---

> ### Author Response · Authors · 2023-10-23
> **Response to Reviewer kvMm**
>
> Dear reviewer, thank you for your suggestions to further improve our manuscript. We addressed your concerns as listed below and uploaded a revised manuscript, where the essential changes are highlighted with blue text color.
>
> # Weaknesses
> * The method is rather involved. I am too far from the field to assess if all the choices are necessary: (cf question below, is a MoE of GP for the gate functions also necessary?)
>
> **Response**
> The overall model is a MoE, whereas its gate is a vector of length L (= number-of-experts) of classic GPs. We will revisit the description for potential ambiguities.
>
> # Questions
> * I do not understand the formulations “top-down" and “bottom up" in the following sentences: “they then propose to sample training data in a top-down manner from this very distribution, “middle of page 2 “which refine training data bottom-up in a greedy way” bottom of page 4
>
> **Response**
> By “bottom-up" we mean a training data refinement process $X_n$ that is constructed by choosing the n-th input $x_n$ as the optimizer of an AL criterion that is evaluated, given $(X_{n-1}, y_{n-1})$. In contrast, we mean by “top-down" the optimization of an AL criterion in the asymptotic limit ($n\rightarrow \infty$). Then we refine $X_n$ according to this asymptotic criterion, which is sub-optimal for finite (and particularly small) n. However, the more we refine $X_n$, the more optimal it gets, guaranteed by the asymptotic optimality.
>
> * p7 can read: “As already noted in the introduction, the results to LFC and the optimal training density of LPS indicate their problem intrinsic nature, as they reflect no direct dependence on the LPS model.’’ — Except for the order Q correct?
>
> **Response**
> That is correct. Our argument here was meant as follows: For a function $f$ that is exactly (Q+1)-times differentiable, imposing the LPS model of order $Q$ is the canonical choice (as opposed to any smaller choice Q' < Q). In this sense, estimating the LFC of the LPS model of order Q is no arbitrary design choice, but driven by the actual smoothness of $f$. We clarify our interpretation in Sec. 3.2 in the revised manuscript.
>
> * Adding more precision on how the training objective is derived would be helpful. The $q_l$ is the marginal distribution of the l-th component of the MoE ? (Sec. 4.4.1) It is not straightforward to understand also the definition of the $P_{b,l}$. How should one interpret the marginalized? In the definition of the appearing is the test distribution and the appearing?
>
> **Response**
> We add more details on the training objective and fix some notation in the revised manuscript by first deriving it in Sec. 3.4.1 for a single expert and then using this derivation as a building block in Sec 4.4.1.
>
> * Could a simpler model for the gate functions be employed? With which loss on performance?
>
> **Response**
> The gate is a vector of classic GPs (see first response), which is hard to simplify further.
>
> * Could the authors clarify how to adapt their work to “on the fly" sampling, that is when there is not a $X_{pool}$ available from the start? That is crucial for a realistic implementation in ab initio force field as producing the unlabelled data (the configurations) as a pool a priori is required to compute the forces to run the MD simulation and is therefore not cheaper than getting also the labels.
>
> **Response**
> We take up the reviewer's concern in the discussion in Sec. 6 and outline a realistic implementation in Appendix H.3 of the revised manuscript.
>
> * It is arguably not fair to say that "current ML-based FFs typically rely on rather naive exhaustive sampling schemes to gather training data" page 22, cf for instance "Section 3.3.3. Adaptive Sampling" of the cited review by Keith et al 2022.
>
> **Response**
> We suggest weakening our statement to "some commonly deployed ML-based FFs rely on rather naive exhaustive sampling schemes to gather training data" in the revised manuscript.
>
> * The interpretability of LFC (discussion section) for high-dimensional input spaces should be relativized, as one cannot visualise this function except in low-dimension, or using relevant, previously known, low-dimensional projections. Or maybe the authors want to point out that one can locate high LFC points from the pool? In which case the precision would be useful.
>
> **Response**
> The conclusion of the reviewer is correct. What we meant is that the visualization is two-layered: When looking at a single point in input space, a visual assessment of the local structural complexity (e.g., a human can visually detect more complexity to the left of the Doppler function) is challenging in high-dimensional input spaces. Here, the scalar LFC value gives a human assessable quantitative description. Yet, the visualization of the LFC as a function over the whole high-dimensional input space at once is still impossible.
>  We adjusted the discussion in Sec. 6 in the revised manuscript.

---

> ### Comment · Reviewer_kvMm · 2023-11-09
>
> I thank the authors for their detailed answer. They have adressed my concerns except for the minor point of clarifying in the text the meaning of bottom-up/top-down.

---

> > ### Author Response · Authors · 2023-11-10
> > **Response to Reviewer kvMm**
> >
> > Dear reviewer,
> >
> > while we tried to answer your question regarding the meaning of bottom-up/top-down in our earlier response, we did not yet incorporate this clarification in the text. We will gladly add this clarification to the revised manuscript later today.

---

### Decision · Action_Editor_HaDk · 2023-11-17

**Recommendation:** Accept with minor revision

**Comment:**

This paper presents a novel approach to address inhomogeneities in real-world data, particularly focusing on variations in observation noise levels and structural complexities. The authors propose a model-agnostic active learning framework, leveraging local function complexity (LFC) estimates derived from Gaussian Process Regression (GPR). This approach aims to enhance robustness and scalability in handling large input space dimensions.

The paper makes a substantial contribution to the field of active learning by addressing key challenges in handling inhomogeneous data complexities. Its blend of theoretical innovation and practical application positions it as a valuable resource for researchers and practitioners in machine learning and related fields.

However, the reviewers still have some concerns about the writing (clarifying in the text the meaning of bottom-up/top-down, adding an algorithm in the main part), and the experiments (discussing the lower bound of RMSE). To further enhance the quality of the paper, the authors should address these issues in the revised version.

**Audience:**

Yes

**Claims And Evidence:**

Yes

---

> ### Author Response · Authors · 2023-12-08
> **Response to Action Editor HaDk**
>
> Dear Action Editor,
>
> Thank you for thoroughly considering our presented work, leading to this positive outcome.
> We address the final reviewer concerns in the uploaded camera-ready version as follows:
> * We elaborate on the meaning of the terms 'top-down' and 'bottom-up' after their first occurrence on pages 2 and 5, respectively.
> * We moved the main algorithm from the appendix to the beginning of section 4, currently displayed on page 12.
> * We added a discussion on the asymptotic lower bound of the RMSE after its definition in section 5 on page 18. We did so to the best of our knowledge, noting that we are at best able to estimate the law with respect to the training size up to a constant shift. This constant remains unknown, even when given ground truth. In this light, using an imaginary constant, we indicate this lower bound in Figures 6 and 8, as asked by Reviewer ZcVs.